# 3T-VASP: fast ab-initio electrochemical reactor via multi-scale gradient energy minimization

Jonathan P. Mailoa [1,2,3] ✉, Xin Li [3] & Shengyu Zhang [4] ✉

Ab-initio methods such as density functional theory (DFT) is useful for fundamental atomistic-level study and is widely used across many scientific fields, including for the discovery of electrochemical reaction byproducts. However, many DFT steps may be needed to discover rare electrochemical reaction byproducts, which limits DFT's scalability. In this work, we demonstrate that it is possible to generate many elementary electrochemical reaction byproducts in-silico using just a small number of ab-initio energy minimization steps if it is done in a multi-scale manner, such as via previously reported tiered tensor transform (3T) method. We first demonstrate the algorithm through a simple example of a complex floppy organic molecule passivator binding onto perovskite solar cell surface defect site. We then demonstrate more complex examples by generating hundreds of electrochemical reaction byproducts in lithium-ion battery liquid electrolyte (many are verified in previous experimental studies), with most trajectories completed within 50–100 DFT steps as opposed to more than 10,000 steps typically utilized in an ab-initio molecular dynamics trajectory. This approach requires no machine learning training data generation and can be directly applied on any new chemistries, making it suitable for ab-initio elementary chemical reaction byproduct investigation when temperature dependence is not required.

Ab-initio methods such as density functional theory (DFT) are instrumental in the study of physical sciences for many material systems across different scientific fields[1]. Unfortunately, the cost of these ab-initio methods scales quickly with the system size; for example, DFT generally scales as $N^3$ where N is a measure of the system size (such as the number of electrons or basis functions). While the computation cost of individual ab-initio steps is generally still accessible using hardware available today, the large number of DFT calls necessary to observe the desired event in a material simulation may limit its scalability. For example, accelerated high-temperature DFT-based ab-initio molecular dynamics (AIMD) is often used to simulate chemical reactions within Li-ion battery electrolytes and may require 10 ps simulation (10 or 20k DFT steps depending on the hydrogen isotope mass used) to observe 0–3 electrochemical reaction or redox events[2,3]. This may take several weeks per AIMD trajectory with the best hardware available today. Significant efforts have been made to alleviate this problem. Some of the most promising efforts to date involve the usage of reactive classical force field for enabling pre-determined chemical reactions[4–6], neural network force field (NNFF) to replace the computation cost of each DFT step with a significantly lower-cost neural network potential energy equivalent[7–13], and DFT-based metadynamics to reduce the number

[1]College of Computer Science and Artificial Intelligence, Wenzhou University, Wenzhou, Zhejiang, China. [2]Wenzhou University Artificial Intelligence and Advanced Manufacturing Institute, Wenzhou, Zhejiang, China. [3]Tencent Quantum Laboratory, Tencent, Shenzhen, Guangdong, China. [4]Tencent Quantum Laboratory, Tencent, Hong Kong, Hong Kong SAR, China. ✉e-mail: jpmailoa@alum.mit.edu; shengyzhang@tencent.com

of DFT steps (typically into just several ps despite lower temperatures) by bypassing energy barriers and enabling atoms to transition between states more quickly than it would otherwise have under accelerated high-temperature AIMD[14–16]. These state-of-the-art methods are crucial for studying chemical reactions involving temperature dependencies. Other promising reaction exploration approaches are temperature-independent, and they work by modifying the steady-state structure optimization method to escape the local energy minimum. For example, the artificial force-induced reaction (AFIR) approach works by adding multi-parameter empirical artificial forces between all-atom pairs[17]. The approach is further extended in the implementation of Chemoton 2.0 software, where the combination of brute force conformer sampling and empirical artificial force added on chosen reactive sites between candidate reactant molecule pairs are used to accelerate a reaction[18]. When the qualitative characteristics of the intended reactions are already known in advance, reaction transition states can be found using different methods, for example by using the single-ended growing string method which sequentially places intermediate nodes in the delocalized internal coordinates (bonds, angles, or torsions) relevant for reaching the desired reactive outcome[19]. A combination of temperature-dependent dynamics approach with vibrational energy added into the system for generating reaction and subsequent transition state analysis can also be performed[20,21]. We also note that our work has some similarities to internal coordinate methods recently used in non-periodic quantum chemistry software for structure minimization[22–24], albeit with some extensive differences (see Supplementary Discussion). More in-depth reviews of these reaction generation methods are available in the literature[25–27].

In this work, we propose that in some limited circumstances such as the exploration of elementary electrochemical reaction products in complex systems without considering temperature

dependencies, it is sufficient to simply perform multi-scale ab-initio energy minimization. We can enable the system to quickly reach lower-energy structure configurations typically inaccessible without tens of thousands of AIMD steps, mostly within just ~100 DFT steps (several hours). We utilize a previously developed tiered tensor transform (3T) method to transform material structure geometry in a multi-scale manner, enabling the system to explore some lower-energy structure configurations quickly[28]. Unlike the NNFF approach which requires diverse ab-initio training data generation (significant computation and personnel costs) when a large number of complex molecule chemistries are involved, our approach requires no ab-initio training data and can be immediately applied to any new chemistries. However, because our approach is just an energy minimization, it is primarily useful for exploring lower-energy states and cannot be used if temperature-dependent trajectory statistics are needed (such as ionic diffusivity or reaction rate). We first show a simpler example using a single complex floppy organic cation binding on $FAPbI_3$ vacancy surface defect to highlight the way multi-scale structure energy minimization is performed with the aid of PyTorch autograd functionality. We then show more complex examples such as large-scale lithium-ion battery liquid electrolyte electrochemical reduction and oxidation reaction trajectory generation, reproducing many experimentally known lithium-ion battery byproducts in just ~100 DFT steps as opposed to tens of thousands of steps commonly utilized in literature. We demonstrate that ethylene gas and carbonate ions are the primary byproducts of reductive electrolyte electrochemical reactions. We further demonstrate secondary oxidative electrochemical reactions out of these byproducts, producing acetaldehyde, carbon dioxide, ethylene dicarbonate, and other known complex organic species which will form the basis of battery solid-electrolyte-interphase (SEI) construction (Fig. 1d). We believe that the simplicity of the approach may find application in

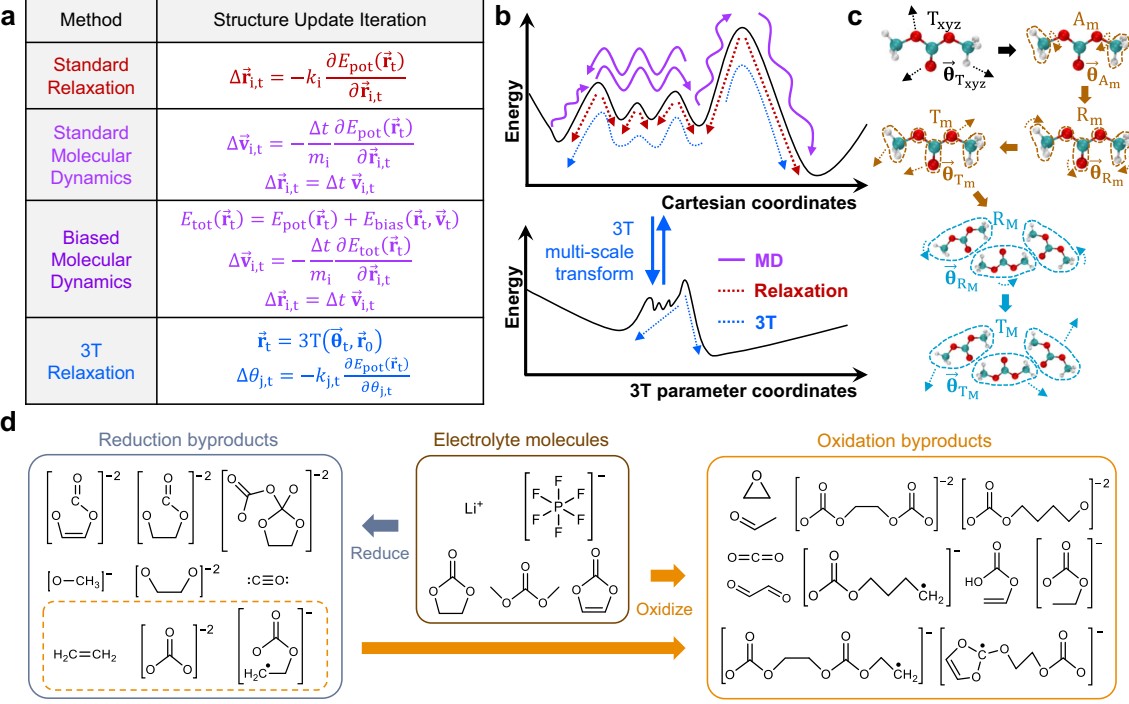

**Fig. 1 | High-level description of multi-scale tiered tensor transform (3T) relaxation procedure and its application. a** Structure update procedure difference between typical structure energy relaxation, molecular dynamics-based structure exploration, and multi-scale 3T structure energy relaxation. **b** 3T multi-scale structure transformation projects the energy minimization problem into higher-dimension 3T parameter hyperspace, where it is possible to explore farther

local energy minima landscape difficult to access by standard relaxation method or longer to reach using molecular dynamics. **c** 3T multi-scale structure transformation mode sequence used in this work. **d** Some of the 3T reduction and oxidation electrochemical reaction byproducts generated from the bulk electrolyte liquid mixture in this work.

more chemistry applications where difficult lower-energy state exploration is desired.

Unlike standard structure relaxation or molecular dynamics methods which directly modify an atom position at step $t$ ($\vec{r}_t$) based on the calculated system potential energy $E_{pot}(\vec{r}_t)$, we define a multi-scale structure transformation function $\vec{r}_t = 3T(\vec{\theta}_t, \vec{r}_0)$ where the parameter $\vec{\theta}_t$ at step $t$ controls the 3T function and is optimized by the gradient of potential energy $E_{pot}(\vec{r}_t)$ with respect to $\vec{\theta}_t$ ($dE_{pot}(\vec{r}_t)/d\vec{\theta}_t$, Fig. 1a). The 3T function is simply the composition of several individual structure geometry transformation functions performed in a hierarchical manner (Fig. 1c), as we will briefly discuss in the next section. This effectively projects the structure optimization problem from the Cartesian coordinates $\vec{r}_t$ into a higher-dimension parameter hyperspace $\vec{\theta}_t$, where coordinated multi-atom structure transformations can be performed efficiently in physically-inspired manners, often bypassing energy barriers that may be difficult to overcome using per-atom structure relaxation (Fig. 1b). Using our approach, it is possible to organically generate chemical reaction products within their explicit solvation structure without having to pre-determine the exact reactants and reaction mechanisms. The reaction byproduct trajectories generated by 3T can be used for further individual reaction energy barrier studies such as those commonly done using nudged elastic band[29,30] or for transition state analysis[31], making 3T a suitable complement to existing state-of-the-art methods which are typically done on systems with pre-determined reactants and mechanisms in the absence of explicit solvation structures.

## Results

### Single-molecule floppy organic cation binding on FAPbI₃ perovskite surface

We first demonstrate our approach on a simpler example problem such as the surface defect passivation of formamidinium lead iodide (FAPbI₃) perovskite solar cells. Perovskite solar cells and their tandem architecture with silicon solar cells have rapidly improved their solar energy conversion efficiencies over the past decade[32–35]. This rapid increase is partially due to the bulk defect tolerance of the perovskite lead iodide materials, and the recent cell efficiency advances have focused on further passivating the surface defects of the perovskite (Fig. 2a)[36]. Several passivating molecules have been investigated for passivating the A-site surface vacancy defect of the perovskite cells, and relatively large and complex molecules such as phenyl-triethylammonium (PTEA) cation has been found to be very good perovskite surface passivator[37]. This organic cation passivation mechanism does not necessarily involve covalent bonding with the perovskite surface and may be a simple non-covalent interaction as demonstrated by DFT[38], but the same 3T method can be used for electrochemical reaction product generation in the next section.

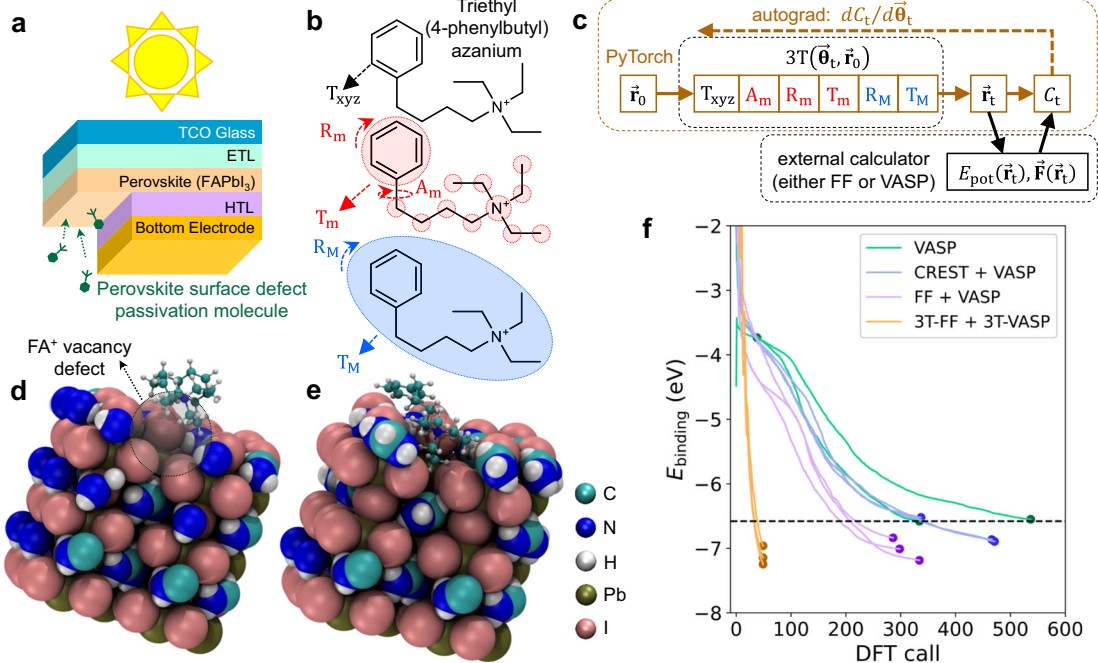

**Fig. 2 | 3T relaxation of floppy organic cation on FAPbI₃ surface vacancy defect site. a** Typical device structure of FAPbI₃ perovskite solar cell deposited on front transparent conducting oxide (TCO) glass sandwiched by electron and hole transport layers (ETL, HTL), with surface passivation molecules being applied on the bottom surface of the perovskite. **b** Tiered tensor transform (3T) multi-scale structure transformation modes enabled for TE4PBA organic cation during ab-initio 3T relaxation. **c** 3T optimization implementation utilizing PyTorch for gradient propagation and external calculator for potential energy and atomic forces calculation (done using either force field (FF) or Vienna Ab-initio Simulation Package (VASP)). The initial structure $\vec{r}_0$ is transformed into $\vec{r}_t$ using the 3T function (parametrized by $\vec{\theta}_t$), which is composed of structure transformation functions $T_{xyz}$, $A_m$, $R_m$, $T_m$, $R_M$, and $T_M$. After structure energy ($E_t$) and atomic forces ($\vec{F}_t$) evaluation by the external calculator, a proxy cost function $C_t$ is calculated in PyTorch, and gradient backpropagation is used to update $\vec{\theta}_t$. Repeating these optimization steps quickly minimizes the structure energy in a multi-scale manner.

The final minimized TE4PBA-FAPbI₃ structures obtained using **d** standalone VASP relaxation (cation still floats above the defect site) and **e** 3T-VASP relaxation (the cation embeds itself deep into the defect site), with the cation originally floating above the defect. **f** DFT binding energy vs the number of DFT call comparison between different methods in combination with VASP: standalone (VASP, green), Conformer-Rotamer Ensemble Sampling Tool (CREST + VASP, blue), force field (FF + VASP, purple), and 3T-VASP (3T-FF + 3T-VASP, orange). Standalone (VASP) fails to find a cation binding pose on the FAPbI₃ surface even after 500 DFT steps (cation floats above the defect site at a higher local energy minimum), while (CREST + VASP) only finds lower-energy cation binding pose which slightly enters the defect site after 5 days of conformation search. Both (FF + VASP) and (3T-FF + 3T-VASP) can find a cation binding pose that is deeply embedded into the defect site with the lowest energies, but (3T-FF + 3T-VASP) can do so in significantly less DFT calls (<50). Source data are provided as a Source Data file.

Virtual screening of other complex cations may be useful to discover better molecules for perovskite passivation than PTEA, where we can first pre-screen whether a candidate passivating molecule binds into the $FAPbI_3$ defect site or not before further experimental verification.

Triethyl(4-phenylbutyl)azanium (TE4PBA) is a good example of such a cation candidate. TE4PBA shares several characteristics with PTEA, such as the hydrophobic phenyl group and the three carbon chains attached to the $N^+$ atom on the opposite end. The cation is relatively large and floppy however, and it is unclear whether it can bind into the surface vacancy defect site of $FAPbI_3$ or not (Fig. 2b). Standard DFT relaxation procedure based on conjugate gradient atomic forces is employed for three different initial TE4PBA conformations placed directly on top of the defect site. All three DFT structure relaxation attempts ended with energy-minimized structures where the cation floats above the formamidinium cation ($FA^+$) vacancy defect site (Figs. 2d, and S1). In the standard relaxation technique, each relaxation step with respect to the individual nuclear coordinates attempts to minimize the total system energy, making it difficult to perform a coordinated structure transformation that can overcome uphill energy barriers such as those presented by the process of embedding a passivating molecule into the perovskite surface defect site.

In the 3T approach, the TE4PBA structure is automatically segmented hierarchically (see "Methods" section, Fig. 2b) into different micro-groups (based on its rotatable bonds) and a macro-group (the entire molecule)[28]. In addition to the atom-level translation used in the standard relaxation approach ($T_{xyz}$), different structure transformation modes are enabled: $A_m$, $R_m$ and $T_m$ for micro-groups' sidechain rotations, center rotations, and translations; $R_M$ and $T_M$ for macro-groups' center rotations and translations (see Fig. 2b, Supplementary Discussion). We set our 3T function as a sequential (output from one transformation is input to the next transformation) and hierarchical (first atom level, then micro-group level, and finally macro-group level) compound geometry transformation of these six functions, each governed by the parameter $\vec{\theta}_t$ at step $t$ (Figs. 1c and 2c). In mathematical terms, we have:

$$\vec{r}_{temp_1, t} = T_{xyz}\left(\vec{r}_0, \vec{\theta}_{T_{xyz}, t}\right) \tag{1}$$

$$\vec{r}_{temp_2, t} = A_m\left(\vec{r}_{temp_1, t}, \vec{\theta}_{A_m, t}\right) \tag{2}$$

$$\vec{r}_{temp_3, t} = R_m\left(\vec{r}_{temp_2, t}, \vec{\theta}_{R_m, t}\right) \tag{3}$$

$$\vec{r}_{temp_4, t} = T_m\left(\vec{r}_{temp_3, t}, \vec{\theta}_{T_m, t}\right) \tag{4}$$

$$\vec{r}_{temp_5, t} = R_M\left(\vec{r}_{temp_4, t}, \vec{\theta}_{R_M, t}\right) \tag{5}$$

$$\vec{r}_t = T_M\left(\vec{r}_{temp_5, t}, \vec{\theta}_{T_M, t}\right) \tag{6}$$

with $\vec{r}_{temp, t}$ representing the intermediate coordinates of all atoms in the system between 3T structure transformation modes (see Supplementary Discussion for each equation details). Initially, we have $\vec{\theta}_0 = 0$ and hence the initial transformations are identity functions. The resulting geometry $\vec{r}_t$ is evaluated using an external software to calculate the potential energy $E_t = E_{pot}(\vec{r}_t)$ and atomic forces $\vec{F}_t = \vec{F}(\vec{r}_t)$ of the system. To determine the $\vec{\theta}_t$ parameter update, we should compute the gradient of $E_t$ with respect to each of the $\vec{\theta}_t$ above. The gradients can then be used to determine the updated $\vec{\theta}_{t+1}$, in combination with the gradient scaler $\vec{k}_t$ generated by any suitable optimization algorithm. To simplify notation, we will first flatten the $\vec{\theta}_t$ where

j indicates its flattened indices:

$$\vec{\theta}_t = \text{flatten}\left(\left[\vec{\theta}_{T_{xyz}, t}, \vec{\theta}_{A_m, t}, \vec{\theta}_{R_m, t}, \vec{\theta}_{T_m, t}, \vec{\theta}_{R_M, t}, \vec{\theta}_{T_M, t}\right]\right) \tag{7}$$

$$dE_t/d\vec{\theta}_t = \left[\partial E_t/\partial\theta_{1, t}, \ldots, \partial E_t/\partial\theta_{j_{max}, t}\right] \tag{8}$$

$$\theta_{j, t+1} = \theta_{j, t} - k_{j, t}\frac{\partial E_t}{\partial\theta_{j, t}} \tag{9}$$

While the last two steps above are very cumbersome to implement manually, in practice this looks just like a neural network parameter update. The differentiation can be automated using the PyTorch autograd function, while the parameter update can be automated using any of PyTorch built-in optimizers. In our previous work, $E_{pot}$ was the minimization cost function and the evaluation is fully performed in PyTorch, making $\vec{\theta}_t$ gradient-based updates on each step straightforward: $dE_t/d\vec{\theta}_t = \text{autograd}(E_t)$[28]. In this work, the usage of external energy calculator such as classical force field or DFT software Vienna Ab-initio Simulation Package (VASP) requires us to develop a chain-rule-based cost function $C_t = -\sum_i r_{i, t}F_{i, t}$, with i representing atom indices. Performing $dC_t/d\vec{\theta}_t = \text{autograd}(C_t)$ will store identical PyTorch gradients to $\vec{\theta}_t$ as if the whole $dE_t/d\vec{\theta}_t = \text{autograd}(E_t)$ calculation is performed inside PyTorch instead ($F_{i, t}$ needs to be treated as a gradient-less constant, see "Methods" section). Afterward, we simply use PyTorch Adam optimizer to calculate $\vec{\theta}_{t+1} = \text{optimizer\_step}\left(\vec{\theta}_t, dC_t/d\vec{\theta}_t\right)$, in a manner identical to any standard neural network training using PyTorch to minimize a loss function $C$.

We randomly place three different conformations of TE4PBA on top of the $FA^+$ defect site and run classical force field-based 3T (3T-FF, negligible computation cost) for 5 cycles with 200 steps each followed by DFT-based 3T (3T-VASP, because DFT software VASP is used) for 50 steps. At the end of 3T-FF's last cycle step $T$, we simply use the final 3T-FF structure $\vec{r}_{3T-FF, T}$ as the new initial structure for 3T-VASP $\vec{r}_{3T-VASP, 0}$ which ensures continuous 3T trajectory (see "Methods" section). We note that the force field should be of sufficient quality to produce qualitatively reasonable geometries because 3T-VASP minimization should at least be started from physically meaningful $\vec{r}_{3T-FF, T}$. Systems containing transition metal may require methods with higher accuracy than FF. For all three random initial conformations, TE4PBA is deeply embedded into the surface vacancy defect site (Fig. 2e). If we define the binding energy $E_{binding} = E_t - E_{surface} - E_{mol}$ with $E_{surface}$ and $E_{mol}$ representing the standalone minimized defective-$FAPbI_3$ surface and TE4PBA molecule VASP energies respectively, we can see that the 3T structures reach energies 0.38–0.66 eV lower than the best standard DFT-minimized structures (which converges after more than 300 and 500 DFT calls) within less than 50 DFT calls. We note that if we first utilize the force field pipeline we have developed to relax the structure (just per-atom relaxation, without 3T) prior to running standard VASP relaxation (FF + VASP in Fig. 2f), this baseline performance can be considerably improved. This improved baseline will also be able to generate low-energy structures with the cation being deeply embedded into the defect site (Fig. S3), equivalent to those generated by 3T-VASP, albeit using significantly more (300-350) DFT calls. Additional transition state and energy barrier analysis using snapshots extracted from the 3T energy minimization trajectory is available (Fig. S4).

We have also included an additional baseline comparison where the TE4PBA cation conformation search on the perovskite surface is performed using the Conformer-Rotamer Ensemble Sampling Tool (CREST)[39] before VASP relaxation is performed (see Supplementary Discussion). This method requires a relatively long conformation search (5 days) and slightly less DFT calls than standard VASP relaxation. However, it still requires significantly more DFT calls than 3T-VASP. The

generated cation structure still has relatively higher energy than 3T relaxation (0.36–0.73 eV higher than the best 3T structure) and can only slightly enter the defect site (CREST + VASP in Fig. 2f, Fig. S2).

## Electrochemical reaction in Bulk Li-ion battery electrolyte liquid

The example above demonstrates the benefit of 3T for simpler cases such as single complex organic molecule relaxation on a material surface. We believe however that 3T can be more useful for the fast exploration of electrochemical reaction byproducts in significantly more complex material mixtures, such as lithium-ion battery liquid electrolytes. Lithium-ion batteries have experienced rapid cost reduction and stability improvements over the past decade[40–43], and their utility in mobility electrification is set to continue to rise in the foreseeable future[44]. One fundamental component in the production of lithium-ion batteries is the formation of solid-electrolyte-interphase (SEI) layers, which strongly contributes to the battery's stability and performance over its lifetime. However, the formation of SEI layers is poorly understood because it is difficult to experimentally determine its compositions or formation process (which varies depending on the electrolyte and additive molecule compositions)[45,46]. Studying the electrochemical reaction byproducts in such electrolyte mixtures is useful for enabling better SEI formation processes. Due to the large number of molecular species and diverse reactions in the electrolytes, it is also difficult to train a good NNFF for accelerating the AIMD simulation of these electrolytes because the number of required local environment ab-initio training data increases significantly with the increasing number of molecule/reaction combinations.

Using the 3T approach, we propose that we can study and replicate experimentally observed lithium-ion electrolyte electrochemical reaction byproducts by simply putting the component molecules in a box and calling 50–100 DFT steps to generate the reactions (a fast ab-initio electrochemical reactor). Previous ab-initio nanoreactor approaches utilizes AIMD[47], while our approach only uses static DFT calls. First, we consider the case of electrolyte electrochemical reduction. We create a periodic box of $14 \times 14 \times 14 \, \text{Å}^3$ (Fig. 3a) and randomly place ethylene carbonate (EC), dimethyl carbonate (DMC), vinylene carbonate (VC), lithium cation ($Li^+$), and hexafluorophosphate anion ($PF_6^-$). See the "Methods" section and Fig. S5 for the automated 3T hierarchical segmentation setup for these molecules. The number of EC and DMC molecules are kept constant ($n_{EC} = n_{DMC} = 10$), while the rest of the molecule counts are randomized ($n_{VC} = 1$–3, $n_{Li^+} = 2$–8, $n_{PF_6^-} = 1$–2). These electrolyte systems (mass density = 1.24–1.46 g cm$^{-3}$) are intentionally not charge-compensated (additional ions are not added to neutralize the system). This approach follows a previous AIMD-based study of electrochemical redox in liquid[2,3]. Inter-molecule electron transfer happens in most of the reactions that we observe (Figs. S10, S12, S13). We run 3T-FF for 5 cycles with 200 steps each to disperse the molecules inside the box while disabling chemical reactions, followed by 3T-VASP for 5 cycles with 50 steps each to further relax the molecules while enabling electrochemical reactions to proceed (see "Methods" section). The transition from one 3T cycle to the next is identical to the transition from 3T-FF to 3T-VASP in our perovskite example, making the trajectory fully continuous. Like any other energy minimization techniques, the 3T structure eventually settles into a local energy minimum ($(dE_{pot}(\vec{r}_t))/d\vec{\theta}_t \approx 0$). However, the transition into a new 3T cycle transforms the energy landscape into a completely new optimization problem with non-zero $\vec{\theta}_t$ gradient for further optimization (Fig. S7) despite identical $\vec{r}_t$ during the 3T cycle transition.

We run 63 trajectories for this 3T electrolyte mixture electrochemical reductions and produce byproducts such as carbonate ions ($CO_3^{2-}$), ethylene gas ($C_2H_4$), open-ring EC radical (oEC$^-\bullet$), carbon monoxide (CO), ethane-1,2-diolate[48], methoxide[49], etc. (Fig. S10, Table S1) as well as their reaction mechanism trajectories. First, we separate these 63 trajectories based on the excess charge

($n_{excess} = n_{Li^+} - n_{PF_6^-}$) of the system and analyze the system's final 3T-minimized states. For trajectories with $n_{excess} = 0$, neither reduction reaction nor charge reduction happens in the system (Fig. 3b–c). This makes experimental sense because we know that simple LiPF$_6$ salt solvation in battery electrolytes should not spontaneously produce electrolyte decomposition reactions. For $n_{excess} = 1$, we start seeing reduction reactions (EC and VC), and some EC molecules are also getting reduced into their [−1] charge states (Fig. S11). At higher $n_{excess}$ levels, more reduction reactions start forming (dominated by EC reactions forming $CO_3^{2-}$ and $C_2H_4$), with an average of $\sim \frac{1}{2} n_{excess}$ reactions per trajectory for $2 \leq n_{excess} \leq 4$. Similarly, DMC and finally, $PF_6^-$ molecules start reacting at higher $n_{excess}$ levels. In this VASP DFT simulation with a neutral charge, the formal charges on Li and $PF_6$ are preferentially $Li^+$ and $PF_6^-$, and $n_{excess}$ electrons must be distributed to the remaining molecules (higher $n_{excess}$ levels correspond to more reducing environments). The decomposition of EC into $CO_3^{2-}$ and $C_2H_4$ (our primary byproducts) requires 2 electrons, which is why we observe no electrochemical reactions for $n_{excess} < 2$. We note that this specific reaction is known to occur much more quickly in AIMD (within 50 fs of rerun after frames are extracted from a 17 ps AIMD trajectory and reconfigured to modify their charges) compared to other reactions which require much longer additional AIMD steps (within ps time frame)[2], which is consistent with our finding. The second most abundant reduction gas byproduct species is CO. This compares favorably to experimental observation with $C_2H_4$ being the most abundant battery electrolyte reduction reaction gas byproducts, followed by CO[50]. Similar to the experimental observation, we do not observe any $C_2H_6$ gas formation when the EC : DMC electrolyte mixture is used[51]. We also note that $PF_6^-$ is the most difficult component to undergo reduction reaction in our system (rare reaction), which is consistent with experiments describing LiPF$_6$ decomposition being more easily observed at elevated temperatures[52–54]. The 3T $PF_6^-$ decomposition produces the PF$_3$ gas, which has previously been predicted and reported experimentally[6,55–57]. There will be additional decomposition products due to the presence of water molecules which are not included in our 3T system. Including all the intermediate reaction and charge reduction events, most of these 3T-VASP reduction reactions (97%) and charge reductions (90%) happen within the first 100 3T DFT calls (Fig. 3d–e). See Figs. S8–S9 for the details on the molecule reaction detection algorithm.

We further investigate the reverse situation (oxidating environment) using 3T. In addition to the five input species that we use in the reduction case, we add $CO_3^{2-}$, $C_2H_4$, and oEC$^-\bullet$ as input molecules to explore the EC decomposition pathways. The first two are relatively abundant in the electrolyte (from EC reduction), while the last one is a highly reactive radical species that has been proposed to be responsible for EC dimerization reactions[58]. In particular, the generated oEC$^-\bullet$ radical may very quickly attack nearby electrolyte compounds by reducing them, with the oEC$^-\bullet$ radical itself being oxidized in the process. We set $n_{EC} = n_{DMC} = 10$, $n_{VC} = 1$–3, $n_{C_2H_4} = n_{CO_3^{2-}} = 1$–2, $n_{oEC^-\bullet} = 1$–2, $n_{Li^+} = 1$–4, and $n_{PF_6^-} = 1$ to artificially increase the probability of reactions involving EC reduction byproducts. In this oxidative reaction case (larger number of atoms), we only perform 3T-FF for 5 cycles with 200 steps each followed by 3T-VASP for 3 cycles with 50 steps each. We separate the 3T trajectories based on the excess charge, this time defined as $n_{excess} = n_{PF_6^-} + 2n_{CO_3^{2-}} + n_{oEC^-\bullet} - n_{Li^+}$. In this oxidizing environment case, a VASP DFT with neutral system charge will need to extract $n_{excess}$ electrons from the molecules. For $n_{excess} = 0$, there is no electrochemical reaction, and the molecules stay as their original input molecule species (Fig. 4b). For $n_{excess} \geq 1$, we see electrochemical reactions dominated by reactions involving oEC$^-\bullet$. Most of these reactions (including the intermediate reactions) are finished by 3T-VASP step 150, although it is likely that a small number of additional intermediate reactions will be observed with an additional 50-step 3T-VASP cycle (Fig. 4c). Further analysis shows that all

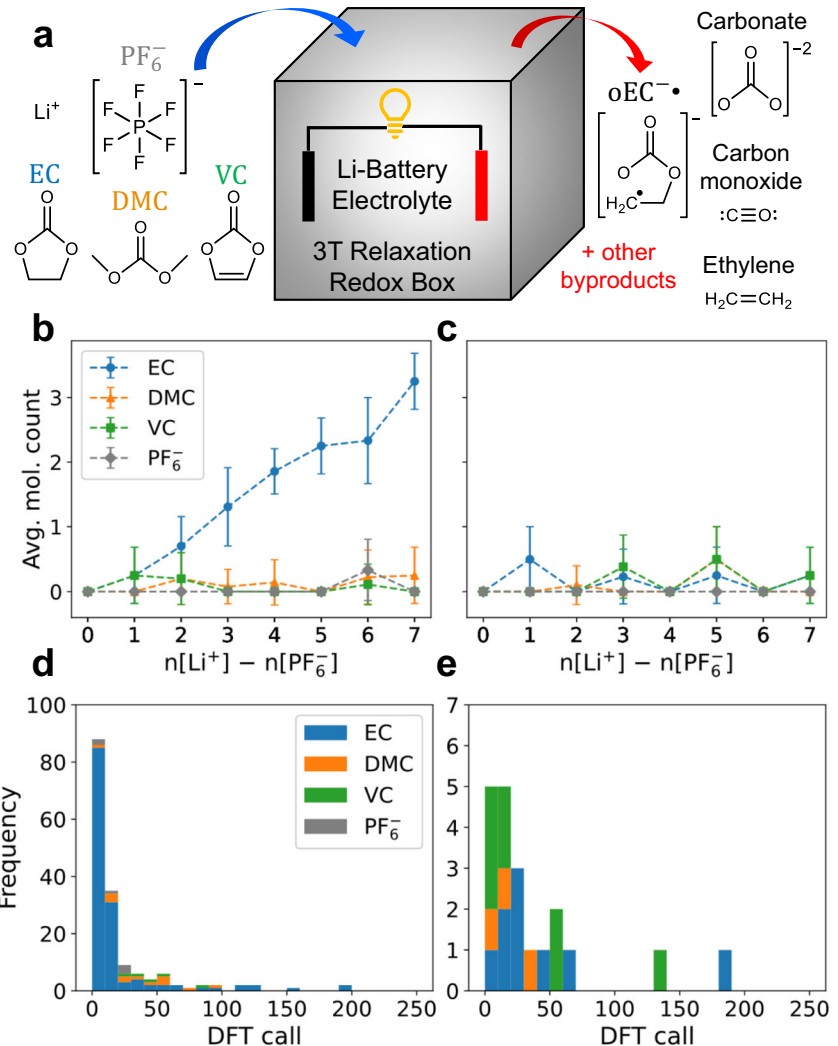

**Fig. 3 | Fast Li-ion battery liquid electrolyte reduction reactions using 3T energy relaxation. a** Liquid battery electrolyte molecule reactants ($Li^+$, $PF_6^-$, EC, DMC, and VC) and example byproduct molecules generated within $14 \times 14 \times 14$ Å³ periodic box. The fast 3T reaction enables us to generate 63 bulk liquid reaction trajectories with random initial configurations to reduce the electrolyte molecules. Average number of molecules per trajectory which experience unique reduction reaction (**b**) and charge reduction (**c**) events in the bulk electrolyte, summarized from hundreds of reactions within the trajectories and separated by the number of excess charges, defined as $n_{excess} = n_{Li^+} - n_{PF_6^-}$. The numbers of sample trajectories for each excess charge from 0–7 are [8, 8, 10, 13, 7, 4, 9, 4] respectively, and the data are presented as mean values ± SD. Ethylene gas and carbonate ions are by far the most abundant immediate byproducts, in line with experimental observations. The DFT step count when these reduction reactions and charge reductions occur are shown in the stacked histogram (**d** and **e**) respectively, with a majority occurring within just 50 DFT calls. Source data are provided as a Source Data file.

the observed oxidation reactions always involve either $oEC^- \bullet$ or $CO_3^{2-}$ as reactant (Figs. S9–S10, Table S1). The primary oxidation reaction byproducts are primarily $CO_2$ gas[59], acetaldehyde[60], and ethylene oxide. We also observe ethylene dicarbonate and glyoxal formation (both are experimentally known[58,59]) in addition to many long hydrocarbon chain oxidation byproducts which are more difficult to be confirmed experimentally because they are not released as gas molecules during the electrolyte decomposition experiment (Fig. 4a, Figs. S12–S13). Both $CO_3^{2-}$ and $oEC^- \bullet$ react with each other and with other molecules such as EC, VC, and ethylene, although only $oEC^- \bullet$ reacts with DMC. It has been suggested that ethylene oxide is a moderately stable output molecule that quickly isomerizes into acetaldehyde[61]. However, $CO_2$, ethylene oxide and acetaldehyde have all been experimentally observed as EC : DMC electrolyte oxidation decomposition gas byproducts[62–64]. Nuclear magnetic resonance (NMR) experiment has previously indicated that glyoxal is an electrolyte oxidation decomposition product[59], and our result further

suggests that glyoxal is produced through VC reaction with $CO_3^{2-}$ (Fig. S13). We also observe molecules such as EC, DMC, and VC occasionally experiencing other processes such as charge oxidation to their [+1] charge states (Fig. S14). Additional preliminary results for 3T electrochemical reactions on lithium metal and lithiated graphite anode surfaces are shown in Fig. S15. A brief comparison with an AIMD-VASP baseline, highlighting 3T-VASP reaction diversity and DFT call reduction, is shown in Fig. S16.

## Discussion

### 3T coordinated multi-scale optimization speed

In this section, we briefly discuss about the relatively fast convergence speed of our 3T multi-scale optimization approach. In our work, this fast convergence speed iteration is observed during both 3T-FF and 3T-VASP stages, and it seems to be relatively independent from the underlying atomic forces/energy calculator being used. During the FF stage, we observe 3T being capable of rapidly changing the floppy

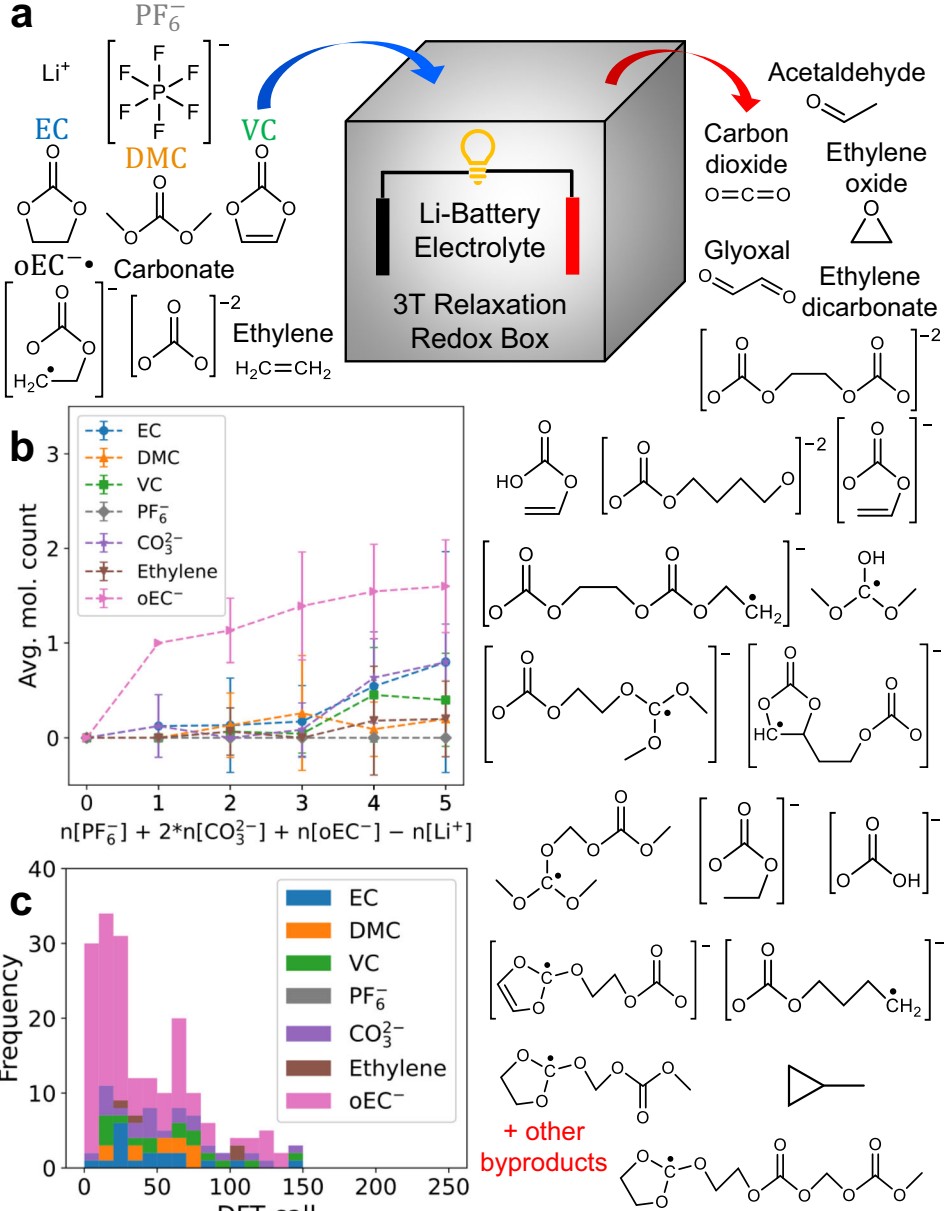

**Fig. 4 | Fast Li-ion battery liquid electrolyte oxidation reactions using 3T energy relaxation. a** Liquid battery electrolyte molecule reactants (Li$^+$, PF$_6^-$, EC, DMC, VC, oEC$^-$•, CO$_3^{2-}$, and C$_2$H$_4$) and example byproduct molecules generated within $14 \times 14 \times 14$ Å$^3$ periodic box. **b** Average number of molecules per trajectory which experience unique oxidation reaction events in the bulk electrolyte, summarized from hundreds of reactions within the trajectories and separated by the number of excess charges, defined as $n_{excess} = n_{PF_6^-} + 2n_{CO_3^{2-}} + n_{oEC^-\bullet} - n_{Li^+}$. The numbers of sample trajectories for each excess charge from 0–5 are [1, 8, 15, 23, 11, 5] respectively, and the data are presented as mean values ± SD. The DFT step count when these oxidation reactions occur is shown in the stacked histogram (**c**), with a large majority of reactions completed within 150 DFT calls. Source data are provided as a Source Data file.

cation conformations on the perovskite surface and being able to rapidly disperse the electrolyte molecules into its liquid form inside the periodic boundary condition box without having to do any MD. During the VASP stage, we observe 3T being capable of rapidly minimizing perovskite-cation energy or initiating and converging a chemical reaction between nearby molecules. We note that reactions do not happen during the 3T-FF stage, as our classical force field does not allow chemical reactions to happen. However, all these fast convergence behaviors will be turned off if we disable micro-group (A$_m$, R$_m$ and T$_m$) and macro-group (R$_M$ and T$_M$) level structure transformations and optimizations. In such case, we end up with a standard per-atom optimization algorithm that suffers from atoms being trapped in local energy minimums. When the micro-group and macro-group level optimizations are enabled, the 3T optimizer can automatically optimize the rotation and translation parameters for many atoms in a coordinated manner, forcibly moving an atom up its local energy barrier hill while still taking the atom's local energy gradient into account during the optimization. While the gradient calculation is done by 3T in the PyTorch framework using autograd, the actual optimization step is performed by PyTorch Adam optimizer (which works better than another Hessian-based gradient approach on our bulk electrolyte systems, see Figs. S18–S19). This also allows our workflow to further benefit from neural network advances made by the machine learning community.

## Summary

In summary, we have demonstrated that simple ab-initio energy minimization can be used to quickly reach low-energy states even in complex material systems if the energy minimization is done in a multi-scale manner. In the present work, we utilize the 3T algorithm to hierarchically transform the initial structure into its low-energy geometry, utilizing both classical force field and VASP DFT software as its multi-scale optimization cost function calculator. We demonstrate 3T ability on the fast ab-initio binding of complex organic molecule on perovskite surface defect, as well as on the generation of hundreds of elementary electrochemical reaction products in lithium-ion battery electrolyte liquid mixtures, with reactions mostly completed within 100–150 DFT calls (a fast ab-initio electrochemical reactor). The algorithm is decoupled from the cost function calculator, making it straightforward to plug in other computationally expensive ab-initio methods more suitable to study specific scientific problems of interest (3T-NWChem for example, see Fig. S17).

We envision that 3T-VASP (or its future variants when coupled with lower or higher accuracy ab-initio methods) can be useful for the exploration of chemical reactions in complex systems including those involving interfaces such as battery SEI formation, reactions on catalyst surfaces, or coating deposition and degradation on semiconductor device surfaces. Our early effort in this direction is shown in Fig. S15. However, we note that a relatively good force field is necessary to first perform 3T-FF and generate physically reasonable initial structure for the subsequent 3T-VASP. In addition, while our workflow can efficiently explore nearby local energy minimum structures, it is currently incapable of incorporating temperature dependency for the generated reaction products.

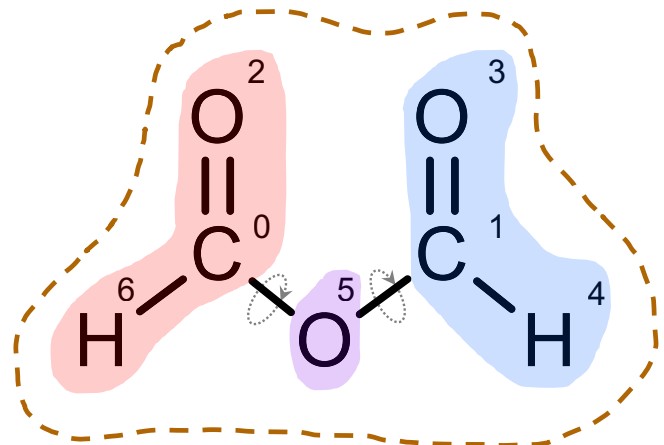

**Fig. 5 | Schematic diagram of hierarchical segmentation of micro-group and macro-group structures.**

## Methods

### Micro-group and macro-group hierarchical segmentation

For the defective-FAPbI$_3$ surface structure, we designate the FA$^+$, Pb$^{2+}$, and I$^-$ cations and anions as their own individual micro-groups, while only allowing 3T structure transformation to be performed on the first 3 layers of atoms on the A-site vacancy surface (15 FA$^+$, 8 Pb$^{2+}$, and 32 I$^-$ for a total of 55 micro-groups without any macro-group level segmentation). For the molecules (both the perovskite and battery electrolyte examples), the molecules' segmentation is done by converting the molecules into *.mol2 file format, utilizing RDKit to obtain their rotatable bond indices[65], and separating the molecules into bond subgraphs separated by their rotatable bonds (micro-group m). All micrographs in a molecule are further grouped into a macro-group M. For example, an entire molecule with 7 atom indices [0,1,2,3,4,5,6] may be segmented into micro-group m = [0,2,6], [1,3,4], and [5] and macro-group M = [0,1,2,3,4,5,6]. For computational convenience reasons, we encode M as a nested list of m instead ([[0, 2, 6], [1, 3, 4], [5]]). Segmentation of an example molecule (formic anhydride) is illustrated below (Fig. 5).

### Classical force field assignment

The classical force field epsilon and sigma parameters for FAPbI$_3$ (perovskite) and Li$^+$ cations (electrolyte) are taken from OpenKIM's 'universal' shifted Lennard-Jones model[66]. The partial charges for FAPbI$_3$ are assigned based on previous work[67], while the partial charge for Li$^+$ cations are set to +0.5. The classical force field parameters for individual electrolyte molecules are automatically assigned using SwissParam webserver[68]. CO$_3^{2-}$ and oEC$^-\bullet$ are the exceptions because SwissParam webserver will encounter errors while assigning the force field on these unusual molecules. We only need the classical force field to be good enough to maintain the molecules' geometry while preventing chemical reactions, so we assign force field parameters with slight modifications from those assigned by the LigParGen webserver[69]. LigParGen cannot handle oEC$^-\bullet$ radical, so we had to input oEC$^{2-}$ and manually modify the produced force field to maintain

the planar improper angle geometry of the $sp^2$ radical carbon group instead of $sp^3$. As for the CO$_3^{2-}$ group, LigParGen is similarly unable to handle this; because of that, we have to extract the charge/bond/angle/improper parameters corresponding to the CO$_3$ group of oEC$^{2-}$ and modify the parameters slightly to ensure correct CO$_3$ planar geometry symmetry (see Supplementary Table 1–2 for the manually assigned force field parameters and Figure S6). The force field parameters produced by LigParGen is in the LAMMPS format[70], which we further converted into the LAMMPS force field styles compatible with our work (the default styles generated by GROMACS-LAMMPS conversion using InterMol)[28,71–73]. These manual parameters are only designed to maintain decent 3T-FF molecule geometries prior to the more robust ab-initio 3T-VASP, and should not be used for actual classical force field MD.

### 3T chain-rule gradient coupling with external software

To minimize the material structure's energy to facilitate chemical reactions, 3T relies on calculating the gradient $dE_t/d\vec{\theta}_t = \text{autograd}(E_t)$ in PyTorch[74], followed by optimization of $\vec{\theta}_t$ using the Adam optimizer (learning rate = 0.03)[75]. In our previous work where all the structure transformation operation $\vec{r}_t = 3T(\vec{\theta}_t, \vec{r}_0)$ and potential energy calculation $E_t = E_{pot}(\vec{r}_t)$ are all done inside PyTorch[28], $\vec{\theta}_t$ gradient is calculated using a simple autograd function which produces the gradient $dE_t/d\vec{\theta}_t$, where its individual components can be expanded into: $dE_t/d\theta_{j,t} = \sum_i \frac{dE_t}{dr_{i,t}} \frac{dr_{i,t}}{d\theta_{j,t}} = -\sum_i F_{i,t} \frac{dr_{i,t}}{d\theta_{j,t}}$. Note that j represents the indices of the 3T parameter $\vec{\theta}_t$, while i represents the individual atom indices. When we need to calculate $E_t$ using external software such as VASP, this straightforward PyTorch autograd functionality is no longer available. We have instead developed a PyTorch cost function $C_t = -\sum_i r_{i,t} F_{i,t}$ where $F_{i,t}$ is treated as gradient-less NumPy constant[76] within PyTorch to take advantage of the derivative chain rule. Under this arrangement, we calculate $\vec{r}_t$ in PyTorch, use it as the input to external software for $E_t$ and $\vec{F}_t$ calculation, followed by the calculation of $C_t$ in PyTorch while treating $\vec{F}_t$ as constant multipliers. We then perform autograd on $C_t$ to produce $dC_t/d\theta_{j,t} = -\sum_i \frac{dr_{i,t}}{d\theta_{j,t}} F_t$, which is identical to the fully-PyTorch $dE_t/d\theta_{j,t}$ above. We note that while 3T-FF 'external software' $E_t$ and $\vec{F}_t$ calculations are also done within PyTorch (unlike 3T-VASP where this is done in the VASP software), we cut off their gradients by converting the PyTorch tensors into NumPy arrays and then back into PyTorch gradient-less tensors before $C_t$ calculation to ensure 3T-FF gradient correctness.

### Inter-cycle 3T parameter hyperspace transition

Just like any other structure energy minimization methods, 3T structures will eventually fall into a local energy minimum with

$dE_{\text{pot}}(\vec{r}_{\text{n,T}})/d\vec{\theta}_{\text{n,T}} \approx 0$, with n denoting a cycle index. At this point in the cycle (step $t = T$), further DFT calls will produce negligible additional 3T structure transformations, making the optimization stagnant or inefficient. We can mitigate this problem by simply projecting the optimization into a new 3T $\vec{\theta}$ parameter hyperspace (a new 3T cycle). Note that $dE_{\text{pot}}(\vec{r}_{\text{n,T}})/d\vec{\theta}_{\text{n,T}} \approx 0$ does not mean $dE_{\text{pot}}(\vec{r}_{\text{n+1,0}})/d\vec{r}_{\text{n+1,0}} \approx 0$. When we transition into a new 3T cycle, we re-initialize the 3T inputs with $\vec{r}_{\text{n+1,0}} = \vec{r}_{\text{n,T}}$ and $\vec{\theta}_{\text{n+1,0}} = 0$. Under this problem re-initialization framework, the trajectories are continuous ($\vec{r}_{\text{n+1,0}} = \vec{r}_{\text{n,T}}$). However, their gradients are no longer identical: $dE_{\text{pot}}(\vec{r}_{\text{n+1,0}})/d\vec{\theta}_{\text{n+1,0}} \neq dE_{\text{pot}}(\vec{r}_{\text{n,T}})/d\vec{\theta}_{\text{n,T}}$ because the two 3T structure transformations that generate identical $\vec{r}$ represent the same function with different inputs and parameters. The impact is most immediately obvious during 3T-FF of liquid electrolyte, where at the end of a cycle the liquid molecules look as if they are already frozen in a local energy minimum, but immediately disperse further into a more uniform liquid molecule distribution after transitioning into a new 3T cycle (Fig. S4). This inter-cycle transition is utilized during the transition between any cycle transition (between 3T-FF cycles, from 3T-FF into 3T-VASP, and between 3T-VASP cycles). It is relatively inexpensive to over-minimize the structure during 3T-FF cycles, but we suggest that the user first perform several 3T-VASP cycles with 50 steps each to determine the appropriate cycle/step number settings for new chemistries before performing large-scale 3T-VASP trajectory generations.

## Hardware setup

The 3T and AIMD computations in this work are done using Intel Xeon Cascade Lake 8255C machines with 4 Nvidia V100 GPU cards on them.

## Reporting summary

Further information on research design is available in the Nature Portfolio Reporting Summary linked to this article.

## Data availability

All the raw and source data generated in this study have been deposited in a GitHub repository under accession code https://www.github.com/jpmailoa/External_3T[77]. Source data are also provided with this paper. Source data are provided with this paper.

## Code availability

All the code used in this study has been deposited in a GitHub repository under accession code https://www.github.com/jpmailoa/External_3T[77] and preserved as a CodeOcean compute capsule under accession code https://codeocean.com/capsule/9554531/tree/v1[78].

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

## Acknowledgements

The authors thank Dr. Jingxuan Zhang for insightful discussions on Li-ion battery electrolyte electrochemical reaction simulation setup. This work is performed in Tencent Cloud and Tencent Elastic First-Principles Simulation (TEFS) platform. Ming Shao from Tencent Quantum Laboratory is acknowledged for TEFS and NEB technical support.

## Author contributions

J.P.M. is responsible for the 3T algorithm development and integration with VASP and the TEFS platform. L.X. and J.P.M. are responsible for the reaction trajectory analysis. J.P.M. prepares the VASP DFT simulation settings. S.Z. provides feedback and guides the research. The manuscript is drafted by J.P.M. and L.X. and is reviewed by all the authors.

## Competing interests

The authors declare no competing interests.
