## [Transparent Peer Review file · Nature Communications]

3T-VASP: Fast Ab-Initio Electrochemical Reactor via Multi-Scale Gradient Energy Minimization

Corresponding Author: Dr Jonathan Mailoa

Version 0:

Reviewer comments:

Reviewer #1

(Remarks to the Author)

Mailoa and coworkers present a new method for exploring chemical reaction intermediates based on the tiered tensor transformation (3T) method previously reported by the same authors. In locating local intermediates on the potential energy surface, the 3T method traverses potential energy barriers that conventional structure minimization cannot, and molecular dynamics simulations may be ineffective for. The authors demonstrate the utility of the 3T method on two examples: **(i)** binding of an organic cation molecule onto a perovskite surface vacancy defect site, which standard geometry optimization techniques are unsuccessful for, and **(ii)** exploration of reaction intermediates in a complex mixture of small molecules typically found in Li-ion battery electrolyte solutions, obtaining various intermediates, of which several have previously been observed experimentally. The method will likely be useful in studies that aim to identify low-energy intermediates generated by electrochemical reactions (and possibly other reactivity).

The manuscript is generally well written and easy to understand, supported by results presented in a clear manner. However, the manuscript also contains some ambiguities, and lacks some detail and discussion of the state of the art in reaction discovery. In my comments below, I outline the points for which I consider revision necessary. Upon satisfactorily addressing these comments, publication of the manuscript in *Nature Communications* can be justified.

Comment 1:

Page 2, Lines 14–15: The statement that “many are experimentally verified” should be modified to emphasize that the intermediates have been observed in previous experimental studies. In its current phrasing, the statement is easily misconceived as that they are experimentally verified in the present study.

Comment 2:

Page 3, Lines 13–20: The authors mention various methods for discovering new reactions, but limit their discussion to dynamics-based methods. There are several methods that use information about the potential energy surface (PES) rather than ab initio molecular dynamics (AIMD) simulations to explore reactivity. These methods should be mentioned and discussed. They are particularly relevant as the present work does not rely on AIMD-based methods, but rather static density-functional theory (DFT) for exploration of the PES. What makes 3T different/better than these methods? Here are some example references that should be cited (and discussed): *WIREs Comput Mol Sci.* **2021**, *11*, e1538; *J. Chem. Theory Comput.* **2022**, *18*, 5393; *J. Comput. Chem.* **2021**, *42*, 2036; *J. Comput. Chem.* **2015**, *36*, 601.

Comment 3:

Page 7, Line 11: An ionic bond is also a chemical bond. Perhaps the authors mean “covalent” instead of “chemical”? Or perhaps “non-covalent interaction” instead of “ionic bonding”? Also, to me the cited paper (ref. 26: *J. Mater. Chem. A* **2020**, *8*, 8313) seems to suggest that the passivation mechanism involves a chemical bond between the pyridine N and a Pb ion, and not that it is ionic? Or are the authors referring to the effects of the chloride? Some clarification/further details would be appreciated.

Comment 4:

Page 7, Line 22: It is not surprising that conventional optimization only yields the cation “floating” above the defect site as there must be an energy barrier associated with the binding into the defect site. Standard relaxation techniques are incapable of overcoming such energetically uphill barriers, since they attempt to minimize the energy with respect to nuclear

coordinates in each step. The authors should comment on why standard optimization techniques fail to bind the cation into the defect site. The authors should also characterize the pathway associated with the binding of the organic cation in the defect site — i.e., calculate the minimum energy path (MEP) including local minimum and maximum structures (intermediates and transition states, TSs) for a more complete discussion on this.

Comment 5:

Page 7, Line 25: Does “micro groups based on its rotatable bonds” refer to dihedral angles, or internal coordinates in general? If so, does one micro group correspond to one row of the Z-matrix representation of the internal coordinates? Is the “macro group” just the entire molecule expressed in internal coordinates (entire Z-matrix), i.e., invariant to rotation and translation? Some clarification would be helpful.

Comment 6:

Page 8, Line 14: Are these random conformations of TE4PBA or random placements and orientations (i.e., translation and rotation)? How is the randomization performed?

Comment 7:

Page 8, Line 15: Is there a rationale for using a classical force field (FF) apart from the low computational cost? Does the initial classical FF exploration limit the applicability of the method, as classical FFs may yield qualitatively incorrect geometries for certain systems (e.g., transition metal-containing systems)? How are the numbers of cycles and steps for both the 3T-FF and 3T-VASP procedures determined?

Comment 8:

Page 8, Line 17: How do the structures look after the 3T-FF relaxation (but before 3T-VASP relaxation)? Please include these in the supporting information.

Comment 9:

Page 8, Line 19: I would like to see a plot included in the supporting information for the potential energy vs bond-forming atom-pair distance (or cation–defect site distance), showing the approximate pathway/barrier for binding, which the standard relaxation methods cannot overcome.

Comment 10:

It would be interesting to see characterization of all stationary points along the pathway leading to incorporation of the organic cation into the defect site (this relates to Comment 4). While I understand that this is not the main goal of this manuscript, I believe it may highlight the usefulness of the 3T-VASP method due to the significant efforts that go into reaction mechanism elucidation. For example, if the method is capable of yielding an approximate MEP, the local energy minima and maxima can be optimized to intermediates (including reactant(s) and product(s)) and TSs, respectively. These can then be connected by a MEP resulting from calculation of the intrinsic reaction coordinate (IRC) starting from the characterized TSs. If direct TS optimization from the maximum energy points along the path fails, the authors may attempt to locate TSs by double-ended optimization techniques starting from intermediate pairs using, for example nudged elastic band (NEB). I consider this comment particularly relevant because of the authors' last sentence in the introduction: “The reaction byproduct trajectories generated by 3T can be used for further individual reaction energy barrier study such as those commonly done using nudged elastic band making 3T a suitable complement to existing state-of-the-art methods.”

Comment 11:

Figure 2: The abbreviations “TCO”, “ETL” and “HTL” are not spelled out in full. Also, I believe there is a typographical error in subfigure b, possibly a missing “for” (“[...]enabled for TE4PBA[...]").

Comment 12:

Page 10, Lines 17–20: The analogy of an ab initio reactor warrants a citation to Martinez and coworkers' ab initio nanoreactor although the methods differ in that Martinez and coworkers used AIMD whereas the present work uses static DFT: *Nature Chem.* **2014**, *6*, 1044.

Comment 13:

Page 14, Line 20: What does “difficult to be resolved experimentally” mean? Does this mean that the hydrocarbon chain products have not been identified experimentally? Clarification would be appreciated.

Comment 14:

Page 17, Lines 10–12: I would like the authors to comment on what kinds of reactivity the authors imagine that 3T-VASP could be used to study in the future. Potential limitations?

Reviewer #2

(Remarks to the Author)

In this manuscript, the authors propose a multiscale approach to identify products of chemical reactions. They focus on two systems: a complex floppy organic molecule passivator binding onto perovskite solar cell 20 surface, and lithium-ion battery liquid electrolyte. The multiscale approach (named 3T) is described in a manuscript available online, but not peer-reviewed. The approach is quite complicated to apply in practice from what I understand, but I simply consider it as an efficient way of generating low energy atomic configurations, enabling exploration of a wide energy landscape by avoiding local minimas. I

find the research work not well presented, but I also see some possible interest for the community.

I find several issues with the way the work is presented in this manuscript. First of all, I find the comparison/reference to MD somewhat misleading in the introduction. The methodology should instead be compared to other approaches that attempt to identify reaction products, such as ["Elementary Decomposition Mechanisms of Lithium Hexafluorophosphate in Battery Electrolytes and Interphases" by Evan Walter Clark Spotte-Smith, Thea Bee Petrocelli, Hetal D. Patel, Samuel M. Blau, and Kristin A. Persson, ACS Energy Letters 2023 8 (1), 347-355, DOI: 10.1021/acseenergylett.2c02351] for example, in the field of lithium-ion battery liquid electrolyte.

I also find that, given that the 3T methodology has been described elsewhere, the authors should focus on what is specific to the physical systems presented here, and how the reactions products are actually coming out of about 50 DFT calculations.

This is in my opinion the most important and interesting aspect of the approach, and the level of details in the current manuscript is insufficient in my opinion. Do they carry out 50 relaxation steps at the DFT level starting with the configurations obtained from 3T? Or something else? How do they know the relative importance of each reactant observed? What are the groups used in 3T for the electrolyte?

I feel in general that the paper is lacking focus on some the important aspects of interest to the community, while diverging on less important aspects such as implementation details. The manuscript also relies too much on information found in the supplementary materials, precisely because the authors are trying to cover too much materials. They should focus the main text on explaining the main ideas and rely on supplementary materials for implementation details.

Reviewer #3

(Remarks to the Author)

The authors have recently developed a new method that is able to efficiently find minima on the PES of molecular systems based on a representation in non-Cartesian, multi-scale coordinates. In the present work, they extend and apply it to reactive systems to identify geometries of floppy molecules binding to a surface and to reaction products of battery electrolytes in reductive and oxidative environments. The method can identify possible products, but apparently does not analyse the pathways and barriers to them, therefore it cannot assess if they are thermally accessible at the moment. Such a method is still very useful, but also not new. Many such methods exist, and a major weakness of the present work is that literature on this topic is not mentioned at all. See e.g. some recent reviews (10.1002/wcms.1354, 10.1021/acs.jpca.8b10007, 10.1007/s11244-021-01543-9) and more recent developments.

In my opinion, a brief description of the state of the art and an assessment of improvements of the newly proposed method is essential for scientific work. Furthermore, a substantial improvement should be shown for publication in a high impact journal.

The authors propose a 2-level approach (1st FF, then DFT), which seems to be important. How does this compare to crest + DFT (10.1021/acs.jctc.9b00143, 10.1039/C9CP06869D), a method I consider nowadays state of the art, not just simple energy minimisation, which is used for comparison by the authors. A comparison of results and computational demands of both methods would strengthen the work considerably.

What is "reduction ionization"?

Line 180: "These electrolyte systems (mass density = 1.24 – 1.46 g/cm³) are intentionally not charge-compensated". How are calculations for periodic non-charge neutral boxes possible? Their energy should be infinity? (a page later the text suggests that the boxes are made charge neutral by adding or removing electrons. Please make sure to avoid ambiguities in your description. (line 203, "In this VASP DFT simulation with neutral charge".))

In the main text, there is no information how the "simple ab initio energy minimization" can overcome reaction barriers. Some description should be moved from the SI into the main text to make this important point clearer.

There is no analysis of barriers. Therefore, it is unclear if products are chemically accessible. It seems they are actually potential products. There are more efficient algorithms. In reaction mechanism exploration, such candidates are usually checked for accessibility, see reviews mentioned above and comment on this limitation. The addition and removal of electron seems barrierless. Are there actually barriers AFTER adding the electrons?

Has the new representation a higher or a lower dimensionality than one using internal coordinates? (In other words: are the large-scale fragments replacing the micro objects or are these additional coordinates?)

In summary, I cannot recommend publication of the work at the present stage. With a major revision, the work will probably become publishable, but it will depend on how much improvement (if any) can be demonstrated over existing methods whether it will be suitable for Nature Communication.

Reviewer #4

(Remarks to the Author)

In this manuscript, Mailoa et al. demonstrate that energy minimization within a basis of "structure transformation modes" accesses chemically interesting low-energy states for molecular adsorption and electrochemical reactivity within a modest number of DFT gradient evaluations. The observed electrochemical reduction reaction products seem fairly reasonable, and

the potential utility of leveraging the identified species in order to aid in the construction of a chemical reaction network is self evident.

However, I do have to take issue with some of the specifics on the oxidation side. In particular, the inclusion of oEC- as a reactant on the oxidation side makes no sense to me. The authors themselves described oEC- as "a highly reactive radical species", which is precisely why it is unphysical to expect that oEC- would diffuse from the negative electrode, where it is created, all the way to the positive electrode, where it could participate in oxidation reactions, without reacting with anything else along the way. In contrast, it is reasonable that Li₂CO₃ exists at the positive electrode. Further, the majority of the oxidation side reactivity that is reported is facilitated by the unphysical presence of oEC- near the positive electrode. While I recognize that this choice does not take away from the demonstrated utility of the method itself, I do find it very distracting and feel that it undermines the oxidation side example.

More generally, while the benefits of the approach in comparison to "per-atom" Cartesian minimization or Cartesian AIMD are emphasized, the authors never mention or compare to the conceptually similar and decades old idea of internal coordinates. Indeed, efficient molecular geometry optimization as implemented in every single molecular DFT package critically depends on transforming from Cartesian coordinates into a basis of "internal" coordinates representing distances between pairs of atoms, three atom angles, and four atom dihedral angles. (More recently, translation and rotation internal coordinates have also been introduced which explicitly describe the collective translations and rotations of entire molecules, or parts of molecules.[1]) Changing any single internal coordinate effectively modifies a large collection of Cartesian coordinates, allowing for optimizers to reach lower energy minima with far fewer gradient calls than is possible when optimizing in pure Cartesian coordinates.[2]

With that context, the question must be asked - have Mailoa et al. simply "rediscovered" internal coordinates? There could still be impact/value here - internal coordinates are not typically used for periodic systems, and the need to back-transform from internal to Cartesian coordinates at each optimization step also introduces numerical challenges when angle or dihedral coordinates become invalid (though the addition of dummy atoms resolves that [3]) while the 3T approach's clever use of AutoGrad seems to avoid the need to back-transform at all. However, modern molecular optimization algorithms are also quasi-second-order, leveraging a guess Hessian (which is reasonable to initially approximate as diagonal in internal coordinates) and then updating that Hessian based on gradient information at each step, while the 3T procedure is purely first order, which seems like a drawback. Further, the current state-of-the-art optimizer Sella (while it is marketed as a transition state optimizer, it is just as effective at optimizing to minima) leverages geodesic stepping to more efficiently move towards a lower energy at each iteration[3].

All of this to say that my ability to evaluate the novelty of the approach described in this manuscript requires the authors to place their work in the context of internal coordinate optimization approaches and more generally compare to modern molecular optimizers like Sella. Indeed, in Sella the user can define whatever custom set of internal coordinates that they would like (or to allow Sella to do so automatically) - how different is that from the 3T procedure where a structure is "automatically segmented hierarchically into different micro-groups (based on its rotatable bonds) and a macro-group (the entire molecule)"? To me, they seem very similar, and how much additional utility the 3T approach provides over a similarly careful use of Sella remains unclear.

I enjoyed reading the paper and hope to better understand the novelty and value-add of the 3T approach after a revision by the authors.

Sincerely,
Samuel Blau

- 1: <https://doi.org/10.1063/1.4952956>
- 2: <https://doi.org/10.1021/acs.jctc.3c00188>
- 3: <https://doi.org/10.1021/acs.jctc.2c00395>

Version 1:

Reviewer comments:

Reviewer #1

(Remarks to the Author)

I would like to thank the authors for their thorough response to my comments on the previous version of the manuscript.

Regarding my previous comment 2, I feel that it has not been sufficiently addressed. While the discussion has been somewhat extended by mentioning a few automated reaction mechanism discovery methods, I believe that these methods should also be represented in Figure 1, where standard relaxation, molecular dynamics and 3T relaxation are compared.

The authors claim that the growing string method requires both reactant and product to be known in advance, but the single-ended growing string method (which is cited) only requires a single intermediate to be known (for the double-ended growing string method, both reactant and product are needed).

I would not say that AutoMeKin's BXDE ("a combination of temperature-dependent metadynamics and transition state

analysis") is "routinely" performed, since it is to the best of my knowledge not very commonly used.

Technically, I think that the description "temperature-dependent metadynamics" is inaccurate. Both metadynamics and boxed dynamics use a bias, but these are different — in MTD through collective variables and in BXDE through the potential energy.

Regarding my previous comment 5, the authors added to the Methods section an explanation that for example a molecule with seven atom indices can be segmented into three micro groups $m = [0,2,6], [1,3,4], [5]$. However, it is not clear to me how exactly such a molecule is segmented. Could the authors perhaps include a graphic of the molecule including the atom index numbering to visualize this?

Reviewer #2

(Remarks to the Author)

In my first review, my general comment was that the paper is lacking focus on some the important aspects of interest to the community, while diverging on less important aspects such as implementation details, relies too much on information found in the supplementary materials, precisely because the authors are trying to cover too much materials. While the authors have made some efforts to improve the manuscript, I am still feeling the same about this manuscript in general. And I do not think this manuscript is well suited for Nature Communications.

For instance, to my first comment about how complicated it seems to apply methodology, the authors respond that the software is easy to use. A scientific manuscript however, in my opinion, should not consider the software as a black box and let the reader guess what happens inside the box. A scientific report should describe the methodology and ideas so that a reader could reproduce the research and write the same software on their own. The authors still fail at that task in my opinion in this revised manuscript.

I see this manuscript as describing the application of 3T algorithm previously proposed by the same authors to two applications, with description of results. But no added understanding to what the algorithm is doing for these applications. For instance, there is no details about the choice of the theta variables for these applications. In the 3T original paper, there was a mention of energy "kicks" and their values. I don't see nay mention of that in the present manuscript.

Another point that I missed in my first review: the authors refer a lot to PyTorch functionalities. PyTorch should be an implementation "detail", not used in general presentation. Mathematical functions should be used in manuscript.

As I reread the manuscript and the responses to the review comments, I also realized the following: to minimize DFT calls, FF can be run for a while, before DFT calls are used to "refine" solution. That would be the way to appropriately evaluate the effect of the 3T multiscale approach (in figure 2 for instance).

Reviewer #3

(Remarks to the Author)

The manuscript has been improved considerably by including a better relation to existing methods, demonstrating the improvements of the new method, and providing a clearer presentation. Therefore, I think it is publishable now. The manuscript is, at the same time quite technical. This is probably necessary due to the complexity of the topic. While I expect that the methods will be used by others and the work will receive attention, I am not sure if a more technical journal may be better for this work and leave this decision to the editor.

Reviewer #4

(Remarks to the Author)

I greatly appreciate all of the work that the authors did in response to my review. Indeed, they have gone above and beyond what I would have expected in their construction of Fig S16 and their implementation of Simple3T. I find both very interesting/useful - thank you for the time that was put into this! Hopefully the authors found the extra work to be instructive and worthwhile.

However, the authors have drawn key incorrect conclusions regarding the applicability of Sella, and thus they have not done the most important thing I would like to see, which is a direct comparison of 3T-VASP with Sella (employing the VASP ASE calculator) on the electrolyte example. Admittedly, there is currently an issue with the translation-rotation internal coordinates in Sella (see my open issue here: <https://github.com/zadorlab/sella/issues/32>), but even still, I think a direct comparison with Sella as it currently functions is crucial to establishing the utility of the work. Assuming that 3T yields the reaction products reported in the manuscript in substantially fewer DFT calls than the state-of-the-art internal coordinate optimizer, then I fully support publication of this manuscript in Nature Communications. If instead Sella is just as efficient and effective as 3T or requires even fewer DFT calls to yield the observed reaction products, then I question the value of the approach, despite its ingenuity.

Let me clarify some of the misconceptions, and then I will be very clear about what I am asking for and how I would suggest carrying it out. Referring to the author's numbered points on "how our 3T approach differs from the internal coordinate methods TRIC and Sella":

1. I can't speak for the GeomeTRIC code, but Sella works with periodic boundary conditions. The internal coordinates automatically wrap across the periodic boundaries via the minimum image convention. Further, there is no inherent limit to the number of atoms in a

system being optimized with Sella. 325 atoms in a periodic box using VASP should be no problem at all.

2. Minor point of clarification here - people do often employ $>3N$ "redundant" internal coordinates. I'm not sure if Sella always sticks to the usual $3N-6$ internal coordinates, but just for the authors' information - this comes back to a decades old debate about whether the absolute fewest number of coordinates is best for optimization, or if having more coordinates which effectively allow you to "cheat" through a higher dimensional space is better. David Baker was the historical proponent of $3N-6$ aka "delocalized" internal coordinates while Bernard Schlegel was the historical proponent of $>3N-6$ aka "redundant" internal coordinates. This information does not impact the manuscript - I just wanted to provide historical context.

3. I again cannot speak for GeomeTRIC, but while Sella contains the functionality to impose all sorts of explicit constraints, as chemists often like to do when performing bespoke transition state optimization, no constraints are automatically defined, and constraints are in no way central to Sella's default function.

4. I agree that the multi-level aspect of 3T is novel.

5. I agree. However, it remains unclear if the ML-centric advantages of Adam are actually more valuable than the quasi-second-order machinery of Sella (which molecular geometry optimizers have employed in some form or another for decades) in the context of molecular optimization.

6. The way this is worded makes me worried that there is confusion between "constraints" and "coordinates". Sella can add dummy atoms on-the-fly in order to add new internal coordinates that effectively replace a previous coordinate that has gone invalid - i.e. a three-atom angle going to 180 degrees, or a dihedral angle going to 0 or 180 degrees. (Side note - the fact that 3T effectively employs a flavor of "internal coordinates" which never go invalid is a point in its favor that could probably be mentioned somewhere in the manuscript.) Internal coordinates going invalid is an issue that plagues other molecular geometry optimizers, but is alleviated in Sella. But any actual constraints - e.g. fixing a specific bond length or angle - defined by the user when employing Sella do not change over the course of the optimization. Back to coordinates - while Sella can dynamically add dummy atoms and modify the internal coordinates to alleviate issues with coordinate validity, it does not seek to regularly re-define the internal coordinates during an optimization trajectory. Thus, I believe that re-starting an optimization from a previously optimized structure, as done with 3T, will re-define the internal coordinates and perhaps open up new avenues for further optimization.

With those misconceptions clarified, let me return to my ask. I am sorry to ask for even more work from the authors when they have already done so much, but as I stated previously, I think this is critical: please run Sella atop the VASP ASE calculator on one of your electrolyte boxes and compare to 3T! I do not think you should need to do anything fancy - just use the standard ASE VASP calculator. For Sella, be sure to set `order=0` (i.e. you're trying to optimize to a minimum, not a saddle point) and `internal=True` (i.e. you want to use internal coordinates). Please run a sequence of five iterations of 50 steps each starting from the geometry obtained from 3T-FF. As I stated above, stopping Sella after 50 iterations and restarting it fresh will re-define the internal coordinates and perhaps be beneficial in the same way that it is for 3T. The computational cost of this experiment should be equivalent to the 3T-VASP experiment, and thus I hope it will not be burdensome. I badly want to know the outcome of this experiment - while I do not expect you to run into any serious issues, if you do, I would be happy to help, so feel free to email me at smblau@lbl.gov.

Sincerely,
Sam Blau

Version 2:

Reviewer comments:

Reviewer #1

(Remarks to the Author)

I would like to thank the authors for their efforts to address my concerns — these have now been addressed sufficiently. Overall, I feel that the manuscript has been improved significantly over these rounds of revisions by addressing the comments of all reviewers. It is now also easier to evaluate the impact of 3T-VASP, as the manuscript contains a more appropriate comparison with existing methods. I believe that 3T-VASP is an interesting and important development, with implications on prediction of reaction outcomes in complex environments.

Just one small comment: in the "Discussion on 3T Comparison with Sella for Bulk Electrolyte Electrochemical Reactions" section of the revised supporting information, the authors say "We instruct Sella to work on optimization (`order=0`)". Even if `order=1`, it performs an optimization, but to a first order saddle point instead of a minimum. Instead of saying just "optimization", the authors should specify that it is an optimization to a minimum.

Reviewer #2

(Remarks to the Author)

I appreciate the work the authors have done to improve their manuscript. But in my opinion, this is still not a good fit for that journal. I don't see a big impact in general. In my opinion, such a paper should be a longer paper in a more technical and

specific journal, not a communication with so much supplementary materials.

Reviewer #4

(Remarks to the Author)

I thank the authors for their additional work in response to my comments. I am now satisfied and support the publication of the manuscript in Nature Communications.

Reviewer #1

Mailoa and coworkers present a new method for exploring chemical reaction intermediates based on the tiered tensor transformation (3T) method previously reported by the same authors. In locating local intermediates on the potential energy surface, the 3T method traverses potential energy barriers that conventional structure minimization cannot, and molecular dynamics simulations may be ineffective for. The authors demonstrate the utility of the 3T method on two examples: (i) binding of an organic cation molecule onto a perovskite surface vacancy defect site, which standard geometry optimization techniques are unsuccessful for, and (ii) exploration of reaction intermediates in a complex mixture of small molecules typically found in Li-ion battery electrolyte solutions, obtaining various intermediates, of which several have previously been observed experimentally. The method will likely be useful in studies that aim to identify low-energy intermediates generated by electrochemical reactions (and possibly other reactivity).

The manuscript is generally well written and easy to understand, supported by results presented in a clear manner. However, the manuscript also contains some ambiguities, and lacks some detail and discussion of the state of the art in reaction discovery. In my comments below, I outline the points for which I consider revision necessary. Upon satisfactorily addressing these comments, publication of the manuscript in Nature Communications can be justified.

We thank the reviewer for this positive comment.

Comment 1:

Page 2, Lines 14–15: The statement that “many are experimentally verified” should be modified to emphasize that the intermediates have been observed in previous experimental studies. In its current phrasing, the statement is easily misconceived as that they are experimentally verified in the present study.

We thank the reviewer for this comment. The abstract has been modified to remove this ambiguity.

- Abstract **page 2, line 13**. We modified the sentence:

We then demonstrate more complex examples by generating hundreds of electrochemical reaction byproducts in lithium-ion battery liquid electrolyte (many are verified in previous experimental studies), with most trajectories completed within 50–100 DFT steps as opposed to more than 10,000 steps typically utilized in an ab-initio molecular dynamics trajectory.

Comment 2:

Page 3, Lines 13–20: The authors mention various methods for discovering new reactions, but limit their discussion to dynamics-based methods. There are several methods that use information about the potential energy surface (PES) rather than *ab initio* molecular dynamics (AIMD) simulations to explore reactivity. These methods should be mentioned and discussed. They are particularly relevant as the present work does not rely on AIMD-based methods, but rather static density-functional theory (DFT) for exploration of the PES. What makes 3T different/better than these methods? Here are some example references that should be cited (and discussed): *WIREs Comput Mol Sci.* 2021, 11, e1538; *J. Chem. Theory Comput.* 2022, 18, 5393; *J. Comput. Chem.* 2021, 42, 2036; *J. Comput. Chem.* 2015, 36, 601.

We thank the reviewer for this suggestion. The literature review section in the introduction has now been expanded to include these methods and references.

□ Main text **page 3, line 20**. We added the sentences:

Other promising reaction exploration approaches are temperature-independent, and they work by modifying steady-state structure optimization method to escape local energy minimum. For example, the AFIR approach works by adding multi-parameter empirical artificial forces between all atom pairs.^[1] The approach is further extended in the implementation of Chemoton 2.0 software, where the combination of brute force conformer sampling and empirical artificial force added on chosen reactive sites between candidate reactant molecule pairs are used to accelerate a reaction.^[2] When the reactants and products are already known in advance, reaction transition states can be found using different methods, for example by using the growing string method which sequentially places intermediate nodes in the delocalized internal coordinates (bonds, angles, or torsions) relevant for reaching the desired reactive outcome.^[3] A combination of temperature-dependent metadynamics approach for generating reaction and subsequent transition state analysis is also routinely performed.^[4-5] We also note that our work has some similarities to internal coordinate methods recently used in non-periodic quantum chemistry software for structure minimization,^[6-8] albeit with some extensive differences (see **Supplementary Information**). More in-depth reviews of these reaction generation methods are available in the literature.^[9-10]

- [1] Maeda, S. & Harabuchi, Y. Exploring paths of chemical transformations in molecular and periodic systems: An approach utilizing force. *Wiley Interdiscip. Rev. Comput. Mol. Sci.* **11**, e1538 (2021).
- [2] Unsleber, J. P., Grimmel, S. A. & Reiher, M. Chemoton 2.0: Autonomous exploration of chemical reaction networks. *J. Chem. Theory Comput.* **18**, 5393–5409 (2022).
- [3] Zimmerman, P. M. Single-ended transition state finding with the growing string method. *J. Comput. Chem.* **36**, 601–611 (2015).
- [4] Marónez-Núñez, E. *et al.* AutoMeKin2021: An open-source program for automated reaction discovery. *J. Comput. Chem.* **42**, 2036–2048 (2021).
- [5] Jara-Toro, R. A., Pino, G. A., Glowacki, D. R., Shannon, R. J. & Marónez-Núñez, E. Enhancing automated reaction discovery

with boxed molecular dynamics in energy space. *ChemSystemsChem* 1, e1900024 (2019).

- [0] Wang, L. P. & Song, C. Geometry optimization made simple with translation and rotation coordinates. *J. Chem. Phys.* 144, (2016).
- [1] Shajan, A., Manathunga, M., Götz, A. W. & Merz, K. M. Geometry optimization: A comparison of different open-source geometry optimizers. *J. Chem. Theory Comput.* 19, 7533–7541 (2023).
- [2] Hermes, E. D., Sargsyan, K., Najm, H. N. & Zádor, J. Sella, an open-source automation-friendly molecular saddle point optimizer. *J. Chem. Theory Comput.* 18, 6974–6988 (2022).
- [3] Dewyer, A. L., Argüelles, A. J. & Zimmerman, P. M. Methods for exploring reaction space in molecular systems. *Wiley Interdiscip. Rev. Comput. Mol. Sci.* 8, e1534 (2018).
- [4] Simm, G. N., Vaucher, A. C. & Reiher, M. Exploration of reaction pathways and chemical transformation networks. *J. Phys. Chem. A* 123, 385–399 (2019).
- [5] Steiner, M. & Reiher, M. *Autonomous reaction network exploration in homogeneous and heterogeneous catalysis. Topics in Catalysis* vol. 65 (Springer US, 2022).

Comment 3:

*Page 7, Line 11: An ionic bond is also a chemical bond. Perhaps the authors mean "covalent" instead of "chemical"? Or perhaps "non-covalent interaction" instead of "ionic bonding"? Also, to me the cited paper (ref. 26: *J. Mater. Chem. A* 2020, 8, 8313) seems to suggest that the passivation mechanism involves a chemical bond between the pyridine N and a Pb ion, and not that it is ionic? Or are the authors referring to the effects of the chloride? Some clarification/further details would be appreciated.*

We thank the reviewer for this suggestion on naming conventions. As for the cited reference, the authors' comments on the interaction between pyridine N and Pb ion refers to two different papers below [1-2], which utilize either PTQ10 polymer or 3-hydroxypyridine for passivation. Both compounds are neutral compounds which have N atoms in the carbon ring with an exposed electron pair, making it easy for the nitrogen atom to form strong interaction with Pb atoms. The cited papers show the reduction of metallic lead (XPS figures) after the introduction of the passivator, leading the cited author [2] to conclude that the N–Pb bond is responsible for this charge transfer. However, the passivator agent used by Wu *et al.* is an EPC salt, where the Cl⁻ anion and the organic cation passivate different perovskite defect sites [3]. Wu *et al.* then associated the reduction in Pb atoms observed in their XPS data to be caused by the same mechanism proposed by Fu *et al.* [2]. The cation used by the author does have a nitrogen atom in the carbon ring, but it is an N⁺ atom without any exposed electron pair. This N atom being surrounded by carbon atoms (and it being positively charged) makes it difficult for the N atom to directly interact with the Pb²⁺ atom, so we are not sure if Wu *et al.*'s direct analogy on N–Pb charge transfer is applicable for EPC salt. Nevertheless, the DFT snapshot by Wu *et al.* (Figure 5 in [3]) confirms that the organic EPC cation N and perovskite Pb atoms are far apart. In this case, Cl⁻ interacts with the Pb²⁺, while the organic cation forms no covalent bond with the perovskite surface. In our manuscript, we are interested in organic cation passivating the A-site vacancy defect of perovskite, and we were referring to this non-covalent interaction between the cation and the perovskite, as demonstrated by Wu *et al.*'s DFT work. We have now made our reference to [3] more specific.

[1] L. Meng, *et al. J. Am. Chem. Soc.* **140**, 17255 (2018). DOI: 10.1021/jacs.8b10520.

[2] R. Fu, *et al. Chem. Commun.* **53**, 1829 (2017). DOI: 10.1039/C6CC09492A.

[3] X. Wu, *et al. J. Mater. Chem. A* **8**, 8313 (2020). DOI: 10.1039/d0ta02222e.

Main text **page 7, line 10**. We modified the sentence:

This organic cation passivation mechanism does not necessarily involve covalent bonding with the perovskite surface and may be simple non-covalent interaction as demonstrated by DFT, but the same 3T method can be used for electrochemical reaction product generation in the next section.

Comment 4:

Page 7, Line 22: It is not surprising that conventional optimization only yields the cation "floating" above the defect site as there must be an energy barrier associated with the binding into the defect site. Standard relaxation techniques are incapable of overcoming such energetically uphill barriers, since they attempt to minimize the energy with respect to nuclear coordinates in each step. The authors should comment on why standard optimization techniques fail to bind the cation into the defect site. The authors should also characterize the pathway associated with the binding of the organic cation in the defect site — i.e., calculate the minimum energy path (MEP) including local minimum and maximum structures (intermediates and transition states, TSs) for a more complete discussion on this.

We thank the reviewer for this suggestion and the corresponding explanation has been added. As for MEP and TS, please refer to our response to **point #10**.

Main text **page 7, line 24**. We added the sentence:

In the standard relaxation technique, each relaxation step with respect to the individual nuclear coordinates attempts to minimize the total system energy, making it difficult to perform a coordinated structure transformation which can overcome uphill energy barriers such as those presented by the process of embedding a passivating molecule into the perovskite surface defect site.

Comment 5:

Page 7, Line 25: Does "micro groups based on its rotatable bonds" refer to dihedral angles, or internal coordinates in general? If so, does one micro group correspond to one row of the Z-matrix representation of the internal coordinates? Is the "macro group" just the entire molecule expressed in internal coordinates (entire Z-matrix), i.e., invariant to rotation and translation? Some clarification would be helpful.

We thank the reviewer for this clarification request. For micro-group assignment, the phrase refers to separating a molecule graph into its sub-graphs based on the existence of rotatable bonds, as we have explained in the **Methods** section. There is no dihedral angle / internal coordinate consideration here,

because we simply separate a molecule's atoms into separate smaller groups of atom indices. For example, if an entire molecule consists of atom [0,1,2,3,4,5,6], its micro-group can potentially be [0,2,6], [1,3,4], and [5] depending on where the rotatable bonds are. A more visual explanation is available in **Figure 2b** and **Supplementary Figure 2**. Physically, the macro-group of this molecule is simply [0,1,2,3,4,5,6]. For computational convenience reasons, inside our code the macro-group is encoded as a nested list of these micro-groups instead ([[0,2,6], [1,3,4], [5]]). Micro-groups and macro-groups are just list of atom indices to be used in subsequent PyTorch tensor-based geometry transformations.

- Main text **page 8, line 4**. We modified the sentence:
In the 3T approach, the TE4PBA structure is automatically segmented hierarchically (see **Methods, Figure 2b**) into different micro-groups (based on its rotatable bonds) and a macro-group (the entire molecule).
- **Methods** section **page 25, line 10**. We added the sentence:
For example, an entire molecule with 7 atom indices [0,1,2,3,4,5,6] may be segmented into micro-group $m = [0,2,6]$, [1,3,4], and [5] and into macro-group $M = [0,1,2,3,4,5,6]$. For computational convenience reasons, we encode M as a nested list of m instead ([[0,2,6],[1,3,4],[5]]).

Comment 6:

Page 8, Line 14: Are these random conformations of TE4PBA or random placements and orientations (i.e., translation and rotation)? How is the randomization performed?

We thank the reviewer for this clarification request. We randomize the rotation and translation anywhere within a cubic space of $10 \times 10 \times 10 \text{ \AA}^3$ centered directly above the defect site using packmol, with the bottom of the cube being 1 \AA above the highest perovskite atom. There is no conformation randomization being done during the initial structure placement, although it will happen very quickly after 3T-FF (the 3T- classical force field part) is activated.

- **Supplementary information page 4, line 4**. We added the sentence:
The initial structure for 3T-FF was constructed by placing the cation with random rotation and translation within a cubic space of $10 \times 10 \times 10 \text{ \AA}^3$ centered directly above the defect site using packmol, with the bottom of the cube being 1 \AA above the highest perovskite atom.

Comment 7:

Page 8, Line 15: Is there a rationale for using a classical force field (FF) apart from the low computational cost? Does the initial classical FF exploration limit the applicability of the method, as classical FFs may yield qualitatively incorrect geometries for certain systems (e.g., transition metal-containing systems)? How are the numbers of cycles and steps for both the 3T-FF and 3T-VASP procedures determined?

We thank the reviewer for this comment. Our primary rationale is indeed the low computational cost of FF. We intend 3T-FF simply to be a generator for approximately acceptable initial configurations for 3T-VASP, as we believe that FF geometries are 'wrong' and higher accuracy methods such as VASP (or more accurate ones) should be used. FF is sufficient for the two examples in this manuscript, but we agree with the reviewer that qualitatively incorrect geometries may arise for transition-metal containing systems, and this warning will be included in the manuscript.

We wanted our workflow to be relatively parameter-free (except for the number and type of molecules that the user want to place in the reaction periodic boundary condition box). When we first developed 3T-FF, we noticed that the structure energy will decrease very rapidly due to the efficient multi-scale gradient optimization but would then fall into a relatively higher local energy minimum (liquid molecules not fully dispersed throughout the PBC box, for example) after several hundred steps. However, transitioning to a new cycle immediately lets the system escape from the said local energy minimum (see **Methods** section on Inter-Cycle 3T Parameter Hyperspace Transition, as well as **Supplementary Figure 4** for the reason behind the observed gradient discontinuity). This led us to conclude that more cycles with a smaller number of steps are generally more efficient as long as we first let the energy spike which often appear in the beginning of a new 3T cycle settles down into a lower-level energy plateau. 3T-FF computation cost is extremely low, so we are able to repeat many configurations of cycles and steps and spare the computation resources to over-minimize. This leads us to choose 5 cycles x 200 steps for both the perovskite and the electrolyte systems.

The setting for the subsequent 3T-VASP is more complicated because of the high computational cost of DFT. Because we knew that the perovskite 3T-VASP was 'complete' within 50 steps, we set 50 steps per cycle for our electrochemical reactions. A few initial trials for the 3T-VASP electrochemical reduction (<10) showed us that reactions are typically completed within two cycles, but we did observe a few unfinished reactions at the end of the second cycle (intermediates which do not look very chemically stable). To maximize our chances of obtaining finished reaction products, we chose to over-minimize the 3T-VASP energy by running 5 cycles x 50 steps for the 63 3T-VASP trajectories generated during electrochemical reduction. We subsequently learned that we do not generate much meaningful additional reactions beyond the first 3 cycles, so 3 cycles x 50 steps were chosen for the additional 63 trajectories generated during electrochemical oxidation part of the manuscript. Unfortunately it is difficult to know in advance how many 3T-VASP cycles are suitable for a given new chemistry because it is system-dependent. Some reactions like oEC radical generation happens frequently and very quickly, while some other reactions are rare or need to undergo multiple intermediate steps in the 3T-VASP trajectory. We suggest that the user starts with 5 cycles x 50 steps of 3T-VASP steps to first see whether the reactions are generated and if they look like they are already chemically stable or not. For the 3T-FF, the user can simply 'over-minimize' by increasing the number of cycles.

Unfortunately we are unable to perform a comprehensive study of the most efficient number of cycles and steps for a given system because statistics over a large number of trajectories are needed to draw a conclusion for a given new system chemistry (this requires a large amount of GPU computation cost, which has to be purchased by our department). However, a future extension of this work can be to create an automated molecular graph analyzer which analyzes the molecular graph and the minimization energy

curve during the runtime, and then decide whether the software should continue the cycle / transition to a new cycle / end the trajectory. This will further improve 3T-VASP automation and computational efficiency but is beyond the scope of the current manuscript.

- Main text **page 8, line 25**. We added the sentence:

We note that the force field should be of sufficient quality to produce qualitatively reasonable geometries because 3T-VASP minimization should at least be started from physically meaningful $\vec{T}_{3T_FF,T}$. Systems containing transition metal may require methods with higher accuracy than FF.

- **Methods** section **page 27, line 20**. We added the sentence:

It is relatively inexpensive to over-minimize the structure during 3T-FF cycles, but we suggest that the user first perform several 3T-VASP cycles with 50 steps each to determine the appropriate cycle/step number settings for new chemistries before performing large-scale 3T-VASP trajectory generations.

Comment 8:

Page 8, Line 17: How do the structures look after the 3T-FF relaxation (but before 3T-VASP relaxation)? Please include these in the supporting information.

We thank the reviewer for this feedback. We have added the requested structures in **Supplementary Information**. We have also included the initial structure (before 3T-FF, this complements our response to **point #6**). For all 3 initial structures, 3T-FF brought the cation into the perovskite defect site and 3T-VASP further relaxes the perovskite-cation bound state into their relaxed configurations.

- **Supplementary Figure 1**. We modified the figure:

Supplementary Figure 1 | Final FAPbI₃-TEP4PBA structures for both standard DFT relaxation and 3T relaxation with different initial TEP4PBA positions. The three standard DFT relaxations based on conjugate gradient atomic forces fail to find TEP4PBA conformation entering the vacancy defect site, while the three 3T structures all embed TEP4PBA deep into the FA⁺ vacancy defect sites with larger DFT binding energy. Relatively different 3T TEP4PBA final poses indicate several deep local energy minimums. For 3T relaxation, we show the initial structure (prior to 3T-FF), the structure after 3T-FF relaxation, and the structure after 3T-VASP relaxation.

Comment 9:

Page 8, Line 19: I would like to see a plot included in the supporting information for the potential energy vs bond-forming atom-pair distance (or cation–defect site distance), showing the approximate pathway/barrier for binding, which the standard relaxation methods cannot overcome.

We thank the reviewer for this feedback. This analysis is best done in combination with the transition state and energy barrier analysis. Please refer to our response to **point #10** below.

Comment 10:

It would be interesting to see characterization of all stationary points along the pathway leading to incorporation of the organic cation into the defect site (this relates to Comment 4). While I understand that this is not the main goal of this manuscript, I believe it may highlight the usefulness of the 3T-VASP method due to the significant efforts that go into reaction mechanism elucidation. For example, if the method is capable of yielding an approximate MEP, the local energy minima and maxima can be optimized to intermediates (including reactant(s) and product(s)) and TSs, respectively. These can then be connected by a MEP resulting from calculation of the intrinsic reaction coordinate (IRC) starting from the characterized TSs. If direct TS optimization from the maximum energy points along the path fails, the authors may attempt to locate TSs by double-ended optimization techniques starting from intermediate pairs using, for example nudged elastic band (NEB). I consider this comment particularly relevant because of the authors' last sentence in the introduction: "The reaction byproduct trajectories generated by 3T can be used for further individual reaction energy barrier study such as those commonly done using nudged elastic band making 3T a suitable complement to existing state-of-the-art methods."

We thank the reviewer for this feedback. We have added a section in the **Supplementary Information** about transition state and energy barrier analysis, which is primarily done using nudged elastic band (NEB) method.

- Main text **page 9, line 9**. We added the sentence:

Additional transition state and energy barrier analysis using snapshots extracted from 3T energy minimization trajectory is available in the **Supplementary Information**.

- **Supplementary Information**. We added the section on additional perovskite surface passivation transition state and energy barrier analysis, as well as **Supplementary Figure 15**.

Perovskite Surface Passivation Transition State and Energy Barrier Analysis

We further perform transition state analysis and determine the energy barrier of our FAPbI₃ A-site vacancy defect passivation using TE4PBA cation, using the trajectory generated by our 3T minimization. The cation embedding is first observed during the 3T-FF cycle, so we extracted a snapshot from the end of 3T-FF (post-snapshot, with the cation already deeply embedded into the defect site), a snapshot in the 3T-FF trajectory where the cation has not been embedded into the defect site yet (pre-snapshot), and a third snapshot with the cation being positioned somewhere in the middle of the pre-snapshot and post-snapshot (mid-snapshot) which is generated by nudged elastic band (NEB). The pre-snapshot and post-snapshot are then relaxed using standard VASP relaxation, as we need these snapshots to be in their respective local energy minimums. We then perform NEB using VASP which includes VTST (Transition State Tools for VASP) tools on the three structures (relaxed pre-snapshot, mid-snapshot, and relaxed post-snapshot). An intermediate saddle-point structure is obtained, with an energy barrier of 0.932 eV higher than the relaxed pre-snapshot.

This intermediate snapshot is then used to perform a transition state analysis. Because the TE4PBA is a very large and floppy molecule, we observe 2 trivial sidechain rotations vibration modes (one -CH₃ group and one -CH₂ group rotation) in addition to the real transition state vibration mode representing the cation embedding into the defect site. Five individual hydrogen atoms which are located away from the core vibration region of the transition state vibration mode are restricted from moving in the x axis to eliminate the trivial sidechain rotations, and only the real transition state vibration mode representing the molecule embedding event remains. We then slightly perturb this cation structure according to the transition state vibration mode in both directions (toward the relaxed pre-snapshot and the relaxed post-snapshot directions) and relax the structure using standard VASP relaxation, producing two trajectories where the transition state TE4PBA cation moves back in the direction of either the relaxed pre-snapshot or to the relaxed post-snapshot. Note the structure and energy gap between the relaxed pre/post-snapshots and the structures relaxed from the NEB transition state. This indicates that there are additional transition states and energy barriers in the reaction trajectory landscape because the TE4PBA cation is a very large and floppy molecule. We also use this trajectory to plot the energy vs specific cation-perovskite atom distance, as shown in **Supplementary Figure 15** below.

Supplementary Figure 15 | Transition state and energy barrier analysis using nudged elastic band (NEB). The structures are obtained by first relaxing the pre-snapshot and post-snapshots extracted from 3T-FF trajectory using VASP. The transition state structure is then obtained using NEB utilizing the relaxed pre-snapshot and post-snapshot, in addition to a 3T-FF snapshot located in between the pre- and post-snapshots. The transition state structure is perturbed slightly in either direction according to the transition state vibration mode, and further relaxed using VASP to generate additional trajectory in the immediate vicinity of the transition state. We show the structures' DFT energy (**a**), distance between bonding C and I atoms $r_{C-1,1}$ (**b**) and $r_{C-1,2}$ (**c**), as well as the z axis Cartesian coordinate of TE4PBA nitrogen atom (+z axis points into the surface) (**d**) which undergoes a clear step function during the state transition. The relaxed pre-snapshot (**e**), transition state (**f**), and relaxed post-snapshot (**g**) are shown. The sidechain H atoms which are restricted from moving along the x axis during transition state vibration mode analysis are highlighted using the red circles, while the N atom of TE4PBA is highlighted using the light blue circle. The bonding C and I atoms $r_{C-1,1}$ and $r_{C-1,2}$ are also shown.

Comment 11:

Figure 2: The abbreviations "TCO", "ETL" and "HTL" are not spelled out in full. Also, I believe there is a typographical error in subfigure b, possibly a missing "for" ("[...]enabled for TE4PBA[...]").

We thank the reviewer for noticing these errors. The figure caption has now been fixed.

. **Figure 2** caption. We modified the sentence:

- a) Typical device structure of FAPbI₃ perovskite solar cell deposited on front transparent conducting oxide (TCO) glass sandwiched by electron and hole transport layers (ETL, HTL), with surface passivation molecules being applied on the bottom surface of the perovskite.
- b) 3T multi-scale structure transformation modes enabled for TE4PBA organic cation during ab-initio 3T relaxation.

Comment 12:

Page 10, Lines 17–20: The analogy of an ab initio reactor warrants a citation to Martinez and coworkers' ab initio nanoreactor although the methods differ in that Martinez and coworkers used AIMD whereas the present work uses static DFT: Nature Chem. 2014, 6, 1044.

We thank the reviewer for this suggestion. The reference has now been added.

. Main text **page 11, line 20**. We added the sentence citing the suggested reference:

Previous ab-initio nanoreactor approaches utilizes AIMD,^[1] while our approach only uses static DFT calls.

- [1] Wang, L.-P. *et al.* Discovering chemistry with an ab initio nanoreactor. *Nat. Chem.* **6**, 1044–1048 (2014).

Comment 13:

Page 14, Line 20: What does "difficult to be resolved experimentally" mean? Does this mean that the hydrocarbon chain products have not been identified experimentally? Clarification would be appreciated.

We thank the reviewer for this clarification request. Our software has been utilized by a customer electric vehicle battery company for almost a year. It has been relatively easier for our collaborator to confirm the existence of many of the small compounds generated by our electrochemical reaction generator through their experiments (by analyzing battery electrolyte decomposition gas byproduct using mass spectrometry, gas chromatography, Fourier transform infrared, etc). However, longer hydrocarbon chain molecules generated in the electrolyte are not released as gas molecules and will further react to form the polymer part of the solid electrolyte interphase (SEI) inside the battery. It is correspondingly very difficult to validate the generated 3T-VASP byproduct if they are not small enough to be released in the gaseous form during the electrolyte decomposition experiment.

. Main text **page 15, line 20**. We modified the sentence:

We also observe ethylene dicarbonate and glyoxal formation (both are experimentally known) in addition to many long hydrocarbon chain oxidation byproducts which are more difficult to be confirmed experimentally because they are not released as gas molecules during the electrolyte decomposition experiment (**Figure 4a, Figure S9–S10**).

Comment 14:

Page 17, Lines 10–12: I would like the authors to comment on what kinds of reactivity the authors imagine that 3T-VASP could be used to study in the future. Potential limitations?

We thank the reviewer for this suggestion. We have added more of our thoughts on potential usage and limitations of the methodology.

□ Main text **page 19, line 13**. We added the paragraph:

We envision that 3T-VASP (or its future variants when coupled with lower or higher accuracy ab-initio methods) can be useful for the exploration of chemical reactions in complex systems including those involving interfaces such as battery SEI formation, reaction on catalyst surfaces, or coating deposition and degradation on semiconductor device surfaces. Our early effort in this direction is shown in **Figure S12**. However, we note that a relatively good force field is necessary to first perform 3T-FF and generate physically reasonable initial structure for the subsequent 3T-VASP. In addition, while our workflow can efficiently explore nearby local energy minimum structures, it is currently incapable of incorporating temperature dependency for the generated reaction products.

Reviewer #1 (Remarks on code availability):

The code presented by Mailoa and coworkers is well documented with clear instructions on installation and execution. I was able to successfully run the example calculations provided by the authors, however I did not attempt to explore any other chemical systems. Running the provided examples yielded the same results as those presented by the authors.

We thank the reviewer for this positive comment.

Reviewer #2

1. *In this manuscript, the authors propose a multiscale approach to identify products of chemical reactions. They focus on two systems: a complex floppy organic molecule passivator binding onto perovskite solar cell 20 surface, and lithium-ion battery liquid electrolyte. The multiscale approach (named 3T) is described in a manuscript available online, but not peer-reviewed. The approach is quite complicated to apply in practice from what I understand, but I simply consider it as an efficient way of generating low energy atomic configurations, enabling exploration of a wide energy landscape by avoiding local minimas. I find the research work not well presented, but I also see some possible interest for the community.*

We thank the reviewer for this comment. The reviewer's interpretation of the manuscript's approach and goal is accurate, and we will do our best to address the reviewer's concerns. While we agree that the mathematical construction of the method looks complicated, in practice the software user simply needs to input reactant molecules' xyz files or surface structure VASP lattice POSCAR file and specify how many of the reactant molecules to be put into the reaction box. The rest of the workflow (hierarchical segmentation, classical force field assignment, and 3T FF/ab-initio optimizations) are automated. Beyond this, we believe that this work represents a novel way of integrating machine learning framework's autograd capabilities, tensor-based geometry structure transformations, and ab-initio calculation for efficient computational material science workflow.

2. *I find several issues with the way the work is presented in this manuscript. First of all, I find the comparison/reference to MD somewhat misleading in the introduction. The methodology should instead be compared to other approaches that attempt to identify reaction products, such as ["Elementary Decomposition Mechanisms of Lithium Hexafluorophosphate in Battery Electrolytes and Interphases" by Evan Walter Clark Spotte-Smith, Thea Bee Petrocelli, Hetal D. Patel, Samuel M. Blau, and Kristin A. Persson, ACS Energy Letters 2023 8 (1), 347-355, DOI: 10.1021/acseenergylett.2c02351] for example, in the field of lithium-ion battery liquid electrolyte.*

We thank the reviewer for this feedback. We do see our work as a potential alternative to AIMD-based approaches (but only when temperature dependency is not required, because we do not have temperature in our workflow) because the value of our work lies in the exploration of chemical reactions when the products / hypothesized reaction pathways are not yet known. Our approach is more on organically generating the reactions in an ab-initio manner without any prior knowledge, like the ab-initio nanoreactor work of Martinez et al [1].

We believe that our approach is complimentary to the work by E.W.C. Spotte-Smith *et al* instead [2]. Their work [2] relies on utilizing prior knowledge about possible reaction mechanisms/reactants/products and utilizing DFT (or other methods) to test the reaction hypothesis on these selected sets of molecules, and finally determine the possible transition states for these reactions. For example, the authors know in

advance that they should react $\text{H}_2\text{O}+\text{PF}_5$ molecules, or $\text{PF}_5+\text{Li}_2\text{CO}_3$ to reach certain known electrolyte reaction byproducts (POF_3). The reaction pathways of these pre-determined reactants are performed without their explicit solvation structure using the Jaguar software. Our 3T approach 'generates' the reaction trajectories (and correspondingly automatically determines the products and reactants) within their explicit solvation structure without having to pre-determine which/how many molecules are the reactants. It is more useful to utilize the reaction coordinates extracted from the 3T trajectory as potential input candidates to the workflow of [2] instead, where a more thorough analysis for specific reactions of interest (such as transition state analysis) can be performed.

[1] L.-P. wang, *et al.* *Nat. Chem.* **6**, 1044 (2014).

[2] E.W.C. Spotte-Smith, *et al.* *ACS Energy Lett.* **8**, 347 (2023).

We have now modified our manuscript to make this comparison more obvious and included the corresponding reference to [2] for comparison.

In addition, in conjunction to our response to Reviewer 1, we have also shown an example of how a transition state analysis can be performed 'after' the 3T workflow (3T-FF + 3T-VASP) has generated the reaction trajectories, adding on our comparison to transition-state analysis like that performed by Spotte-Smith, *et al.*

. Main text **page 5, line 19**. We modified the sentences and added the references:

Using our approach, it is possible to organically generate chemical reaction products within their explicit solvation structure without having to pre-determine the exact reactants and reaction mechanisms. The reaction byproduct trajectories generated by 3T can be used for further individual reaction energy barrier study such as those commonly done using nudged elastic band^[1-2] or for transition-state analysis,^[3] making 3T a suitable complement to existing *state-of-the-art* methods which are typically done on systems with pre-determined reactants and mechanisms in the absence of explicit solvation structures.

[1] Henkelman, G., Uberuaga, B. P. & Jonsson, H. A climbing image nudged elastic band method for finding saddle points and minimum energy paths. *J. Chem. Phys.* **113**, 9901–9904 (2000).

[2] Ásgeirsson, V. *et al.* Nudged elastic band method for molecular reactions using energy-weighted springs combined with eigenvector following. *J. Chem. Theory Comput.* **17**, 4929–4945 (2021).

[3] Spotte-Smith, E. W. C., Petrocelli, T. B., Patel, H. D., Blau, S. M. & Persson, K. A. Elementary decomposition mechanisms of lithium hexafluorophosphate in battery electrolytes and interphases. *ACS Energy Lett.* **8**, 347–355 (2023).

. Main text **page 9, line 9**. We added the sentence related to transition state analysis:

Additional transition state and energy barrier analysis using snapshots extracted from 3T energy minimization trajectory is available in the **Supplementary Information**.

. **Supplementary Information**. We added the section on additional perovskite surface passivation transition state and energy barrier analysis, as well as **Supplementary Figure 15**:

Perovskite Surface Passivation Transition State and Energy Barrier Analysis

We further perform transition state analysis and determine the energy barrier of our FAPbI_3 A-site vacancy defect passivation using TE4PBA cation, using the trajectory generated by our 3T minimization. The cation embedding is first observed during the 3T-FF cycle, so we extracted a snapshot from the end of 3T-FF (post-snapshot, with the cation already deeply embedded into the defect site), a snapshot in the 3T-FF trajectory where the cation has not been embedded into the defect site yet (pre-snapshot), and a third snapshot with the cation being positioned somewhere in the middle of the pre-snapshot and post-snapshot (mid-snapshot) which is generated by nudged elastic band (NEB). The pre-snapshot and post-snapshot are then relaxed using standard VASP relaxation, as we need these snapshots to be in their respective local energy minimums. We then perform NEB using VASP which includes VTST (Transition State Tools for VASP) tools on the three structures (relaxed pre-snapshot, mid-snapshot, and relaxed post-snapshot). An intermediate saddle-point structure is obtained, with an energy barrier of 0.932 eV higher than the relaxed pre-snapshot.

This intermediate snapshot is then used to perform a transition state analysis. Because the TE4PBA is a very large and floppy molecule, we observe 2 trivial sidechain rotations vibration modes (one $-\text{CH}_3$ group and one $-\text{CH}_2$ group rotation) in addition to the real transition state vibration mode representing the cation embedding into the defect site. Five individual hydrogen atoms which are located away from the core vibration region of the transition state vibration mode are restricted from moving in the x axis to eliminate the trivial sidechain rotations, and only the real transition state vibration mode representing the molecule embedding event remains. We then slightly perturb this cation structure according to the transition state vibration mode in both directions (toward the relaxed pre-snapshot and the relaxed post-snapshot directions) and relax the structure using standard VASP relaxation, producing two trajectories where the transition state TE4PBA cation moves back in the direction of either the relaxed pre-snapshot or to the relaxed post-snapshot. Note the structure and energy gap between the relaxed pre/post-snapshots and the structures relaxed from the NEB transition state. This indicates that there are additional transition states and energy barriers in the reaction trajectory landscape because the TE4PBA cation is a very large and floppy molecule. We also use this trajectory to plot the energy vs specific cation-perovskite atom distance, as shown in **Supplementary Figure 15** below.

Supplementary Figure 15 | Transition state and energy barrier analysis using nudged elastic band (NEB). The structures are obtained by first relaxing the pre-snapshot and post-snapshots extracted from 3T-FF trajectory using VASP. The transition state structure is then obtained using NEB utilizing the relaxed pre-snapshot and post-snapshot, in addition to a 3T-FF snapshot located in between the pre- and post-snapshots. The transition state structure is perturbed slightly in either direction according to the transition state vibration mode, and further relaxed using VASP to generate additional trajectory in the immediate vicinity of the transition state. We show the structures' DFT energy (**a**), distance between bonding C and I atoms $r_{C-1,1}$ (**b**) and $r_{C-1,2}$ (**c**), as well as the z axis Cartesian coordinate of TE4PBA nitrogen atom (+z axis points into the surface) (**d**) which undergoes a clear step function during the state transition. The relaxed pre-snapshot (**e**), transition state (**f**), and relaxed post-snapshot (**g**) are shown. The sidechain H atoms which are restricted from moving along the x axis during transition state vibration mode analysis are highlighted using the red circles, while the N atom of TE4PBA is highlighted using the light blue circle. The bonding C and I atoms $r_{C-1,1}$ and $r_{C-1,2}$ are also shown.

3. *I also find that, given that the 3T methodology has been described elsewhere, the authors should focus on what is specific to the physical systems presented here, and how the reactions products are actually coming out of about 50 DFT calculations. This is in my opinion the most important and interesting aspect of the approach, and the level of details in the current manuscript is insufficient in my opinion. Do they carry out 50 relaxation steps at the DFT level starting with the configurations obtained from 3T? Or something else?*

We thank the reviewer for this clarification request. 3T is just a multi-scale energy optimization algorithm. It relies on other method which can calculate system atomic forces (such as classical force field or VASP DFT) to perform its work. For each material system in our manuscript, we use both 3T-FF and 3T-VASP. In the perovskite system, the structure is optimized using 5×200 calls to classical force field atomic forces, followed by 50 calls to VASP DFT atomic forces. Unlike traditional approaches which optimize the structure 'per-atom', we use 3T algorithm to move the atoms in a coordinated manner. Computation-wise, this means during 3T-FF, we call FF - \rightarrow 3T - \rightarrow FF - \rightarrow 3T - \rightarrow FF - \rightarrow ... And during 3T-VASP, we call VASP - \rightarrow 3T - \rightarrow VASP - \rightarrow 3T - \rightarrow VASP - \rightarrow ... Note that the FF and VASP function calls only generate atomic forces and do not generate any atom movement; only the 3T structure optimization calls are allowed to move the atoms. The VASP calls are individual-frame self-consistent field (SCF) calls.

The two generated 3T-FF and 3T-VASP trajectories are continuous in nature (the initial structure for 3T-VASP is the final structure from the 3T-FF trajectory). This has previously been mentioned in the original main text (in this revision it is main text **page 8, line 23**) and further discussed for 3T inter-cycle transition coordinate transfer in the **Methods** section (in this revision it is **page 27, line 12**).

Please see reply to **point #5** below for our combined manuscript revision addressing the concerns outlined by the reviewer in **point #3** regarding the need to highlight the most important and interesting aspect of the approach.

- **Figure 2** caption. We modified the sentence:

c) 3T optimization implementation utilizing PyTorch for gradient propagation and external calculator for potential energy and atomic forces calculation (see main text for symbol details). During the 3T-FF, the computation workflow is FF \rightarrow 3T \rightarrow FF \rightarrow 3T \rightarrow ... and during the 3T-VASP it is VASP \rightarrow 3T \rightarrow VASP \rightarrow 3T \rightarrow ... FF and VASP are only atomic force/energy calculator of static atom frames, and only 3T is allowed to optimize the atomic structure in the PyTorch framework.

- Main text **page 8, line 23**. The original sentence:

At the end of 3T-FF's last cycle step , we simply use the final 3T-FF structure $\vec{r}_{i,0}$ as the new initial structure for 3T-VASP $\vec{r}_{i,0}$, which ensures continuous 3T trajectory (see **Methods**).

- Methods section **page 27, line 12**. The original sentences:

We re-initialize the 3T inputs into a new cycle with $\vec{r}_{i,1} = \vec{r}_i$ and $\vec{r}_{i,1} = \vec{r}_i$. Under this problem re-initialization framework, the trajectories are continuous ($\vec{r}_{i,1} = \vec{r}_i$).

4. *How do they know the relative importance of each reactant observed? What are the groups used in 3T for the electrolyte?*

3T reactions are generated organically for given reactant combinations without prior assumptions, making this aspect somewhat similar to AIMD. We do not know in advance the relative importance of each reactant observed, and we simply try to keep the reactant proportion similar to what one will find in a lithium-ion battery electrolyte (mass density, electrolyte composition, salt concentration). It is only after sufficient reaction trajectory statistics has been generated, that we observe specific reactions being significantly more prevalent than other reactions (**Supplementary Table 1**). It is up to the user to decide which reactants are of interest for further study. In our case, this was done through a discussion with our electric vehicle battery customer. After the reduction experiment, we decided to proceed with artificially increasing the reactant concentrations CO_3^{2-} (due to its prevalence) and oEC^{\cdot} (because it is a very reactive radical which will rapidly attack other electrolyte compounds) during our subsequent oxidation experiment (originally in main text **page 15, line 1**).

The groups used in 3T for the electrolyte has been referred to in the original main text and was included in the original **Supplementary Information** of the manuscript. We have tried to keep this implementation detail out of the main text for simplicity.

- Main text **page 15, line 1**. The original sentences:

We further investigate the reverse situation (oxidating environment) using 3T. In addition to the five input species which we use in the reduction case, we add CO_3^{2-} , C_2H_4 , and oEC^{\cdot} as input molecules to explore the *EC* decomposition pathways. The first two are relatively abundant in the electrolyte (from *EC* reduction), while the last one is a highly reactive radical species which has been proposed to be responsible for *EC* dimerization reactions.

- Main text **page 11, line 24**. The original sentence:

See **Methods** and **Figure S2** for the automated 3T hierarchical segmentation setup for these molecules.

- Supplementary Information **page 6, line 1**. The original paragraph: **Electrolyte**

Molecule Segmentation and 3T Redox Box Setup

The molecules are segmented into micro-groups based on their rotatable bonds as defined by RDKit. Entire molecules are grouped into separate macro-groups. Based on this segmentation criteria, the micro-group segmentation is shown below for the electrolyte input molecules:

Supplementary Figure 2 | Micro-group segmentation of electrolyte input molecules.

5. *I feel in general that the paper is lacking focus on some the important aspects of interest to the community, while diverging on less important aspects such as implementation details. The manuscript also relies too much on information found in the supplementary materials, precisely because the authors are trying to cover too much materials. They should focus the main text on explaining the main ideas and rely on supplementary materials for implementation details.*

We thank the reviewer for this feedback. We have now recognized that we put too much emphasis on method details and results in the manuscript, while lacking enough details on the physical concept aspects of our method.

We think that 3T is fast in converging its reaction trajectory generation, primarily because of its multi-scale gradient optimization approach. As an analogy, a typical per-atom structure optimization approach can be thought of as individual soldiers deciding on his/her own best course of action, based on the soldier's local environment information. The 3T approach on the other hand will involve optimization on multiple scales, where soldiers can only make their optimization decisions within the constraint of high-level commands issued by their general and commanders (micro-group and macro-group level optimizations). If it is not optimal for a molecule's atom to move in certain directions, but beneficial for a molecule fragment or the molecule as whole to rotate/translate in such manner to optimize the system energy, the 3T coordinated molecule movement can force the atom to jump over its local energy barrier regardless of the atom's local energy minimum gradients. This seems to make 3T relatively good for the exploration of diverse local energy minimums. Correspondingly, this means 3T is not guaranteed to reach the global system energy minimum.

We have now added a discussion section explaining our observations and the idea behind our approach. As for the main text content, we have tried to relegate as many details as possible of the 3T inner working to the **Supplementary Information** in the original version of the manuscript (exact equations, segmentation details, force field, etc). We believe that the micro / macro-group hierarchical segmentation and the coupling of atomistic force gradient to PyTorch should be kept in the main text because they are fundamentally important for describing the high-level workflow of the algorithm. Can we receive a suggestion from the reviewer on which additional content details the reviewer think is less relevant to the community and should be moved into the **Supplementary Information**?

- Main text **page 18, line 1**. We added the section:

Discussion on 3T Coordinated Multi-Scale Optimization Speed

In this section, we briefly discuss about the relatively fast convergence speed of our 3T multi-scale optimization approach. In our work, this fast convergence speed iteration is observed during both 3T-FF and 3T-VASP stages, and it seems to be relatively independent from the underlying atomic forces/energy calculator being used. During the FF stage, we observe 3T being capable of rapidly changing the floppy cation conformations on the perovskite surface and being able to rapidly disperse the electrolyte molecules into its liquid form inside the periodic boundary condition box without having to do any MD. During the VASP stage, we observe 3T being capable of rapidly minimizing

perovskite-cation energy or initiating and converging a chemical reaction between nearby molecules. We note that reactions do not happen during the 3T-FF stage, as our classical force field does not allow chemical reactions to happen. However, all these fast convergence behaviors will be turned off if we disable micro-group (A_m , R_m and T_m) and macro-group (R_M and T_M) level structure transformations and optimizations. In such case, we end up with a standard per-atom optimization algorithm which suffers from atoms being trapped in local energy minimums. When the micro-group and macro-group level optimizations are enabled, the 3T optimizer can automatically optimize the rotation and translation parameters for many atoms in a coordinated manner, forcibly moving an atom up its local energy barrier hill while still taking the atom's local energy gradient into account during the optimization. While the gradient calculation is done by 3T in the PyTorch framework using autograd, the actual optimization step is performed by PyTorch Adam optimizer. This also allows our workflow to further benefit from neural network advances made by the machine learning community.

Reviewer #3

1. *The authors have recently developed a new method that is able to efficiently find minima on the PES of molecular systems based on a representation in non-Cartesian, multi-scale coordinates. In the present work, they extend and apply it to reactive systems to identify geometries of floppy molecules binding to a surface and to reaction products of battery electrolytes in reductive and oxidative environments. The method can identify possible products, but apparently does not analyse the pathways and barriers to them, therefore it cannot assess if they are thermally accessible at the moment. Such a method is still very useful, but also not new. Many such methods exist, and a major weakness of the present work is that literature on this topic is not mentioned at all. See e.g. some recent reviews (10.1002/wcms.1354, 10.1021/acs.jpca.8b10007, 10.1007/s11244-021-01543-9) and more recent developments.*

In my opinion, a brief description of the state of the art and an assessment of improvements of the newly proposed method is essential for scientific work. Furthermore, a substantial improvement should be shown for publication in a high impact journal.

We thank the reviewer for this suggestion. The literature review section in the introduction has now been expanded to include additional methods and references. We have also included the reference to the review articles above to further introduce the state-of-the-art on reaction generation literature. In our response to **point #2**, we perform a comparison of our method with CREST + DFT as recommended by the reviewer. We have also added an example of reaction energy barrier and transition state analysis for our perovskite passivation example.

□ Main text **page 3, line 20**. We added the sentences:

Other promising reaction exploration approaches are temperature-independent, and they work by modifying steady-state structure optimization method to escape local energy minimum. For example, the AFIR approach works by adding multi-parameter empirical artificial forces between all atom pairs.^[1] The approach is further extended in the implementation of Chemoton 2.0 software, where the combination of brute force conformer sampling and empirical artificial force added on chosen reactive sites between candidate reactant molecule pairs are used to accelerate a reaction.^[2] When the reactants and products are already known in advance, reaction transition states can be found using different methods, for example by using the growing string method which sequentially places intermediate nodes in the delocalized internal coordinates (bonds, angles, or torsions) relevant for reaching the desired reactive outcome.^[3] A combination of temperature-dependent metadynamics approach for generating reaction and subsequent transition state analysis is also routinely performed.^[4-5] We also note that our work has some similarities to internal coordinate methods recently used in non-periodic quantum chemistry software for structure minimization,^[6-8] albeit with

some extensive differences (see **Supplementary Information**). More in-depth reviews of these reaction generation methods are available in the literature.^[9-10]

- [1] Maeda, S. & Harabuchi, Y. Exploring paths of chemical transformations in molecular and periodic systems: An approach utilizing force. *Wiley Interdiscip. Rev. Comput. Mol. Sci.* **11**, e1538 (2021).
- [2] Unsleber, J. P., Grimm, S. A. & Reiher, M. Chemoton 2.0: Autonomous exploration of chemical reaction networks. *J. Chem. Theory Comput.* **18**, 5393–5409 (2022).
- [3] Zimmerman, P. M. Single-ended transition state finding with the growing string method. *J. Comput. Chem.* **36**, 601–611 (2015).
- [4] Martínez-Núñez, E. *et al.* AutoMeKin2021: An open-source program for automated reaction discovery. *J. Comput. Chem.* **42**, 2036–2048 (2021).
- [5] Jara-Toro, R. A., Pino, G. A., Glowacki, D. R., Shannon, R. J. & Martínez-Núñez, E. Enhancing automated reaction discovery with boxed molecular dynamics in energy space. *ChemSystemsChem* **1**, e1900024 (2019).
- [6] Wang, L. P. & Song, C. Geometry optimization made simple with translation and rotation coordinates. *J. Chem. Phys.* **144**, (2016).
- [7] Shajan, A., Manathunga, M., Götz, A. W. & Merz, K. M. Geometry optimization: A comparison of different open-source geometry optimizers. *J. Chem. Theory Comput.* **19**, 7533–7541 (2023).
- [8] Hermes, E. D., Sargsyan, K., Najm, H. N. & Zádor, J. Sella, an open-source automation-friendly molecular saddle point optimizer. *J. Chem. Theory Comput.* **18**, 6974–6988 (2022).
- [9] Dewyer, A. L., Argüelles, A. J. & Zimmerman, P. M. Methods for exploring reaction space in molecular systems. *Wiley Interdiscip. Rev. Comput. Mol. Sci.* **8**, e1534 (2018).
- [10] Simm, G. N., Vaucher, A. C. & Reiher, M. Exploration of reaction pathways and chemical transformation networks. *J. Phys. Chem. A* **123**, 385–399 (2019).
- [11] Steiner, M. & Reiher, M. *Autonomous reaction network exploration in homogeneous and heterogeneous catalysis*. *Topics in Catalysis* vol. 65 (Springer US, 2022).

- Main text **page 9, line 9**. We added the sentence:

Additional transition state and energy barrier analysis using snapshots extracted from 3T energy minimization trajectory is available in the **Supplementary Information**.

- Supplementary Information**. We added the section on additional perovskite surface passivation transition state and energy barrier analysis, as well as **Supplementary Figure 15**.

Perovskite Surface Passivation Transition State and Energy Barrier Analysis

We further perform transition state analysis and determine the energy barrier of our FAPbI₃ A-site vacancy defect passivation using TE4PBA cation, using the trajectory generated by our 3T minimization. The cation embedding is first observed during the 3T-FF cycle, so we extracted a snapshot from the end of 3T-FF (post-snapshot, with the cation already deeply embedded into the defect site), a snapshot in the 3T-FF trajectory where the cation has not been embedded into the defect site yet (pre-snapshot), and a third snapshot with the cation being positioned somewhere in the middle of the pre-snapshot and post-snapshot (mid-snapshot) which is generated by nudged elastic band (NEB). The pre-snapshot and post-snapshot are then relaxed using standard VASP relaxation, as we need these snapshots to be

in their respective local energy minimums. We then perform NEB using VASP which includes VTST (Transition State Tools for VASP) tools on the three structures (relaxed pre-snapshot, mid-snapshot, and relaxed post-snapshot). An intermediate saddle-point structure is obtained, with an energy barrier of 0.932 eV higher than the relaxed pre-snapshot.

This intermediate snapshot is then used to perform a transition state analysis. Because the TE4PBA is a very large and floppy molecule, we observe 2 trivial sidechain rotations vibration modes (one -CH₃ group and one -CH₂ group rotation) in addition to the real transition state vibration mode representing the cation embedding into the defect site. Five individual hydrogen atoms which are located away from the core vibration region of the transition state vibration mode are restricted from moving in the x axis to eliminate the trivial sidechain rotations, and only the real transition state vibration mode representing the molecule embedding event remains. We then slightly perturb this cation structure according to the transition state vibration mode in both directions (toward the relaxed pre-snapshot and the relaxed post-snapshot directions) and relax the structure using standard VASP relaxation, producing two trajectories where the transition state TE4PBA cation moves back in the direction of either the relaxed pre-snapshot or to the relaxed post-snapshot. Note the structure and energy gap between the relaxed pre/post-snapshots and the structures relaxed from the NEB transition state. This indicates that there are additional transition states and energy barriers in the reaction trajectory landscape because the TE4PBA cation is a very large and floppy molecule. We also use this trajectory to plot the energy vs specific cation-perovskite atom distance, as shown in **Supplementary Figure 15** below.

Supplementary Figure 15 | Transition state and energy barrier analysis using nudged elastic band (NEB). The structures are obtained by first relaxing the pre-snapshot and post-snapshots extracted from 3T-FF trajectory using VASP. The transition state structure is then obtained using NEB utilizing the relaxed pre-snapshot and post-snapshot, in addition to a 3T-FF snapshot located in between the pre- and post-snapshots. The transition state structure is perturbed slightly in either direction according to the transition state vibration mode, and further relaxed using VASP to generate additional trajectory in the immediate vicinity of the transition state. We show the structures' DFT energy (**a**), distance between bonding C and I atoms $r_{C-I,1}$ (**b**) and $r_{C-I,2}$ (**c**), as well as the z axis Cartesian coordinate of TE4PBA nitrogen atom (+z axis points into the surface) (**d**) which undergoes a clear step function during the state transition. The relaxed pre-snapshot (**e**), transition state (**f**), and relaxed post-snapshot (**g**) are shown. The sidechain H atoms which are restricted from moving along the x axis during transition state vibration mode analysis are highlighted using the red circles, while the N atom of TE4PBA is highlighted using the light blue circle. The bonding C and I atoms $r_{C-I,1}$ and $r_{C-I,2}$ are also shown.

- 2. The authors propose a 2-level approach (1st FF, then DFT), which seems to be important. How does this compare to crest + DFT (10.1021/acs.jctc.9b00143, 10.1039/C9CP06869D), a method I consider nowadays state of the art, not just simple energy minimisation, which is used for comparison by the authors. A comparison of results and computational demands of both methods would strengthen the work considerably.*

We thank the reviewer for this feedback. CREST works by generating multiple likely conformer geometries for a given molecule in the absence of explicit solvation structure (or other molecule-confining structures), which can be used for further processing such as DFT-based structure relaxation. This method is most useful when working with molecules with large number of possible conformers, as it is then possible to narrow down the range of most energetically favorable geometries. In the context of our work, CREST seems to be more relevant to be used in our perovskite example because our cation is a large floppy organic molecule placed on a relatively empty space on the perovskite surface. Most of the reactants in our liquid electrolyte example use large number of small molecules with small number of possible molecule conformations and they are placed in a relatively packed liquid state in a PBC box, making it less suitable for CREST. We have correspondingly added an additional CREST + DFT baseline comparison for our perovskite work.

- Main text **page 9, line 11**. We added the paragraph:

We have also included an additional baseline comparison where the TE4PBA cation conformation search on the perovskite surface is performed using CREST before VASP relaxation is performed (see **Supplementary Information**). This method requires relatively long conformation search (5 days) and slightly less DFT calls than standard VASP relaxation. However, it still requires significantly more DFT calls than 3T-VASP. The generated cation structure still has relatively higher energy than 3T relaxation (0.36–0.73eV higher than the best 3T structure) and can only slightly enter the defect site (**Figure 2f, Figure S14**).

- Main text **Figure 2**. We modified **Figure 2f**:

Figure 2 | 3T relaxation of floppy organic cation on FAPbI₃ surface vacancy defect site. **a)** Typical device structure of FAPbI₃ perovskite solar cell deposited on front transparent conducting oxide (TCO) glass sandwiched by electron and hole transport layers (ETL, HTL), with surface passivation molecules being applied on the bottom surface of the perovskite. **b)** 3T multi-scale structure transformation modes enabled for TE4PBA organic cation during ab-initio 3T relaxation. **c)** 3T optimization implementation utilizing PyTorch for gradient propagation and external calculator for potential energy and atomic forces calculation (see main text for symbol details). During the 3T-FF, the computation workflow is FF-**3T-**FF-**3T-**... and during the 3T-VASP it is VASP-**3T-**VASP-**3T-**... FF and VASP are only atomic force/energy calculator of static atom frames, and only 3T is allowed to optimize the atomic structure in the PyTorch framework. Final TE4PBA-FAPbI₃ structures obtained using **d)** standard VASP relaxation and **e)** 3T-VASP relaxation, with the cation originally floating above the defect site. In the case of **d)** the cation still floats above the defect site, while in the case of **e)** the cation embeds itself deep into the FAPbI₃ surface vacancy defect site. **f)** DFT binding energy vs the number of DFT call comparison between standard VASP, CREST-VASP, and 3T-VASP relaxation. Whereas standard relaxation fails to find a cation binding pose on FAPbI₃ surface even after 500 DFT steps, 3T relaxation consistently finds deep defect binding pose with significantly lower DFT binding energy within 50 DFT steps. CREST-VASP can achieve lower energies than standard VASP relaxation, but the cations are still not deeply embedded into the perovskite defect site in addition to still taking significantly more DFT steps to converge compared to 3T-VASP.********

□ **Supplementary Information.** We added the section on additional CREST-VASP baseline for perovskite passivator structure optimization, as well as **Supplementary Figure 14.**

Additional Perovskite Surface Cation Passivation Baseline Comparison using CREST + VASP Relaxation

It is also possible to obtain better passivation structure for single TE4PBA cation passivation on the FAPbI₃ surface vacancy defect by first performing CREST-based TE4PBA conformation search on the FAPbI₃ surface before performing standard VASP relaxation. Multiple conformers for the large floppy TE4PBA cation have been generated in the presence of defective FAPbI₃ perovskite surface using CREST.

The TE4PBA cation is very large and floppy, so the conformation search took a long time and we had to stop it after 5 days of conformation search.

To enhance the sampling of molecular structure, we initially conducted molecular dynamics simulations using xTB under the GFN0-xTB framework. The simulations were performed at a temperature of 600 K with a time step of 1 fs for a total duration of 10 ps. Structures were saved every 5 fs, and the parameters $h_{\text{mass}}=1$ and $\text{shake}=0$ were set, resulting in a total of 2000 sampled structures. These 2000 molecular structures were then subjected to batch optimization using the `mdopt` function of CREST under the GFN0-xTB level of theory. Subsequently, the optimized structures were processed through `isostat` of Molclus for duplicate removal and energy ranking, with an energy duplicate threshold of 0.5 kcal/mol and a structural duplicate threshold of 0.5 Å, without calculating the Boltzmann distribution ratio. Following clustering, 347 structures remained. The top 100 structures were further optimized using the `mdopt` function of CREST under the GFN1-xTB level of theory. Using the same configuration, `isostat` was again employed for duplicate removal and energy ranking for post-optimized structures. This step did not eliminate any structures.

Three TE4PBA conformation structures with the lowest CREST energies are chosen. Because CREST also distorts the perovskite surface structure (even though we have set the CREST settings to freeze the surface atoms), we then transfer the CREST-generated conformer molecule coordinates onto the original defective perovskite surface on top of the A-site vacancy defect. An additional standard VASP relaxation is then performed. We can see from the result in main text **Figure 2f** that while CREST + VASP is better than the baseline standard VASP relaxation, CREST + VASP results are still worse than 3T-VASP (in addition to taking significantly more DFT relaxation steps and prior lengthy cation conformation search). The end structures of the CREST + VASP relaxation structures are shown in **Supplementary Figure 14** below.

Supplementary Figure 14 | TE4PBA cation passivation structure on FAPbI₃ defect site after CREST conformation search followed by VASP relaxation. These cations can slightly enter the defect sites, although the binding energies are still higher than the deeply embedded cation structures found by 3T relaxation.

3. What is "reduction ionization"?

We thank the reviewer for this clarification request. In our work, we observe situations where a molecule chemically reacts into a different compound through a reduction reaction and situations where a molecule simply undergoes a change in its charge state (into [-1] or [-2] state). To differentiate these two cases, we

call the former "reduction reaction", and the latter "reduction ionization". We have now recognized that this may create confusion, so we have renamed the latter "charge reduction" throughout the manuscript.

4. *Line 180: "These electrolyte systems (mass density = 1.24 – 1.46 g/cm³) are intentionally not charge-compensated". How are calculations for periodic non-charge neutral boxes possible? Their energy should be infinity? (a page later the text suggests that the boxes are made charge neutral by adding or removing electrons. Please make sure to avoid ambiguities in your description. (line 203, "In this VASP DFT simulation with neutral charge".))*

We thank the reviewer for this clarification request on the implementation detail. In both classical force field MD and DFT computations, when the number of cations and anions in the system are not charge-balanced, the user typically intentionally adds additional ions to balance the charge in the system (Na⁺ and Cl⁻ are the most common ions to add in classical FF MD simulations), precisely to avoid the infinite energy problem associated with non-charge neutral periodic boundary condition boxes.

In our FF energy/forces calculator, we do not calculate the full infinite periodic boundary condition energy using Ewald summation, and only calculate energy contributions from the PBC box and its adjacent mirror box images. This approximation is suitable because the exact force contribution from the subsequent PyTorch autograd is obtained for all bonded energy contributions (the same is true for all Lennard-Jones interaction), while all Coulombic interaction force contribution within 14 Å radius (and most Coulombic force contributions up to 28 Å radius) is included. We justify that this approximation is valid, given that we only use 3T-FF to generate approximately correct initial structure for the subsequent 3T-VASP energy minimization.

In a DFT software like VASP we do not have the freedom to assign specific partial charges to atoms and by construction the total charge of the VASP system is always zero. For example, if we have one 3 Li atoms and 1 PF₆ molecule in the box in addition to several neutral electrolyte solvent molecules, one possible lowest energy configuration found by 3T-VASP is for VASP to assign an electronic charge of [+1] to the three Li ions, [-1] to the PF₆ molecule, and reduce one of the EC molecules to form CO₂ and C₂H₄. In this way, VASP determines that the lowest energy configuration within a charge neutral box for this system can be obtained by reducing the EC molecule. We have now modified our manuscript to ensure that the term 'not charge-compensated' is not interpreted ambiguously.

□ Main text **page 12, line 1**. We modified the sentences:

These electrolyte systems (mass density = 1.24 – 1.46 g/cm³) are intentionally not charge-compensated (additional ions are not added to neutralize the system). This approach follows a previous AIMD-based study of electrochemical redox in liquid.

5. *In the main text, there is no information how the "simple ab initio energy minimization" can overcome reaction barriers. Some description should be moved from the SI into the main text to make this important point clearer.*

We thank the reviewer for this clarification request. The full implementation detail of the 3T algorithm (such as actual tensor-based geometry transformation equations) are kept in the **Supplementary Information** because they might be too distracting for readers of a broader audience, as pointed out by Reviewer 2. However, we have now recognized that we have not put sufficient discussion into the physical concepts of 3T minimization inner workings, which relies on the availability of coordinated movement of atoms within micro and macro groups. We have now added a corresponding discussion in the main text of the manuscript to make this concept clearer.

□ Main text **page 18, line 1**. We added the section:

Discussion on 3T Coordinated Multi-Scale Optimization Speed

In this section, we briefly discuss about the relatively fast convergence speed of our 3T multi-scale optimization approach. In our work, this fast convergence speed iteration is observed during both 3T-FF and 3T-VASP stages, and it seems to be relatively independent from the underlying atomic forces/energy calculator being used. During the FF stage, we observe 3T being capable of rapidly changing the floppy cation conformations on the perovskite surface and being able to rapidly disperse the electrolyte molecules into its liquid form inside the periodic boundary condition box without having to do any MD. During the VASP stage, we observe 3T being capable of rapidly minimizing perovskite-cation energy or initiating and converging a chemical reaction between nearby molecules. We note that reactions do not happen during the 3T-FF stage, as our classical force field does not allow chemical reactions to happen. However, all these fast convergence behaviors will be turned off if we disable micro-group (A_m , R_m and T_m) and macro-group (R_M and T_M) level structure transformations and optimizations. In such case, we end up with a standard per-atom optimization algorithm which suffers from atoms being trapped in local energy minimums. When the micro-group and macro-group level optimizations are enabled, the 3T optimizer can automatically optimize the rotation and translation parameters for many atoms in a coordinated manner, forcibly moving an atom up its local energy barrier hill while still taking the atom's local energy gradient into account during the optimization. While the gradient calculation is done by 3T in the PyTorch framework using autograd, the actual optimization step is performed by PyTorch Adam optimizer. This also allows our workflow to further benefit from neural network advances made by the machine learning community.

6. There is no analysis of barriers. Therefore, it is unclear if products are chemically accessible. It seems they are actually potential products. There are more efficient algorithms. In reaction mechanism exploration, such candidates are usually checked for accessibility, see reviews mentioned above and comment on this limitation. The addition and removal of electron seems barrierless. Are there actually barriers AFTER adding the electrons?

We thank the reviewer for this feedback. The reviewer is correct, this work has been focused on the reaction acceleration within complex structures such as those involving dense multi-species liquid and solid surfaces. We have commented on the lack of reaction barrier analysis as a limitation of our work and

cited the relevant reaction barrier analysis work in our introduction. We have also added a reaction barrier analysis for our perovskite example (see our response to **point #1**).

As for the addition and removal of electrons, we do not actually add or remove any electron from the entire system. The VASP periodic boundary condition (PBC) system always has zero total system charge. The electrochemical reduction or oxidation in our system happens due to electron transfer between the components in the system (for example the reduction of EC into CO_2 and C_2H_4 may be simultaneously accompanied by two Li^0 oxidation into Li^+). The transfer of electrons between the components in our PBC system may or may not have energy barrier, depending on the exact reaction chemistries which occur.

- Main text **page 12, line 4**. We added a sentence:

Inter-molecule electron transfer happens in most of the reactions that we observe (**Figure S7, S9, S10**).

7. Has the new representation a higher or a lower dimensionality than one using internal coordinates? (In other words: are the large-scale fragments replacing the micro objects or are these additional coordinates?)

We thank the reviewer for this clarification request. Every new stage of 3T geometry transformation operation (micro-group operations A_m, R_m, T_m and macro-group operations R_M, T_M) adds on top of existing coordinate system (initially just the atom coordinates). This makes the 3T optimization parameter hyperspace have a higher dimensionality than the initial per-atom structure optimization problem. This was mentioned in the original main text introduction (now **page 5, line 15**). In physical terms, the operations first move the individual atoms, then move micro-groups of atoms based on A_m, R_m, T_m , and finally move macro-groups of atoms based on R_M, T_M . The macro-groups usually consist of several micro-groups (depending on the molecule species), so the large-scale fragment movement is done after the micro-groups are moved. The optimization is done simultaneously in a hierarchical manner using PyTorch autograd backpropagation.

- Main text **page 5, line 15**. The original sentence:

This effectively projects the structure optimization problem from the Cartesian coordinates $\vec{\mathcal{C}}_t$ into $\vec{\mathcal{C}}^*$ a higher-dimension parameter hyperspace \mathcal{O}_t , where coordinated multi-atom structure transformations can be performed efficiently in physically-inspired manners, often bypassing energy barriers which may be difficult to overcome using per-atom structure relaxation (**Figure 1b**).

- Main text **page 8, line 9**. We modified the sentence:

We set our 3T function as a sequential (output from one transformation is input to the next transformation) and hierarchical (first atom level, then micro-group level, and finally macro-group level) compound geometry transformation of these six functions within PyTorch, governed by the $\vec{\mathcal{C}}^*$ parameter \mathcal{O}_t at step t (**Figure 2c**).

8. In summary, I cannot recommend publication of the work at the present stage. With a major revision, the work will probably become publishable, but it will depend on how much improvement (if any) can be demonstrated over existing methods whether it will be suitable for Nature Communication.

We thank the reviewer for the valuable feedback. We hope that the major revisions we have performed on the manuscript may satisfactorily address the reviewer's concerns.

Reviewer #4

1. *In this manuscript, Mailoa et al. demonstrate that energy minimization within a basis of "structure transformation modes" accesses chemically interesting low-energy states for molecular adsorption and electrochemical reactivity within a modest number of DFT gradient evaluations. The observed electrochemical reduction reaction products seem fairly reasonable, and the potential utility of leveraging the identified species in order to aid in the construction of a chemical reaction network is self evident.*

We thank the reviewer for this positive comment. The reviewer's interpretation of the method's inner working and goals are accurate. We will do our best to address the reviewer's feedback on the manuscript.

2. *However, I do have to take issue with some of the specifics on the oxidation side. In particular, the inclusion of oEC^- as a reactant on the oxidation side makes no sense to me. The authors themselves described oEC^- as "a highly reactive radical species", which is precisely why it is unphysical to expect that oEC^- would diffuse from the negative electrode, where it is created, all the way to the positive electrode, where it could participate in oxidation reactions, without reacting with anything else along the way. In contrast, it is reasonable that Li_2CO_3 exists at the positive electrode. Further, the majority of the oxidation side reactivity that is reported is facilitated by the unphysical presence of oEC^- near the positive electrode. While I recognize that this choice does not take away from the demonstrated utility of the method itself, I do find it very distracting and feel that it undermines the oxidation side example.*

We thank the reviewer for this feedback. We completely agree with the reviewer that the $oEC \bullet$ radical will very quickly react with the nearby electrolyte molecules and will never make it to the positive electrode to undergo oxidation there. In fact, this was never our intent when performing the 'oxidation experiment'. This oxidation experiment was performed based on the request from our electric vehicle battery customer, because they are interested in finding out what happens to these electrochemical reduction byproducts after they are generated (these electrochemical reduction byproducts are not the end products). In fact, $oEC \bullet$ radical is so reactive that it will very likely attack other electrolyte compounds nearby, hence participating in further reactive compounds which eventually form longer polymer chains as parts of the SEI. Some of these reactions will be redox reactions, where the $oEC \bullet$ will reduce other electrolyte molecules, while the $oEC \bullet$ itself (in combination with other nearby electrolyte molecule) is oxidized. The oxidation example we generated are for investigating potential oxidation reactions related to the reactive compounds. In the case of $oEC \bullet$ radical, it is as part of the redox reactions near the negative electrode where the radicals are generated. Evidently, $oEC \bullet$ reacts into many different types of possible compounds, either by decomposing itself or by reacting with nearby electrolyte molecules (**Supplementary Figure 9**). We have now made this point clearer in the manuscript to prevent ambiguity about our oxidation experiment intent.

- **Main text page 15, line 5. We added the sentence:**

In particular, the generated • radical may very quickly attack nearby electrolyte compounds by reducing them, with the • radical itself being oxidized in the process.

3. More generally, while the benefits of the approach in comparison to “per-atom” Cartesian minimization or Cartesian AIMD are emphasized, the authors never mention or compare to the conceptually similar and decades old idea of internal coordinates. Indeed, efficient molecular geometry optimization as implemented in every single molecular DFT package critically depends on transforming from Cartesian coordinates into a basis of “internal” coordinates representing distances between pairs of atoms, three atom angles, and four atom dihedral angles. (More recently, translation and rotation internal coordinates have also been introduced which explicitly describe the collective translations and rotations of entire molecules, or parts of molecules.[1]) Changing any single internal coordinate effectively modifies a large collection of Cartesian coordinates, allowing for optimizers to reach lower energy minima with far fewer gradient calls than is possible when optimizing in pure Cartesian coordinates.[2]

With that context, the question must be asked – have Mailoa et al. simply “rediscovered” internal coordinates? There could still be impact/value here – internal coordinates are not typically used for periodic systems, and the need to back-transform from internal to Cartesian coordinates at each optimization step also introduces numerical challenges when angle or dihedral coordinates become invalid (though the addition of dummy atoms resolves that [3]) while the 3T approach’s clever use of AutoGrad seems to avoid the need to back-transform at all. However, modern molecular optimization algorithms are also quasi-second-order, leveraging a guess Hessian (which is reasonable to Initially approximate as diagonal In Internal coordinates) and then updating that Hessian based on gradient information at each step, while the 3T procedure is purely first order, which seems like a drawback. Further, the current state-of-the-art optimizer Sella (while it is marketed as a transition state optimizer, it is just as effective at optimizing to minima) leverages geodesic stepping to more efficiently move towards a lower energy at each iteration[3].

All of this to say that my ability to evaluate the novelty of the approach described in this manuscript requires the authors to place their work in the context of internal coordinate optimization approaches and more generally compare to modern molecular optimizers like Sella. Indeed, in Sella the user can define whatever custom set of internal coordinates that they would like (or to allow Sella to do so automatically) – how different is that from the 3T procedure where a structure is “automatically segmented hierarchically into different micro-groups (based on its rotatable bonds) and a macro-group (the entire molecule)”? To me, they seem very similar, and how much additional utility the 3T approach provides over a similarly careful use of Sella remains unclear.

We thank the reviewer for highlighting the concept of internal coordinates method. Our method certainly shares a resemblance to the previously published internal coordinates method such as TRIC and its variants [1-2], and less resemblance to that implemented by Sella [3]. For the sake of clarity, we'll divide our responses to the combined points above into comparisons vs TRIC and Sella.

After reading through the principles of TRIC, which freezes the internal coordinates of atoms within a defined sub-unit (for example the individual amino acids of a protein, or entire molecules) and then perform rotation / translation for the given sub-unit, we do see some similarities.

TRIC employs a different representation for one level of sub-unit rotation/translation (using quaternion rotation matrix). While the math of the structure transformation is different, we believe that physically they will do a similar procedure of translating/rotating a set of atoms in a coordinated manner, which accelerates energy minimization. In the case of TRIC, the original Cartesian representation is reduced into a lower dimension set of internal coordinate representation.

However, the difference between 3T and TRIC method is that 3T is a multi-level structure transformation. In 3T, multiple level of hierarchical segmentations can be defined and the optimizations can be performed simultaneously. Unlike the TRIC method which needs to combine the translation and rotation into one layer quaternion matrix multiplication operation (and hence is meant to be performed on one level of user-defined sub-unit), 3T optimization is performed on multiple levels simultaneously (atom level, micro-group level which in our case corresponds to molecule fragments, and macro-group level which in our case corresponds to whole molecules). Hence, our method is not really an internal coordinate method because our parameter optimization hyperspace is of higher dimension than the initial Cartesian coordinate. The snapshot below from `utils/run_utils.py` describes our operation. The input tensor `movable_pos_list` is part of our optimization parameters (equivalent to optimizing per-atom \mathbb{R}^3 translation), in addition to parameters governing the other transformation operations.

```

173 def create_optimizers(model, block):
174     # We directly modify the atom xyz coordinates.
175     # This is just a computation trick equivalent to modifying T_xyz, which saves a bit of compute/memory.
176     theta_atom_translation = [param for param in model.movable_pos_list]
177     optim_params = theta_atom_translation
178
179     # Now we add the micro-groups' translation and rotation
180     theta_micro_translation = model.translation_list
181     theta_micro_rotation = model.rotation_list
182     optim_params += [theta_micro_translation, theta_micro_rotation]
183
184     special_rotation, macro_mode = model.special_rotation, model.macro_mode
185     # Now we add sidechain micro-groups' rotatable bond axis rotation
186     if special_rotation != None:
187         theta_micro_axis_rotation = model.special_rotation_list
188         optim_params += [theta_micro_axis_rotation]
189
190     # Now we add macro-groups' translation and rotation
191     if macro_mode != None:
192         theta_macro_translation = model.macro_mode_translation_list
193         theta_macro_rotation = model.macro_mode_rotation_list
194         optim_params += [theta_macro_translation, theta_macro_rotation]
195
196     optimizer = optim.Adam( optim_params , 3e-2, #1e-2,
197                             weight_decay=0)
198     optimizers = [ optimizer ]
199     return optimizers

```

Because our method relies on the autograd capabilities developed by the machine learning community for high dimension neural network optimization, it is straightforward to add multiple levels of hierarchical segmentation and complex geometry transformation operations to be performed simultaneously, including that involving systems with periodic boundary conditions. One can simply compound new functions, as shown below in `utils/potential_model_3T.py`.

```

351 def arrange_atom_pos(self, movable_pos_list, fixed_pos):
352     if self.special_rotation_idx != None:
353         movable_pos_list = self.axis_rotate(movable_pos_list, fixed_pos)
354
355     movable_pos_list = self.micro_rotate_translate(movable_pos_list)
356
357     if self.macro_mode_idx != None:
358         movable_pos_list = self.macro_rotate_translate(movable_pos_list)
359
360     na = sum([movable_pos.shape[0] for movable_pos in movable_pos_list]) + fixed_pos.shape[0]
361     atom_pos = torch.zeros(na,3).to(self.device)
362     for i in range(len(movable_pos_list)):
363         atom_pos[self.movable_idx_list[i],:] = movable_pos_list[i]
364     atom_pos[self.fixed_idx,:] = fixed_pos
365
366     return atom_pos

```

To the best of our understanding, if we turn off the 3T structure transformation operations per-atom $T(\mathbb{T})$ and micro-group A_m, R_m, T_m (hence we only allow one macro-group rotation R_M and translation operation T_M each), turn off the capability for automatically managing proper intra-group coordinates over periodic boundary condition using PyTorch autograd, and remove the usage of PyTorch Adam optimizer to manage parameter stepping over the calculated gradients, then the 3T method will strongly

resemble that of TRIC (physically at least, as the actual math operations are different). However, this version of 3T will be very different from the work we have presented in the manuscript, as it will no longer be a hierarchical structure optimization method.

We next discuss the comparison between 3T and the Sella optimizer. The Sella optimizer is interesting because it uses a different approach; it uses internal coordinates based on constraints. Bonds, angles, and dihedrals are identified/extracted from a target optimization molecule, which are then utilized as constraints (atom distances, bending angles, dihedral angles). These constraints are utilized for reducing the dimensionality of the optimization problem before performing the optimization based on the method of Lagrange multipliers. In this sense, the Sella optimizer is more like the TRIC method and unlike the 3T method which works in a higher dimension parameter space. Throughout the optimization procedure, these Sella constraints may be violated. However, the final converged structure will fulfil these constraints (will lie on the constraint manifold if the optimization succeeds).

In addition to that, the Sella optimizer can automatically determine the molecule connectivity graph (and hence constraints which must be applied to the molecule) on the fly. In contrast, the 3T-VASP as it is currently designed automatically determines its hierarchical segmentation by processing the individual input molecules using RDKit cheminformatics library and segmenting the molecule based on its rotatable bonds. In a way, our segmentation is more like TRIC's way of separating a material system into sub-units although in our case it is done for many molecules in a condensed phase periodic boundary system. The authors did mention that Sella may add unnecessary constraints if multiple molecules exist in the system (forcing 'bond' between molecules), and in this situation Sella will switch to TRIC's method instead. One potential extension to our work in the future is to implement a molecule reaction analyzer which hierarchically re-segment molecules on the fly after chemical reaction has been detected in the system. This is currently not done in 3T-VASP.

The Sella optimizer approach has been demonstrated to work well compared to the default molecule optimizers used for different quantum chemistry software (Q-Chem, NWChem, etc) for molecules with up to 125 atoms. However, it does not seem like Sella is designed to work with periodic boundary condition system and large number of atoms like VASP (we welcome more feedback from the reviewer if this is incorrect, but it is the impression we get from the papers which cite the Sella optimizer). It does not seem realistic for us to utilize the Sella optimizer for the systems we have investigated in our work (perovskite surface, electrolyte liquid) because VASP is not among the software mentioned as being compatible with the Sella optimizer. Sella developers will be the ones best positioned to extend Sella to work with PBC code like VASP.

In this context, a Hessian-based approach does not seem to be appropriate for 3T, as 3T works in higher dimensional hyperspace instead of the lower-dimension internal coordinate space typically employed in TRIC and Sella. In addition to that, our primary target in this *ab-initio* reactor manuscript work is the VASP DFT software instead of quantum chemistry software like Gaussian and TeraChem. We routinely work with 237–325 atoms in the PBC box, and attempts to build a Hessian matrix on the fly for 3T systems (>1000 parameters are typical) may mean significant optimization step overhead compared to the actual

optimization steps we have now (in general 50-100 VASP DFT steps, or up to 150 steps, seem to be sufficient). The more commonly accepted approach in the neural network community when working in this high dimension parameter system and complex matrix operations is to leverage the computational efficiency of backpropagation autograd and the appropriate choice of parameter optimizer. In this case we choose Adam due to the many benefits it has in non-convex optimization problems (arXiv:1412.6980) including requiring almost no hyperparameter tuning. In fact, all three examples we have shown in our work (perovskite, reduction, oxidation) use the same PyTorch Adam optimizer hyperparameter (just learning rate = 0.03, the other hyperparameters are left at PyTorch Adam default values), which we first decided on based on our observations during the perovskite energy minimization experiment. While Adam optimizer only requires first-order gradient, it adapts individual adaptive learning rates for different parameters from estimates of first and second moments of the gradients.

In summary, we believe this is how our 3T approach differs from the internal coordinate methods TRIC and Sella:

1. TRIC and Sella are primarily designed for non-PBC molecular systems (up to 125 atoms in Sella paper quantum chemistry work, and up to significantly more atoms in TRIC classical force field work), while in this work 3T is primarily designed for larger-scale PBC condensed phase system (up to 325 atoms for 3T VASP work).
2. TRIC and Sella optimize parameters in lower dimension internal coordinates ($< 3n_{\text{atom}}$), while 3T optimize parameters in higher dimensions ($> 3n_{\text{atom}}$).
3. TRIC and Sella constraint some relative coordinates between atoms, while 3T does not impose any such limitation (we still allow individual atom translation).
4. TRIC seems to be a one-level structure minimization (sub-unit scale defined in the software) which is to an extent a subset of 3T multi-level structure minimization, while Sella does optimization based on constraints (which extends from the bond scale up to the dihedral scale).
5. TRIC and Sella attempts to build a partial/approximate Hessian matrix on the fly to accelerate optimization in the reduced parameter space, while 3T relies on machine learning community's Adam optimizer to accelerate optimization in the enlarged parameter space. Adam optimizer is widely used for large-parameter neural network optimization, and it adapts individual learning rates for different parameters from estimates of first and second moments of the gradients.
6. Sella can build new constraints on the fly (within limitations when multiple molecules are involved, see above), while TRIC and 3T are currently segmented in the beginning of the optimization process.

In general, we believe that the internal coordinate methods above are designed for non-PBC system and Sella is designed for usage with quantum chemistry software like NWChem, Q-Chem, etc. Our manuscript attempts to discover reactions within condensed phase and PBC box using VASP, as this work is about obtaining large-scale statistics of diverse chemical reactions. To the best of our understanding, the TRIC and Sella methods above do not work with our condensed phase PBC box system.

However, it is possible to convert our 3T-VASP algorithm to work with the non-PBC systems that TRIC and Sella usually work with. The prior work on 3T was performed in non-PBC protein-ligand system (but only with classical force field). Please see our response below to the reviewer's inquiry about 3T ASE implementation for more details. This is more along the line of creating 3T-NWChem, 3T-QChem, etc instead of our current ab-initio reactor 3T-VASP work. We do note that this typically smaller molecular system for geometry optimization is not what 3T-VASP has been designed for, so it may very well be the case that Sella optimizer's more Hessian-centric approach will work better compared to the Adam optimizer's gradient-based approach for these smaller molecular systems.

In our manuscript, we have added an extended discussion comparing 3T with the internal coordinate methods in the **Supplementary Information**, as we believe that TRIC and Sella cannot be directly used to study our systems of interest without extensive software modifications.

- Main text **page 4, line 6**. We added the sentence and citation to internal coordinate method references:

We also note that our work has some similarities to internal coordinate methods recently used in non-periodic quantum chemistry software for structure minimization,^[1-3] albeit with some extensive differences (see **Supplementary Information**).

- [1] Wang, L. P. & Song, C. Geometry optimization made simple with translation and rotation coordinates. *J. Chem. Phys.* **144**, (2016).
- [2] Shajan, A., Manathunga, M., Götz, A. W. & Merz, K. M. Geometry optimization: A comparison of different open-source geometry optimizers. *J. Chem. Theory Comput.* **19**, 7533–7541 (2023).
- [3] Hermes, E. D., Sargsyan, K., Najm, H. N. & Zádor, J. Sella, an open-source automation-friendly molecular saddle point optimizer. *J. Chem. Theory Comput.* **18**, 6974–6988 (2022).

- **Supplementary Information page 27, line 1**. We added the above extended discussion on 3T comparison with the internal coordinate methods such as TRIC and Sella.
- **Supplementary Information page 33, line 1**. We added the performance comparison between 3T-NWChem and Sella-NWChem optimizer using the ASE interface recommended by the reviewer.

Discussion on 3T Comparison with Sella for Single Molecule Optimization

In this section, we briefly discuss about the advantage of 3T optimizer compared to other state-of-the-art optimizers typically used for expensive quantum chemistry-based single molecule optimization, such as Sella. We develop a simplified version of the 3T optimizer which works with the ASE calculator interface (NWChem ASE calculator is demonstrated in this example). This is a direct comparison to the Sella optimizer which can work with NWChem ASE calculator. We then apply these optimizers on a TE4PBA cation structure freshly downloaded from PubChem (and hence the initial structure is energetically non-optimal). We optimize the geometry of TE4PBA using:

1. Sella optimizer utilizing NWChem
2. 3T optimizer utilizing NWChem
3. 3T optimizer using off-the-shelf organic force field (100 steps, negligible computation cost) followed by 3T utilizing NWChem

From the **Supplementary Figure 16** below, it is immediately clear that the 3T-based optimizer requires less NWChem calls to reach low-energy structure compared to the Sella optimizer, although all 3 methods reach the same optimized energy level and molecule structure at the end. 3T-FF-NWChem requires more NWChem calls than 3T-NWChem because 3T-FF initially generates a slightly less optimal initial conformation due to the usage of classical force field, but it still requires less NWChem calls than Sella-NWChem. No hyperparameter tuning is performed on the 3T optimizer, as we have simply used the default Adam optimizer learning rate (0.03) that we have used throughout the manuscript. We do note that we have not performed the rigorous infrastructure work which has been done by the Sella optimizer in their Python package (such as the i-PI socket protocol to directly communicate with the NWChem executable which is useful for computation overhead reduction), as NWChem single molecule quantum chemistry energy optimization has not been the focus of our manuscript. This infrastructure development, as well as large-scale molecule optimization benchmarking work, can be done in the future work for a 3T manuscript focusing on single molecule quantum chemistry energy optimization.

Supplementary Figure 16 | Comparison between Sella and 3T optimizers for single molecule energy optimization. In this example, we perform the energy optimization on the TE4PBA cation, which is a large and floppy organic cation consisting of 45 atoms. The ab-initio software being used by the optimizers is the quantum chemistry software NWChem. In the case of 3T-FF-NWChem, the structure is first optimized using 3T-FF for 100 steps (negligible computation cost) before starting 3T-NWChem energy minimization. 3T optimizers require less NWChem steps than the Sella optimizer to reach lower energy levels, although the final structures generated by Sella and 3T are equivalent in energy level and geometry.

4. *I enjoyed reading the paper and hope to better understand the novelty and value-add of the 3T approach after a revision by the authors.*

Sincerely,

Samuel Blau

1: <https://doi.org/10.1063/1.4952956>

2: <https://doi.org/10.1021/acs.jctc.3c00188>

3: <https://doi.org/10.1021/acs.jctc.2c00395>

We thank the reviewer for the feedback and for bringing the quantum chemistry molecular geometry optimization community's internal coordinate method to our attention. We hope that we have satisfactorily addressed the reviewer's concerns.

Reviewer #4 (Remarks on code availability):

The code seems to be fairly standard academic quality, regrettably without any sort of test suite that you would expect to see in a more mature codebase. However, my biggest criticism of the code is that it employs custom interfaces to external energy and force evaluations from a classical forcefield and VASP rather than using a more standardized interface like an ASE calculator. This both dramatically increased the work that the authors had to do while implementing the code and further substantially increases the amount of work that would be required for anyone who wants to extend the 3T approach to a different source of energy and forces e.g. an alternate quantum chemistry program. It will also admittedly make a direct comparison to Sella, like I advocate for in my review, much more difficult and time-consuming. However, if the authors do seek to make such a comparison, I would suggest they use the VASP calculator as implemented in QuAcc (<https://github.com/Quantum-Accelerators/quacc>) which further leverages the extensive VASP infrastructure (included automated error correction) implemented within the Materials Project software ecosystem.

We thank the reviewer for this feedback regarding an ASE calculator. These were design decisions we took primarily because of the first stage of this work, 3T-FF. As the reviewer might have noticed from our manuscript and reviewer responses, 3T-FF has its own important role in creating a physically decent initial structure for the subsequent 3T-VASP procedure. 3T-FF requires a classical force field to work, and as much as possible, we do not want to require our users to manually find/assign the classical force field styles and parameters for each new system. This is very tedious and undesirable for large-scale automation. We were able to find open-source classical FF parametrization webserver services for random organic molecules (SwissParam & LigParGen). However, these services effectively lock us into some specific force field styles as defined by those webserver. No existing ASE calculator exists for this combination of force field styles,

and it was much easier for us to manually write the entire force field calculator ourselves outside of the ASE framework.

However, we see good reasons to utilize the ASE calculator for other methods beyond 3T-FF. In our current code, the original 3T-VASP implementation is kept, as we know it to be bug-free and all the results in this manuscript have been produced using that function. However, it is straightforward to enable a calculation based on the ASE interface by small modification in `utils/potential_model_3T.py`. We have shown an example below, which will be added into the released version of our External_3T code github after the manuscript publication.

```
552     def create_ase_obj(self):
553         mass_dict = {1:'H', 7:'Li', 12:'C', 14:'N', 16:'O', 19:'F', 31:'P', 127:'I', 207:'Pb'}
554         atom_pos = self.atom_pos.detach().cpu().numpy()
555         cell = self.cell.detach().cpu().numpy()
556         mass = [round(i) for i in self.atom_mass[ self.atom_type ].detach().cpu().numpy().flatten().tolist()]
557         atom_type = [mass_dict[i] for i in mass]
558         ase_obj = ase.Atoms( atom_type, positions=atom_pos, cell=cell, pbc=[1,1,1])
559         return ase_obj
560
561     def forward(self):
562         # self.atom_pos needs to be updated because self.movable_pos is updated on each epoch
563         self.atom_pos = self.arrange_atom_pos(self.movable_pos_list, self.fixed_pos)
564
565         # create ase object out of current positions for usage with ASE calculators, if desired
566         ase_obj = self.create_ase_obj()
567
568         # Now we use available energy and forces from calculator (force field or DFT)
569         # Mind that these are numpy arrays, not pytorch tensors
570         if self.mode == 'FF':
571             #E_total, F_atoms = self.calc_E_F_forcefield()
572             E_total, F_atoms = calc_E_F_forcefield(self)
573         elif self.mode == 'VASP':
574             E_total, F_atoms = calc_E_F_VASP(self)
575         elif self.mode == 'PMMAT':
576             E_total, F_atoms = calc_E_F_PMMAT(self)
577         else: raise Exception('Unimplemented 3T mode')
578
579         # Finally we define a new output, the cost function C to be used in the backward chain rule.
580         # This output C only makes sense for chain rule purposes to optimize 3T model parameters.
581         F_atoms = torch.Tensor(F_atoms).to(self.device)
582         C_total = -torch.sum( self.atom_pos * F_atoms )
583
584         return E_total, C_total
```

We apologize for the hardcoded `mass_dict` in the first line of the `create_ase_obj()` function. Because 3T-FF utilizes many force field file formats one way or another in the background (CHARMM, GROMACS, and LAMMPS), the remaining information in the model which can be used for identifying atom elements end up being just the atomic masses (LAMMPS format does not allow the inclusion of atom elements, it only allows the inclusion of atom types (consecutive integer numbers) and masses).

We recognize that the ASE modification above may still be inconvenient for others who want to use the 3T method for just single molecule geometry optimization tasks, which can be done efficiently using the Sella optimizer. This is not what 3T-VASP is designed for. Our code is designed as an ab-initio reactor box for many molecule/solid surface system, and in a sense is much more similar to the AIMD work performed by Martinez *et al* (*Nature Chem.* 6, 1044 (2014)).

If a user is more interested in using our work as a standalone optimizer code for single molecule geometry optimization, a form more similar to that of the Sella molecule optimizer may be more desirable. We have included a significantly less complex codebase for just minimizing single molecules (xyz input file) using 3T,

similar to the typical usage of the Sella optimizer. The usage example is shown below. We would like to repeat our assertions above that our work is not designed for quantum chemistry molecule geometry optimization, and further infrastructure work will be necessary to make the overall experience of 3T-NWChem as attractive as the Sella-NWChem optimizer for this new task. We believe that it is beyond the scope of this electrochemical *ab-initio* reactor manuscript, and we welcome the academic community to use/extend our simplified 3T optimizer (with the ASE interface implemented) as they see fit if it is beneficial for their molecule geometry optimization work. This code base (Simple_3T) is included as an additional peer review material and as a sub-folder of External_3T github upon manuscript publication. We think that this new 3T ASE-compatible molecule optimizer is too preliminary to justify an individual release. It should instead be properly designed and compared against the Sella optimizer in a separate publication targeting the large-scale molecule geometry optimization task, which is very different from the electrochemical *ab-initio* reactor focus of this manuscript.

```
1 import ase.io as sio
2 from ase.calculators.nwchem import NWChem
3 from simple3T import Simple3T
4
5 use_pbc = False
6 atoms = sio.read('TE4PBA.xyz')
7 if use_pbc:
8     atoms.pbc = [True, True, True]
9     atoms.cell = np.array([20.0, 20.0, 20.0])
10
11 atoms.calc = NWChem()
12 opt = Simple3T(atoms, id='cache1')
13
14 # Run using automatically assigned force field for 3 cycles, 100 steps each
15 opt.run(100, use_FF=True, out_tag='3T_FF_0')
16 opt.run(100, use_FF=True, out_tag='3T_FF_1')
17 opt.run(100, use_FF=True, out_tag='3T_FF_2')
18
19 # Run using the attached ASE calculator for 1 cycles, 50 steps
20 opt.run(50, use_FF=False, out_tag='3T_ASE_0')
```

Reviewer #1

I would like to thank the authors for their thorough response to my comments on the previous version of the manuscript.

We thank the reviewer for this positive comment and for the thorough feedback on our various manuscript drafts, which has strengthened our manuscript considerably.

Comment 1:

Regarding my previous comment 2, I feel that it has not been sufficiently addressed. While the discussion has been somewhat extended by mentioning a few automated reaction mechanism discovery methods, I believe that these methods should also be represented in Figure 1, where standard relaxation, molecular dynamics and 3T relaxation are compared.

We thank the reviewer for this comment. We have now added an additional method summary in **Figure 1a** (Biased Molecular Dynamics). We have cited quite several additional methods in the manuscript introduction, and methods such as AFIR, growing string, and metadynamics essentially add additional terms in the energy / forces of the system dynamics to bias the system into reaching certain desired states more easily. Therefore, we think it is more informative to the reader to group them into our representation in the Biased Molecular Dynamics row in **Figure 1a**, as this shows the high-level distinction between the different types of methods in the table without going into the specific implementation details of the different biasing mechanisms.

- Main text **Figure 1a** has been modified:

Figure 1 | High-level description of multi-scale 3T relaxation procedure and its application. a) Structure update procedure difference between typical structure energy relaxation, molecular dynamics-based structure exploration, and multi-scale 3T structure energy relaxation. **b)** 3T multi-scale structure transformation mode sequence used in this work. **c)** 3T multi-scale structure transformation projects the energy minimization problem into higher dimension 3T parameter hyperspace, where it is possible to explore farther local energy minima landscape difficult to access by standard relaxation method or longer to reach using molecular dynamics. **d)** 3T ab-initio applications demonstrated in this work, covering complex floppy organic cation binding on perovskite surface defect site and electrochemical redox reactions within liquid Li-ion battery electrolytes. **e)** Some of the reduction and oxidation electrochemical reaction byproducts generated from the bulk electrolyte liquid mixture in this work.

Comment 2:

The authors claim that the growing string method requires both reactant and product to be known in advance, but the single-ended growing string method (which is cited) only requires a single intermediate to be known (for the double-ended growing string method, both reactant and product are needed).

We thank the reviewer for this clarification. We have now fixed this misconception in the manuscript introduction.

- Main text **page 4, line 1**. We modified the sentence:

When the qualitative characteristics of the intended reactions are already known in advance, reaction transition states can be found using different methods, for example by using the single-ended

growing string method which sequentially places intermediate nodes in the delocalized internal coordinates (bonds, angles, or torsions) relevant for reaching the desired reactive outcome.

Comment 3:

I would not say that AutoMeKin's BXDE ("a combination of temperature-dependent metadynamics and transition state analysis") is "routinely" performed, since it is to the best of my knowledge not very commonly used.

We thank the reviewer for this clarification. We have now fixed this misconception in the manuscript introduction. Please refer to our response to **point #4**.

Comment 4:

Technically, I think that the description "temperature-dependent metadynamics" is inaccurate. Both metadynamics and boxed dynamics use a bias, but these are different — in MTD through collective variables and in BXDE through the potential energy.

We thank the reviewer for this clarification.

- Main text **page 4, line 5**. We modified the sentence:
A combination of temperature-dependent dynamics approach with vibrational energy added into the system for generating reaction and subsequent transition state analysis can also be performed.

Comment 5:

Regarding my previous comment 5, the authors added to the Methods section an explanation that for example a molecule with seven atom indices can be segmented into three micro groups $m = [0,2,6], [1,3,4], [5]$. However, it is not clear to me how exactly such a molecule is segmented. Could the authors perhaps include a graphic of the molecule including the atom index numbering to visualize this?

We thank the reviewer for this clarification request. To further aid with clarifying this segmentation indexing example, we have added an indexed formic anhydride molecule, where the micro-groups are separated by rotatable bonds.

- **Methods** section **page 27, line 10**. We extended the paragraph and added an indexed segmented molecule figure:

For example, an entire molecule with 7 atom indices $[0,1,2,3,4,5,6]$ may be segmented into micro-group $m = [0,2,6], [1,3,4],$ and $[5]$ and macro-group $M = [0,1,2,3,4,5,6]$. For computational convenience

reasons, we encode M as a nested list of m instead ($[[[0,2,6],[1,3,4],[5]]]$). Segmentation of an example molecule (formic anhydride) is illustrated below.

Figure 5 | Schematic diagram of hierarchical segmentation of micro-group and macro-group structures.

Reviewer #1 (Remarks on code availability):

I have not done any further review of the code than mentioned in the previous round of revision.

We thank the reviewer for the time spent on reviewing our manuscript code.

Reviewer #2

1. *In my first review, my general comment was that the paper is lacking focus on some the important aspects of interest to the community, while diverging on less important aspects such as implementation details, relies too much on information found in the supplementary materials, precisely because the authors are trying to cover too much materials. While the authors have made some efforts to improve the manuscript, I am still feeling the same about this manuscript in general. And I do not think this manuscript is well suited for Nature Communications.*

We thank the reviewer for this feedback. We will do our best to address the reviewer's remaining concerns.

2. *For instance, to my first comment about how complicated it seems to apply methodology, the authors respond that the software is easy to use. A scientific manuscript however, in my opinion, should not consider the software as a black box and let the reader guess what happens inside the box. A scientific report should describe the methodology and ideas so that a reader could reproduce the research and write the same software on their own. The authors still fail at that task in my opinion in this revised manuscript.*

We thank the reviewer for this important feedback. Please refer to our reply to **point #4**, as we attempt to be more descriptive and explicit about the 3T structure transformation mechanisms in the context of this 3T-VASP manuscript.

3. *I see this manuscript as describing the application of 3T algorithm previously proposed by the same authors to two applications, with description of results. But no added understanding to what the algorithm is doing for these applications. For instance, there is no details about the choice of the theta variables for these applications. In the 3T original paper, there was a mention of energy "kicks" and their values. I don't see nay mention of that in the present manuscript.*

We thank the reviewer for this feedback. While the underlying ideas are the same (fast energy minimization achieved through multi-scale structure transformation), the code and implementation between the two papers are completely different. It is like how there are many different types of molecular dynamics, the types of '3T' implemented in the two papers are very different. Here are some of the differences:

- a. The original 3T was designed specifically for attaching two systems of very different scale: a protein structure with thousands of atoms, and a ligand molecule with at most tens of atoms. Hence, the kind of 3T structure transformation modes which make physical sense are also very different. For example, while it makes physical sense to rotate a small molecule around its center, it makes no physical sense to rotate the entire large protein structure. It makes more sense to look at the individual protein amino acids and assign 3T transformation modes on them. In the present work,

- no such complication arises. We use the same transformation modes for all the micro & macro-groups in the system because the molecules are all relatively small compared to proteins.
- b. The original 3T was only capable of using classical force field calculated using PyTorch, which severely limits its capabilities. The present work comes up with a crucial computation trick which enables any external force/energy calculator to be integrated with 3T (External_3T). The key enabler (development of chain-rule-based cost function $C = - \sum \vec{r}_i \cdot \vec{r}_i$, which can be auto-differentiated by PyTorch) is just an implementation detail, but it lets the 3T algorithm to be relatively software-agnostic (we have so far demonstrated 3T-FF, 3T-VASP, and 3T-NWChem in this manuscript although 3T-NWChem is mostly just a peer review prototype).
 - c. The original 3T paper deals with protein structure in vacuum. This structure has such large energy barriers that ‘vanilla’ 3T energy minimization cannot overcome those barriers to explore possible structure conformations. Consequently, a form of small energetic kick (we put small random initial values on the 3T transformation parameters θ , as 3T mechanism cannot directly control system atom coordinates / total energy) is needed to enable diverse structure exploration. In the present work on 3T-VASP systems, no such energetic kick is needed/used at all because the energy barriers can be overcome by just ‘vanilla’ 3T minimization and hence it is not mentioned in the manuscript.

We agree with the reviewer that efforts can be made to more clearly and visually define what have specifically been done in this manuscript’s 3T workflow, as it is quite different from our previous work. Please see our response to **point #4** below.

4. *Another point that I missed in my first review: the authors refer a lot to PyTorch functionalities. PyTorch should be an implementation “detail”, not used in general presentation. Mathematical functions should be used in manuscript.*

We thank the reviewer for this important feedback. After further discussing how to address **points #2 and #3** above, we have now added several modifications to the manuscript. First, the visualization of the exact 3T transformation modes done in this manuscript (the original 3T transformation modes from the protein work is not discussed because it is for a completely different class of material system) are added into a new subplot in **Figure 1 (Figure 1b)**. Next, we have also added explicit high-level equations in sequence, which detail how the various 3T θ parameters are used (in combination with **Figure 1b**). The exact full low-level equations remain in the **Supplementary Information** to prevent cluttering the manuscript, but they will be referred to in the main text. Finally, we also add the mathematical representation which describes how to update θ over subsequent 3T steps. It is only after these general presentations, that we will mention the practical implementation detail using PyTorch. We believe it is still necessary to mention this implementation detail because it is not practical/efficient to implement the complexity of 3T multi-level gradient update capabilities without the aid of flexible modern auto-differentiation tools available today such as PyTorch or TensorFlow (ver 2.0 or higher, as not even TensorFlow v1 will work for this purpose).

- Main text **Figure 1b** has been added:

Figure 1 | High-level description of multi-scale 3T relaxation procedure and its application. a) Structure update procedure difference between typical structure energy relaxation, molecular dynamics-based structure exploration, and multi-scale 3T structure energy relaxation. **b)** 3T multi-scale structure transformation mode sequence used in this work. **c)** 3T multi-scale structure transformation projects the energy minimization problem into higher dimension 3T parameter hyperspace, where it is possible to explore farther local energy minima landscape difficult to access by standard relaxation method or longer to reach using molecular dynamics. **d)** 3T ab-initio applications demonstrated in this work, covering complex floppy organic cation binding on perovskite surface defect site and electrochemical redox reactions within liquid Li-ion battery electrolytes. **e)** Some of the reduction and oxidation electrochemical reaction byproducts generated from the bulk electrolyte liquid mixture in this work.

- Main text **page 9, line 9**. We modified the paragraphs:

We set our 3T function as a sequential (output from one transformation is input to the next transformation) and hierarchical (first atom level, then micro-group level, and finally macro-group

level) compound geometry transformation of these six functions, each governed by the parameter at step (**Figure 1b** & **Figure 2c**). In mathematical terms, we have:

$$\vec{r}_t = 3T(\vec{\theta}_t, \vec{r}_0)$$

$$\Delta \theta_{j,t} = -k_j \frac{\partial E_{pot}(\vec{r}_t)}{\partial \theta_{j,t}}$$

$$\Delta \vec{r}_{i,t} = -k_i \frac{\partial E_{pot}(\vec{r}_t)}{\partial \vec{r}_{i,t}}$$

$$\Delta \vec{v}_{i,t} = -\frac{\Delta t}{m_i} \frac{\partial E_{pot}(\vec{r}_t)}{\partial \vec{r}_{i,t}}$$

$$\Delta \vec{r}_{i,t} = \Delta t \vec{v}_{i,t}$$

$$\vec{r}_i = (\vec{r}_{i,1}, \vec{r}_{i,2})$$

$$\vec{r}_i = (\vec{r}_{i,1}, \vec{r}_{i,2})$$

with \vec{r}_i representing the intermediate coordinates of all atoms in the system between 3T structure transformation modes (see **Supplementary Information** for each equation details). Initially, we have $\vec{r}_i = \vec{r}_i$ and hence the initial transformations are identity functions. The resulting geometry is evaluated using an external software to calculate the potential energy $E = E(\vec{r})$ and atomic forces $\vec{F} = -\nabla E(\vec{r})$ of the system. To determine the \vec{r}_i parameter update, we should compute the gradient of E with respect to each of the \vec{r}_i above. The gradients can then be used to determine the updated $\vec{r}_i = \vec{r}_i + \Delta \vec{r}_i$, in combination with the gradient \vec{F} generated by any suitable optimization algorithm. To simplify notation, we will first flatten the \vec{r}_i where i indicates its flattened indices:

$$\vec{r} = ([r_{1,1}, r_{1,2}, r_{2,1}, r_{2,2}, \dots, r_{N,1}, r_{N,2}])$$

$$\vec{F} = [F_{1,1}, \dots, F_{N,2}]$$

$$r_{i+1} = r_{i,2}$$

While the last two steps above are very cumbersome to implement manually, in practice this looks just like a neural network parameter update. The differentiation can be automated using PyTorch autograd function, while the parameter update can be automated using any of PyTorch built-in optimizers. In our previous work, E was the minimization cost function and the evaluation is fully performed in PyTorch, making \vec{r}_i gradient-based update on each step straightforward: $\vec{r}_i = \vec{r}_i + \Delta \vec{r}_i$. In this work, the usage of external energy calculator such as classical force field or DFT software VASP requires us to develop a chain-rule-based cost function $E = -\sum_i \vec{F}_i \cdot \vec{r}_i$, with i representing atom indices. Performing $\vec{F} = -\nabla E$ will store identical PyTorch gradients to \vec{F} as if the whole $\vec{F} = -\nabla E$ calculation is performed inside PyTorch instead (\vec{r}_i needs to be treated as gradientless constants, see **Methods**). Afterward, we simply use PyTorch Adam optimizer to calculate $\vec{r}_i = \vec{r}_i + \Delta \vec{r}_i$, in a manner identical to any standard neural network training using PyTorch to minimize a loss function.

- We modified the **Supplementary Information** notations to better match the updated main text:

3T Structure Transformation Modes

In the main text, we have described 6 structure transformation modes utilized in our work: T_1 , T_2 , T_3 , T_4 , T_5 , and T_6 , and T_1 , T_2 , and T_3 are just simple Cartesian coordinate translation functions:

$$\vec{r}_i = \vec{r}_i + \vec{t}_i$$

$$\vec{r}_i = \{ \vec{r}_i + \vec{t}_i \} \quad \text{if } i \in \text{micro-} \\ \text{group}$$

$$\begin{aligned}
 \vec{r}_{ij} &= \vec{r}_{ij} + \vec{t}_{ij} & \text{if } i \in \text{macro-group} \\
 & & \text{otherwise}
 \end{aligned}$$

with i, j , and k representing individual atom, micro-group, and macro-group indices respectively. The shapes of these parameters are 3×3 (for \vec{t}_{ij}), 3×3 (for \vec{t}_{jk}), and 3×3 (for \vec{t}_{ik}) with n, m , and k representing the total number of atoms, micro-groups, and macro-groups.

\vec{r}_{ij} is a micro-group sidechain rotation function. In this work, we only allow this transformation to be done on micro-group which is connected to only one other micro-group through a single rotatable bond. Suppose that an atom B in a sidechain micro-group is connected to an atom A in a neighboring micro-group through a rotatable bond. For every atom i (with coordinates \vec{r}_{ij}) within the sidechain micro-group, we perform the following A-B axis rotation:

$$\begin{aligned}
 \vec{r}_{ij} &= \vec{r}_{ij} - \vec{t}_{ij} \\
 \vec{r}_{ij} &= \vec{r}_{ij} + \vec{t}_{ij} \\
 \vec{r}_{ij} &= \vec{r}_{ij} - \vec{t}_{ij} \\
 \vec{r}_{ij} &= \vec{r}_{ij} + \vec{t}_{ij} \\
 \vec{r}_{ij} &= \vec{r}_{ij} - \vec{t}_{ij} \\
 \vec{r}_{ij} &= \vec{r}_{ij} + \vec{t}_{ij} \times \vec{t}_{ij} \\
 \vec{r}_{ij} &= \vec{r}_{ij} + \vec{t}_{ij} \cos \theta_{ij} + \vec{t}_{ij} \sin \theta_{ij} & \text{if } i \in \text{micro-group} \\
 & & \text{otherwise}
 \end{aligned}$$

The shape of the parameter is 3×1 , with n representing the total number of sidechain micro-groups with just one rotatable bond connection.

\vec{r}_{ij} and \vec{r}_{jk} are three-axis rotation functions centered on either the micro-group or the macro-group centers. For every atom i within the micro-group j (or atom i within macro-group k), we perform:

$$\begin{aligned}
 & \begin{bmatrix} \cos \theta_{j,0} & -\sin \theta_{j,0} & 0 \\ \sin \theta_{j,0} & \cos \theta_{j,0} & 0 \\ 0 & 0 & 1 \end{bmatrix} \\
 & \begin{bmatrix} \cos \theta_{j,1} & 0 & \sin \theta_{j,1} \\ 0 & 1 & 0 \\ -\sin \theta_{j,1} & 0 & \cos \theta_{j,1} \end{bmatrix} \\
 & \begin{bmatrix} 0 & \cos \theta_{j,2} & -\sin \theta_{j,2} \\ 0 & \sin \theta_{j,2} & \cos \theta_{j,2} \end{bmatrix} \\
 \vec{r}_{ij} &= 1 \sum \vec{r}_{ij} & \text{for } i \in \text{micro-group} \\
 \vec{r}_{ij} &= \vec{r}_{ij} + \vec{t}_{ij} & \text{if } i \in \text{micro-group} \\
 & & \text{otherwise}
 \end{aligned}$$

The equations above are written for rotation of micro-group j . For rotation of micro-group k , simply substitute m with M and micro-group j with macro-group k . The shapes of these parameters

are 3×3 (for \vec{t}_{ij}) and 3×3 (for \vec{t}_{jk}) respectively.

5. *As I reread the manuscript and the responses to the review comments, I also realized the following: to minimize DFT calls, FF can be run for a while, before DFT calls are used to “refine” solution. That would be the way to appropriately evaluate the effect of the 3T multiscale approach (in figure 2 for instance).*

We thank the reviewer for this feedback. We have implemented this feedback, where the cation is now initially minimized for 5×200 FF steps (but only in a per-atom manner, without 3T). We then run this perovskite-cation system through the standard VASP minimization procedure. When this is done, the cation is now also deeply embedded into the defect site, just like 3T-VASP relaxation. However, it still requires significantly more DFT calls than 3T-VASP to achieve this performance.

Unlike the ‘only VASP’ and ‘CREST+VASP’ pipeline which can utilize off-the-shelf tools to produce the results in the updated **Figure 2f**, the ‘FF+VASP’ pipeline requires using our FF assignment pipeline which is not currently an off-the-shelf tool and may require significant effort for others to re-implement. Because of this, we have kept both the original ‘only VASP’ and the new ‘FF+VASP’ results in **Figure 2f** to allow the reader to compare the efforts needed to achieve the two different results.

We have updated the main text, **Figure 2**, and added **Supplementary Figure 15** correspondingly.

As for our previous comparison of electrolyte system between 3T-VASP and VASP AIMD in the **Supplementary Figure 13**, we have already run 3T-FF to disperse the electrolyte in the DFT box, prior to running AIMD on the resulting structure to explore possible electrochemical reactions.

- Main text **page 11, line 10**. We added the sentences:

We note that if we first utilize the force-field pipeline we have developed to relax the structure (just per-atom relaxation, without 3T) prior to running standard VASP relaxation (FF+VASP in **Figure 2f**), this baseline performance can be considerably improved. This improved baseline will also be able to generate low energy structures with the cation being deeply embedded into the defect site (**Figure S15**), equivalent to those generated by 3T-VASP, albeit using significantly more (300-350) DFT calls.

- We modified main text **Figure 2f**:

Figure 2 | 3T relaxation of floppy organic cation on FAPbI₃ surface vacancy defect site. **a)** Typical device structure of FAPbI₃ perovskite solar cell deposited on front transparent conducting oxide (TCO) glass sandwiched by electron and hole transport layers (ETL, HTL), with surface passivation molecules being applied on the bottom surface of the perovskite. **b)** 3T multi-scale structure transformation modes enabled for TE4PBA organic cation during ab-initio 3T relaxation. **c)** 3T optimization implementation utilizing PyTorch for gradient propagation and external calculator for potential energy and atomic forces calculation (see main text

for symbol details). During the 3T-FF, the computation workflow is FF→3T→FF→3T→... and during the 3T-VASP it is VASP→3T→VASP→3T→... FF and VASP are only atomic force/energy calculator of static atom frames, and only 3T is allowed to optimize the atomic structure in the PyTorch framework. Final TE4PBA-FAPbI₃ structures obtained using **d)** standard VASP relaxation and **e)** 3T-VASP relaxation, with the cation originally floating above the defect site. In the case of **d)** the cation still floats above the defect site, while in the case of **e)** the cation embeds itself deep into the FAPbI₃ surface vacancy defect site. **f)** DFT binding energy vs the number of DFT call comparison between standard VASP, CREST+VASP, FF+VASP, and 3T-FF+3T-VASP relaxations. Whereas standard relaxation fails to find a cation binding pose on FAPbI₃ surface even after 500 DFT steps, 3T relaxation consistently finds deep defect binding pose with significantly lower DFT binding energy within 50 DFT steps. CREST+VASP can achieve lower energies than standard VASP relaxation, but the cations are still not deeply embedded into the perovskite defect site in addition to still taking significantly more DFT steps to converge compared to 3T-VASP. FF+VASP can achieve similar energies and structures to 3T-VASP relaxation but requires significantly more DFT steps than 3T-VASP.

- We added new section in the **Supplementary Information**, as well as **Supplementary Figure 15**:

Additional Perovskite Surface Cation Passivation Baseline Comparison using FF + VASP Relaxation

It is also possible to utilize our force field pipeline to generate a better baseline for the VASP relaxation without using 3T. We first take the 3 initial structures we previously utilized for standard DFT relaxation and run them through 5×200 steps of FF relaxation steps. This is done using our PyTorch optimization pipeline while turning off the 3T capabilities (this becomes a standard per-atom relaxation, and not a 3T multi-scale relaxation). The resulting structure can then be minimized using VASP. The result using this improved baseline approach is promising, with the TE4PBA cation becoming deeply embedded into the defect site, with energy levels equivalent to that produced using 3T-VASP. However, the FF+VASP approach still requires significantly more DFT steps compared to 3T-VASP (main text **Figure 2f**). The resulting FF+VASP relaxation structures are shown in **Supplementary Figure 15** below.

Supplementary Figure 15 | TE4PBA cation passivation structure on FAPbI₃ defect site after we use our FF per-atom relaxation followed by VASP relaxation. These cations are deeply embedded into the defect site just like the 3T-VASP relaxation version, although they still require significantly more DFT steps than 3T-VASP relaxation.

Reviewer #3

- 1. The manuscript has been improved considerably by including a better relation to existing methods, demonstrating the improvements of the new method, and providing a clearer presentation. Therefore, I think it is publishable now. The manuscript is, at the same time quite technical. This is probably necessary due to the complexity of the topic. While I expect that the methods will be used by others and the work will receive attention, I am not sure if a more technical journal may be better for this work and leave this decision to the editor.*

We thank the reviewer for this positive feedback.

Reviewer #4

1. *I greatly appreciate all of the work that the authors did in response to my review. Indeed, they have gone above and beyond what I would have expected in their construction of Fig S16 and their implementation of Simple3T. I find both very interesting/useful - thank you for the time that was put into this! Hopefully the authors found the extra work to be instructive and worthwhile.*

We thank the reviewer for this positive comment.

2. *However, the authors have drawn key incorrect conclusions regarding the applicability of Sella, and thus they have not done the most important thing I would like to see, which is a direct comparison of 3T-VASP with Sella (employing the VASP ASE calculator) on the electrolyte example. Admittedly, there is currently an issue with the translation-rotation internal coordinates in Sella (see my open issue here: <https://github.com/zadorlab/sella/issues/32>), but even still, I think a direct comparison with Sella as it currently functions is crucial to establishing the utility of the work. Assuming that 3T yields the reaction products reported in the manuscript in substantially fewer DFT calls than the state-of-the-art internal coordinate optimizer, then I fully support publication of this manuscript in Nature Communications. If instead Sella is just as efficient and effective as 3T or requires even fewer DFT calls to yield the observed reaction products, then I question the value of the approach, despite its ingenuity.*

We thank the reviewer for this feedback. Please refer to our response to **point #4**.

3. *Let me clarify some of the misconceptions, and then I will be very clear about what I am asking for and how I would suggest carrying it out. Referring to the author's numbered points on "how our 3T approach differs from the internal coordinate methods TRIC and Sella":*
 1. *I can't speak for the GeomeTRIC code, but Sella works with periodic boundary conditions. The internal coordinates automatically wrap across the periodic boundaries via the minimum image convention. Further, there is no inherent limit to the number of atoms in a system being optimized with Sella. 325 atoms in a periodic box using VASP should be no problem at all.*
 2. *Minor point of clarification here - people do often employ $>3N$ "redundant" internal coordinates. I'm not sure if Sella always sticks to the usual $3N-6$ internal coordinates, but just for the authors' information - this comes back to a decades old debate about whether the absolute fewest number of coordinates is best for optimization, or if having more coordinates*

- which effectively allow you to "cheat" through a higher dimensional space is better. David Baker was the historical proponent of $3N-6$ aka "delocalized" internal coordinates while Bernard Schlegel was the historical proponent of $>3N-6$ aka "redundant" internal coordinates. This information does not impact the manuscript - I just wanted to provide historical context.*
- 3. I again cannot speak for GeomeTRIC, but while Sella contains the functionality to impose all sorts of explicit constraints, as chemists often like to do when performing bespoke transition state optimization, no constraints are automatically defined, and constraints are in no way central to Sella's default function.*
 - 4. I agree that the multi-level aspect of 3T is novel.*
 - 5. I agree. However, it remains unclear if the ML-centric advantages of Adam are actually more valuable than the quasi-second-order machinery of Sella (which molecular geometry optimizers have employed in some form or another for decades) in the context of molecular optimization.*
 - 6. The way this is worded makes me worried that there is confusion between "constraints" and "coordinates". Sella can add dummy atoms on-the-fly in order to add new internal coordinates that effectively replace a previous coordinate that has gone invalid - i.e. a three-atom angle going to 180 degrees, or a dihedral angle going to 0 or 180 degrees. (Side note - the fact that 3T effectively employs a flavor of "internal coordinates" which never go invalid is a point in its favor that could probably be mentioned somewhere in the manuscript.) Internal coordinates going invalid is an issue that plagues other molecular geometry optimizers, but is alleviated in Sella. But any actual constraints - e.g. fixing a specific bond length or angle - defined by the user when employing Sella do not change over the course of the optimization. Back to coordinates - while Sella can dynamically add dummy atoms and modify the internal coordinates to alleviate issues with coordinate validity, it does not seek to regularly re-define the internal coordinates during an optimization trajectory. Thus, I believe that re-starting an optimization from a previously optimized structure, as done with 3T, will re-define the internal coordinates and perhaps open up new avenues for further optimization.*

We thank the reviewer for clarifying some of our misconceptions about the Sella optimizer. We have now made the following modifications in our Supplementary Information comparing 3T with TRIC and Sella.

- **Supplementary Information page 30, line 7.** We modified the paragraphs:

We next discuss the comparison between 3T and the Sella optimizer. The Sella optimizer is interesting because it uses a different approach; it works in the basis of redundant internal coordinates (bonds, angles, and dihedrals extracted from the system, in addition to the dummy atoms added to mitigate 'invalid angle' problems which commonly plague these internal coordinate methods). Constraints are added to avoid an increase in the dimensionality of the optimization problem due to the addition of dummy atoms (additional constraints may be added based on Sella user's chemical intuition), before a Hessian matrix for the system is approximated by iteratively diagonalizing it in the basis of the redundant internal coordinates. In this sense, the Sella optimizer is more like the TRIC method and unlike the 3T method which works in a higher dimension parameter space. 3T also does

→
not project the system into a set of internal coordinates, as it simply uses additional 3T parameters to geometrically transform the atom positions in the original Cartesian coordinate space (and consequently by construction the 3T optimization system cannot go 'invalid').

In addition to that, the Sella optimizer can automatically determine the molecule connectivity graph, analyse if an angle is about to go invalid, and add dummy atoms to mitigate this problem on the fly. In contrast, the 3T-VASP as it is currently designed automatically determines its hierarchical segmentation by processing the individual input molecules using RDKit cheminformatics library and segmenting the molecule based on its rotatable bonds. In a way, our segmentation is more like TRIC's way of separating a material system into sub-units although in our case it is done for many molecules in a condensed phase periodic boundary system. The authors did mention that Sella may add unnecessary constraints if multiple molecules exist in the system (forcing 'bond' between molecules), and in this situation Sella will switch to TRIC's method instead. One potential extension to our work in the future is to implement a molecule reaction analyzer which hierarchically re-segment molecules on the fly after chemical reaction has been detected in the system. This is currently not done in 3T-VASP.

The Sella optimizer approach has been demonstrated to work well compared to the default molecule optimizers used for different quantum chemistry software (Q-Chem, NWChem, etc) for molecules with up to 125 atoms. During the course of the peer review, we have been made aware that the Sella optimizer also works with periodic boundary condition software such as VASP and Quantum Espresso, and in this aspect the Sella and 3T approaches are similar, even if the underlying math are different.

- **Supplementary Information page 32, line 4.** We modified the paragraphs:

In summary, we believe this is how our 3T approach differs from the internal coordinate methods TRIC and Sella:

1. TRIC and Sella optimize parameters in lower dimension internal coordinates ($< 3n_{\text{atom}}$), while 3T optimize parameters in higher dimensions ($> 3n_{\text{atom}}$).
2. TRIC constrains some relative coordinates between atoms (it is also an option available for Sella users), while 3T does not impose any such limitation (we still allow individual atom translation).
3. TRIC seems to be a single-level structure minimization (sub-unit scale defined in the software) which is to an extent a subset of 3T multi-level structure minimization, while Sella does single-level structure minimization in the redundant internal coordinate space (which extends from the bond scale up to the dihedral scale).
4. TRIC and Sella attempts to build a partial/approximate Hessian matrix on the fly to accelerate optimization in the reduced parameter space, while 3T relies on machine learning community's Adam optimizer to accelerate optimization in the enlarged parameter space. Adam optimizer is widely used for large-parameter neural network optimization.
5. TRIC, Sella, and 3T define the system parameter space in advance (internal coordinates for TRIC and Sella, and hierarchical transformation parameters for 3T), although Sella may also update its internal coordinates on the fly to facilitate the addition of new dummy atoms which is used to replace a previous internal coordinate which has gone invalid.
6. By construction, 3T parameter space does not go invalid because they are simply parameters which transform atom coordinates in the Cartesian space, unlike the internal coordinate methods where the Cartesian coordinates may go invalid when projected into the internal coordinate space (such as angles going to 0 or 180 degrees).

4. *With those misconceptions clarified, let me return to my ask. I am sorry to ask for even more work from the authors when they have already done so much, but as I stated previously, I think this is critical: please run Sella atop the VASP ASE calculator on one of your electrolyte boxes and compare to 3T! I do not think you should need to do anything fancy - just use the standard ASE VASP calculator. For Sella, be sure to set order=0 (i.e. you're trying to optimize to a minimum, not a saddle point) and internal=True (i.e. you want to use internal coordinates). Please run a sequence of five iterations of 50 steps each starting from the geometry obtained from 3T-FF. As I stated above, stopping Sella after 50 iterations and restarting it fresh will re-define the internal coordinates and perhaps be beneficial in the same way that it is for 3T. The computational cost of this experiment should be equivalent to the 3T-VASP experiment, and thus I hope it will not be burdensome. I badly want to know the outcome of this experiment - while I do not expect you to run into any serious issues, if you do, I would be happy to help, so feel free to email me at smlau@lbl.gov.*

Sincerely,

Sam Blau

We thank the reviewer for this feedback. We have now added an extended discussion comparing Sella-VASP and 3T-VASP for both the electrolyte reduction and oxidation experiments. While 3T-VASP significantly reduces the computational cost of each trajectory, we ran many trajectories in this manuscript (63 reduction and 63 oxidation trajectories). Running all these post-3T-FF trajectories using Sella-VASP still requires a large cloud computing budget that our department cannot justify. We have instead run Sella-VASP on randomly chosen reduction and oxidation trajectories post-3T-FF (3 of each) and compare them to the corresponding 3T-VASP structure. Unlike the 3T-NWChem single molecule structure optimization task where 3T-NWChem simply reach the same energy with Sella-NWChem but faster, 3T-VASP consistently minimizes the bulk electrolyte energy to lower levels than Sella-VASP faster because Sella-VASP is stuck at higher local energy minimums. Based on the result, we conclude that 3T-VASP generates more physical reaction results (with lower structure energies) faster than Sella-VASP.

We do worry about the Sella-VASP result for electrochemical reduction experiment, as it generates very large energy spikes that we have not seen with the Sella optimizer before. This undesired behavior is not observed for the electrochemical oxidation experiment (we use the same Sella input commands). We are unfortunately not the ones best placed to thoroughly investigate the root cause of this undesired Sella-VASP energy spike behavior. Will the reviewer be able to communicate this observed Sella-VASP problem to the Sella team?

- **Supplementary information page 35, line 1.** We added the section:

Discussion on 3T Comparison with Sella for Bulk Electrolyte Electrochemical Reactions

In this final section, we compare the performance of 3T with the Sella optimizer for the primary task explored in this manuscript, which is the exploration of electrochemical reaction in bulk electrolyte. We attempt to understand the difference between 3T-VASP results compared to when Sella-VASP is utilized on the same system, under default Sella settings and identical cycle/step counts. We want to make sure that the Sella optimizer has access to the same kind of physically meaningful initial structure which has been produced by 3T-FF (so that Sella does not have to waste iteration steps dispersing molecules in the VASP periodic boundary condition box and can focus on the electrochemical reactions instead). Consequently, we take the FF structure we have previously generated using 5×200 steps of 3T-FF. This structure is converted into an ASE atoms object before being attached onto an ASE VASP calculator and passed into a Sella optimizer. We use the ASE calculator input settings for Sella-VASP which generate the same VASP input files (INCAR, KPOINTS, POTCAR) that are used for the 3T-VASP optimizations. We instruct Sella to work on optimization (*order=0*), utilize its internal coordinates (*internal=True*), and run the structure energy minimization for 5×50 steps, just like 3T-VASP (to see if Sella can also benefit from inter-cycle transition like 3T-VASP). We do this for 3 randomly chosen 3T-FF reduction trajectories (**Supplementary Figure 18**), and for 3 randomly chosen 3T-FF oxidation trajectories (**Supplementary Figure 19**). These Sella-VASP trajectories are directly compared to the corresponding 3T-VASP trajectories which were generated from the same final 3T-FF structures. As a general summary, we observe the following trends:

1. Sella-VASP final structures always have higher energies than 3T-VASP final structures (0.63–13.46 eV difference observed)
2. 3T-VASP reactions are typically finished within less DFT calls compared to Sella-VASP reactions
3. 3T-VASP tends to generate the expected common reactions more frequently, while Sella-VASP has higher probabilities of generating more rare reactions. This is likely because Sella-VASP tends to end up with higher-energy structures compared to 3T-VASP. This has advantage (rare reaction exploration) and disadvantage (statistically less meaningful, unphysical/ unfinished reactions), as we will later discuss.
4. Sella-VASP always utilizes more than the 5×50 DFT calls instructed to Sella. From our observation, this is likely related to the introduction/modification of dummy atoms throughout the Sella-VASP optimization.

Supplementary Figure 18 | Comparison between Sella and 3T relaxations for VASP electrolyte reduction reaction exploration. (a) Comparison of the structure energy vs VASP call when 3 different 31-FF final structures are used as the initial structures for relaxation. Please refer to **Supplementary Figure 7** for the exact reaction types we have annotated. **(b)** Novel reactions generated by Sella-VASP which have not been seen in prior 31-VASP electrolyte reduction trajectories.

We first discuss the Sella-VASP electrolyte reduction experiment. From the **Supplementary Figure 18** above, we can see that the energy minimization profile curve of Sella-VASP is not very stable and settles at relatively high energy (3.12–13.46 eV higher) compared to the identical initial structure minimized using 3T-VASP. It seems that the current Sella-VASP bulk electrolyte reduction structure minimization in the periodic boundary condition box tends to get stuck in higher local energy minima, with occasional attempts to escape the minima which results in very high energy spikes before quickly settling back down. As of August 7, 2024, the version of Sella publicly available to us for usage from Github generates very large energy spikes up to more than 200 eV larger compared to the final relaxed Sella-VASP energy values. This behavior is somewhat different than the smooth Sella-NWChem energy minimization curve for single molecule (**Supplementary Figure 17**), and we are unsure whether this is an expected feature of Sella-VASP bulk energy minimization or not. These high energy structures may contribute to the generation of unphysical reactions. While some of the Sella-VASP reactions we observe are expected (they correspond to 3T-VASP reactions **a**, **c**, and **k** from **Supplementary Figure**

7), we also observe two concerning new reactions (observed in the trajectory ID=0x75c830e125132dfc on **Supplementary Figure 18b**, where Sella-VASP final structure energy is significantly higher (+13.46 eV) compared to the 3T-VASP final structure energy). The first reaction between C and VC generates bicyclo[1.1.0]butane (C_4H_6), which is a highly strained organic molecule that we think is an unlikely electrolyte electrochemical reduction reaction byproduct. The second reaction between 2 DMC and PF_6^- generates many reaction byproduct molecules, including C_2H_6 . According to experimental literature, C_2H_6 should not be observed as an electrolyte reduction decomposition byproduct gas when C: DMC electrolyte mixture is used. In contrast, C_2H_6 is never observed as a 3T-VASP byproduct reaction in our 63 electrolyte reduction trajectories, in line with experimental observation. We note that while reaction c and k are also observed in 3T-VASP, they are supposed to be very infrequent (**Supplementary Table 1**). However, these reactions are easily obtained by Sella-VASP within just 3 trajectories. This indicates that Sella-VASP can generate rare reactions more frequently because it is often stuck at higher local energy minimums. This may be a Sella-VASP advantage when rare reaction exploration is desired. On the other hand, 3T-VASP tends to explore lower-energy structures.

Supplementary Figure 19 | Comparison between Sella and 3T relaxations for VASP electrolyte oxidation reaction exploration. (a) Comparison of the structure energy vs VASP call when 3 different 3T-FF final structures are used as the initial structures for relaxation. Please refer to **Supplementary Figure 9** ($\sigma \cdot$) & **10** ($O \cdot$) for the exact reaction types

we have annotated. (b) Novel reactions generated by Sella-VASP which have not been seen in prior 3T-VASP electrolyte oxidation trajectories.

However, the situation is very different for Sella-VASP oxidation reaction trajectories (**Supplementary Figure 19**). While the observed trend holds where Sella-VASP final structure energies are always higher than the corresponding 3T-VASP final structure energies (0.63–5.73 eV higher), we no longer observe the large (>200eV) energy spikes like the ones we observe in Sella-VASP electrolyte reduction trajectories. Sella-VASP still generates reactions more slowly than 3T-VASP, and just like before it can find reactions which are less commonly encountered in 3T-VASP, such as **-(g)** and **-(n)** oxidation reactions (**Supplementary Figure 8, Supplementary Table 1**). However, we no longer observe unphysical reactions. The only reaction which looks somewhat concerning is an $\text{-}\bullet$ decomposition reaction which produces a hydrogen radical. We have observed this structure before, which is a typical 3T-VASP intermediate byproduct which appears before an $\text{-}\bullet$ containing reaction is finished. We believe that this is just an unfinished intermediate byproduct because Sella-VASP needs more steps before it can complete the reaction. We also note that the best-performing Sella-VASP energy minimization (ID=0x51f72a19e29afadd, where the final structure energy is only 0.63 eV above 3T-VASP) ends up generating identical reaction byproducts to the corresponding 3T-VASP version, although the reactions still occur more slowly compared to 3T-VASP.

This leads us to conclude that Sella-VASP, as currently available on August 7, 2024, is not very suitable for exploring electrochemical reduction reactions in bulk electrolyte using periodic boundary condition code such as VASP, as it generates relatively high energy structures which look less physical compared to 3T-VASP. For oxidation reactions, Sella-VASP is relatively competitive to 3T-VASP although it still requires more DFT calls to produce the desired reactions. When low-energy structure bulk electrolyte structure exploration is desired, 3T-VASP is more advantageous than Sella-VASP. However, if high-energy rare structure exploration is desired instead, Sella-VASP's tendency to be trapped in a higher local energy minimum may be advantageous compared to 3T-VASP (although more care is needed to ensure that the generated structures are physical).

Finally, we note that while Sella-VASP has been instructed to perform 5×50 steps of VASP energy minimizations, in practice it seems to automatically perform a few additional VASP calls in each cycle (making the total step count to be more than 250 steps). These extra VASP calls seem to happen when Sella ran into invalid angle internal coordinate problems and need to update the structure using new dummy atoms. This problem does not occur in 3T-VASP.

Reviewer #4 (Remarks on code availability):

I thank the authors for their additional work on the code based on my comments.

We thank the reviewer for this positive comment.

Reviewer #1

I would like to thank the authors for their efforts to address my concerns — these have now been addressed sufficiently. Overall, I feel that the manuscript has been improved significantly over these rounds of revisions by addressing the comments of all reviewers. It is now also easier to evaluate the impact of 3T-VASP, as the manuscript contains a more appropriate comparison with existing methods. I believe that 3T-VASP is an interesting and important development, with implications on prediction of reaction outcomes in complex environments.

We thank the reviewer for this positive comment and for the thorough feedback on our various manuscript drafts, which has strengthened our manuscript considerably.

Comment 1:

Just one small comment: in the "Discussion on 3T Comparison with Sella for Bulk Electrolyte Electrochemical Reactions" section of the revised supporting information, the authors say "We instruct Sella to work on optimization (order=0)". Even if order=1, it performs an optimization, but to a first order saddle point instead of a minimum. Instead of saying just "optimization", the authors should specify that it is an optimization to a minimum.

We thank the reviewer for this suggestion.

- **Supplementary Information page 37, line 14.** We modified the sentence:

We instruct Sella to work on optimization to an energy minimum (order=0), utilize its internal coordinates (internal=True), and run the structure energy minimization for 5×50 steps, just like 3T-VASP (to see if Sella can also benefit from inter-cycle transition like 3T-VASP).

Reviewer #2

I appreciate the work the authors have done to improve their manuscript. But in my opinion, this is still not a good fit for that journal. I don't see a big impact in general. In my opinion, such a paper should be a longer paper in a more technical and specific journal, not a communication with so much supplementary materials.

We thank the reviewer for the thorough feedback on our various manuscript drafts, which has strengthened our manuscript considerably.

Reviewer #4

I thank the authors for their additional work in response to my comments. I am now satisfied and support the publication of the manuscript in Nature Communications.

We thank the reviewer for this positive comment and for the thorough feedback on our various manuscript drafts, which has strengthened our manuscript considerably.